# Reviews and syntheses: $^{210}$Pb-derived sediment and carbon accumulation rates in vegetated coastal ecosystems - setting the record straight

Ariane Arias-Ortiz[1*], Pere Masqué[1,2,3,4], Jordi Garcia-Orellana[1,2], Oscar Serrano[3], Inés Mazarrasa[5], Núria
Marbà[6], Catherine E. Lovelock[7], Paul S. Lavery[3,8] and Carlos M. Duarte[9]

[1]Institut de Ciència i Tecnologia Ambientals, Universitat Autònoma de Barcelona, Bellaterra, 08193 Barcelona, Spain.
[2]Departament de Física, Universitat Autònoma de Barcelona, Bellaterra, 08193 Barcelona, Spain.
[3]School of Science and Centre for Marine Ecosystems Research, Edith Cowan University, 270 Joondalup Drive, Joondalup
WA 6027, Australia.
[4]UWA Oceans Institute & School of Physics, The University of Western Australia, 35 Stirling Highway, Crawley 6009,
Australia.
[5]Environmental Hydraulics Institute "IH Cantabria", Universidad de Cantabria, C/ Isabel Torres N∘15, Parque Científico y
Tecnológico de Cantabria, 39011, Santander, Spain.
[6]Global Change Research Group. IMEDEA (CSIC-UIB) Institut Mediterrani d'Estudis Avançats, C/ Miguel Marqués 21,
07190 Esporles (Mallorca), Spain.
[7]School of Biological Sciences, The University of Queensland, St Lucia, QLD 4072, Australia.
[8]Centro de Estudios Avanzados de Blanes, Consejo Superior de Investigaciones Científicas. Blanes, Spain 17300.
[9]King Abdullah University of Science and Technology (KAUST), Red Sea Research Center (RSRC), Thuwal, 23955-6900,
Saudi Arabia.

*Correspondence to*: Ariane Arias-Ortiz (ariane.arias@uab.cat)

**Abstract.** Vegetated coastal ecosystems, including tidal marshes, mangroves and seagrass meadows, are being increasingly assessed for their potential in carbon dioxide sequestration worldwide. However, there is a paucity of studies that have effectively estimated the accumulation rates of sediment organic carbon ($C_{org}$), also termed blue carbon, beyond the mere quantification of $C_{org}$ stocks. Here, we discuss the use of the $^{210}$Pb dating technique to determine the rate of $C_{org}$ accumulation in these habitats. We review the most commonly used $^{210}$Pb dating methods and assess the limitations in applying them to these ecosystems, which are often composed by heterogeneous sediments, with varying inputs of organic material, and are disturbed by natural and anthropogenic processes causing sediment mixing, changes in sedimentation rates or erosion. Through a range of simulations, we consider the most relevant processes that impact the $^{210}$Pb records in vegetated coastal ecosystems and evaluate the deviations in sediment and $C_{org}$ accumulation rates produced by anomalies in the $^{210}$Pb concentration profiles. Our results show that the deviations in sediment and derived $C_{org}$ accumulation rates relative to those estimated at undisturbed profiles are within 20% if the process causing the anomalies in $^{210}$Pb profiles is well understood. While these uncertainties might be acceptable for the determination of mean sediment and $C_{org}$ accumulation rates over the last century, they may not always allow the determination of a credible geochronology or historical reconstruction. Calculations of accumulation rates, however, might be difficult or impossible at sites with slow accumulation rates and intense mixing, and errors in the

identification of the processes responsible may lead to deviations of up to 30 to 100%. Additional tracers or geochemical, ecological or historical data need to be used to constrain the $^{210}$Pb-derived results and to properly interpret the processes recorded in vegetated coastal sediments. The framework provided in this study can be instrumental in reducing the uncertainties associated with estimates of $C_{org}$ accumulation rates in vegetated coastal sediments.

**Keywords:** $^{210}$Pb, vegetated coastal sediments, carbon accumulation rates, sediment dating, blue carbon.

## 1 Introduction

Recognition of the globally significant role of vegetated coastal habitats, including tidal marsh, mangrove and seagrass, as sinks of carbon dioxide ($CO_2$) (Duarte et al., 2013) has led to a rapid growth in the interest to evaluate the amount of organic

carbon ($C_{org}$) these ecosystems sequester, in order to quantify the potential to mitigate $CO_2$ emissions through their management in an approach described as "*Blue Carbon*" (Duarte et al., 2013; Mcleod et al., 2011; Nellemann et al., 2009). However, efforts to include vegetated coastal ecosystems into existing carbon mitigation strategies have met with an important limitation: there is a paucity of estimates of $C_{org}$ sequestration rates, particularly in seagrass habitats (Johannessen and Macdonald, 2016, 2018; Macreadie et al., 2018).

Two interrelated measurements of importance are the sediment $C_{org}$ content and the sedimentation velocity or sedimentation rate. To date, most of the research has focused in the first term, which informs about the $C_{org}$ stock sequestered in sediments (Howard et al., 2014; Pendleton et al., 2012). However, $C_{org}$ stocks alone cannot be used to fully assess the $C_{org}$ storage capacity or to establish comparisons among sites. Measurements of $C_{org}$ accumulation rates (CAR) address the question of how much $C_{org}$ is sequestered in a specified time period and quantify the ongoing sink capacity. In general, CAR is obtained by measuring

the concentration of $C_{org}$ in sediments and ascribing dates to either the entire profile of interest or to specific intervals, or by estimating sediment accumulation rates. Determination of mean CAR is partially dependent on the time scale of interest and the dating methods used. $^{210}$Pb, with a half-life of 22.3 yr, has been shown to be an ideal tracer for dating aquatic sediments deposited during the last ca. 100 yr, providing a time frame compatible with management actions (Marland et al., 2001) and enabling the determination of CAR and its changes with time due to natural or human impacts. Due to the relatively long

integration period (decades to a century), mean $^{210}$Pb-derived CAR estimates are not affected by interannual variability, hence allowing the assessment of shifts from the "baseline" condition (i.e., the $C_{org}$ that naturally cycles through an ecosystem; Howard et al., 2017). Although several review papers have elaborated the applications of excess $^{210}$Pb as a tracer in lacustrine and marine environments (Appleby, 2001; Baskaran et al., 2014; Du et al., 2012; Kirchner and Ehlers, 1998; Mabit et al., 2014; Sanchez-Cabeza and Ruiz-Fernández, 2012; Smith, 2001), little attention has been paid to the potential limitations of the $^{210}$Pb

dating method in vegetated coastal sediments. Experience shows that vegetated coastal environments often prove to be more challenging than lake or marine sediments (Saderne et al., 2018).

Vegetated coastal ecosystems may act as closed systems, where the sediment accumulation is mainly associated with the build-up of autochthonous organic and inorganic material (McKee, 2011). In this situation, excess $^{210}$Pb is deposited primarily from atmospheric fallout at steady state, with no post depositional mobility except for physical or biological mixing of the sediments (e.g. Alongi et al., 2004; Cochran et al., 1998; Marbà et al., 2015). In some cases, however, the process responsible for incorporating excess $^{210}$Pb into the sediments might be more complex. Vegetated coastal ecosystems may receive both autochthonous and allochthonous sediments from the upstream catchment, coastal erosion or from the offshore zone during storm events (Turner et al., 2007), or in response to land use change (Mabit et al., 2014; Ruiz-Fernández and Hillaire-Marcel, 2009). Their sediments might be reworked through the action of fauna (bioturbation), tides, currents, and waves as well as through boat anchoring, dredging or fishing activities (e.g. Mazarrasa et al., 2017; Sanders et al., 2014; Serrano et al., 2016; Smoak et al., 2013). Effects associated with climate change, such as sea level rise and extreme climatic events, may also have an impact on rates of production and decomposition of organic matter (OM) and on sediment and $C_{org}$ accumulation (Alongi et al., 2008; Arias-Ortiz et al., 2018; Mudd et al., 2010). In such instances, sediment redistribution processes and complex accretion dynamics may violate some of the assumptions of $^{210}$Pb dating models, producing anomalous $^{210}$Pb concentration profiles that are difficult to interpret.

Sediments of vegetated coastal ecosystems are known to be heterogeneous, consisting of coarse grained sediments or bedrock covered by deposits of fine grained sediments that settled as vegetation established (McGlathery et al., 2012; Olff et al., 1997). The percentage of living (e.g. roots) and recently formed organic material is greatest in the upper 10 cm and may be affected by varying inputs of detrital sediment within vegetated coastal ecosystems and by its relative rate of decomposition. While tidal marsh and mangrove sediments have relatively high organic matter content (on average 25%) (Breithaupt et al., 2012; Cochran et al., 1998), mineral deposits account for the majority (>85%) of the accumulated substrate in seagrass sediments (Koch, 2001; Mazarrasa et al., 2015) (Table 1). Excess $^{210}$Pb has a strong affinity for fine sediments (Chanton et al., 1983; Cundy and Croudace, 1995; He and Walling, 1996a) and organic matter (Wan et al., 2005), thus any changes in these parameters due to sediment redistribution processes or to natural heterogeneity may also result in unique types of $^{210}$Pb concentration profiles in sediment cores of vegetated coastal ecosystems, adding complexity to the determination of sediment model age and sedimentation rates.

Here, we present how the processes of mixing, changes in the sedimentation rate, erosion, grain size heterogeneity and OM decay impact the depth distribution of excess $^{210}$Pb in vegetated coastal sediments and assess the deviations (relative to an ideal undisturbed profile) in estimated sediment and $C_{org}$ accumulation rates produced by anomalies in $^{210}$Pb profiles. First, we provide a critical review of the current status of $^{210}$Pb dating methods of vegetated coastal sediments. Then, through a set of simulations, based on examples from the literature and using various $^{210}$Pb dating models, we assess the limitations that apply to the determination of last century $C_{org}$ accumulation rates in such ecosystems. Finally, we provide guidance on

complementary analyses to accompany the [210]Pb dating technique that can improve sediment and derived $C_{org}$ accumulation rates estimates.

## 1.1 [210]Pb dating models

The [210]Pb dating method is based on the principle that excess [210]Pb, produced as a result of [222]Rn decay in the atmosphere and subsequent fallout, is deposited at a supposedly constant rate (over an integration period of years), directly onto the surface of soils and sediments or indirectly, via the water column. [210]Pb is particle-reactive in the marine environment, hence, once in the water it rapidly settles in the sediment, bound to particulate matter (Robbins, 1978). The subsequent burial, with simultaneous radioactive decay (0.0311 yr[-1]), ideally generates a decreasing distribution of [210]Pb specific activity as a function of depth (or preferably, cumulative mass in g cm[-2], to allow for the effects of compaction) (Fig. 1). Most sediments also contain supported [210]Pb, which is part of the sediment matrix and is in equilibrium with [226]Ra. [210]Pb-derived sediment chronologies are based in the interpretation of the rate of decline of excess [210]Pb concentrations with depth in a sediment core. Under ideal circumstances, [210]Pb is able to accurately date sediments back to about 7 half-lives, i.e. about 150 years (the "dating horizon"), where the measurement uncertainty becomes too large to detect any excess [210]Pb. However, chronologies reaching back that far might be rarely achievable in vegetated coastal sediments as these contain relatively low concentrations of [210]Pb. The basis of the distribution of excess [210]Pb in sediments can then be described as (Koide et al., 1972):

$$\frac{\partial \rho C}{\partial t} = \frac{\partial}{\partial z} \cdot \left( D_b \rho \frac{\partial C}{\partial z} \right) - \frac{r \, \partial \rho C}{\partial z} - \lambda \rho C \qquad \text{(Eq. 1)}$$

where $\rho$ is sediment bulk density (g cm[-3]), $C$ is the concentration of excess [210]Pb (Bq kg[-1]), $z$ is depth below the sediment–water interface (cm), $D_b$ is a coefficient characterizing the sediment mixing rate (cm[2] yr[-1]), $r$ is the sedimentation rate (cm yr[-1]), $\lambda$ is the [210]Pb decay constant (yr[-1]) and $t$ is time (yr). Commonly, depth ($z$) is represented as mass depth ($m$) to correct for compaction. Mass depth (g cm[-2]) results from the multiplication of z and $\rho$, and sedimentation rates are expressed as mass accumulation rates (MAR) in g cm[-2] yr[-1], which can be described as $MAR = \rho(v + q)$ where $v$ and $q$ are the accretion and compaction velocities, respectively (Abril, 2003b) (Eq.2).

$$\frac{\partial C}{\partial t} = \frac{\partial}{\partial m} \left( k_m \frac{\partial C}{\partial m} \right) - MAR \frac{\partial C}{\partial m} - \lambda C \qquad \text{(Eq. 2)}$$

where $k_m$ an effective mixing coefficient (g[2] cm[-4] yr[-1]).

The [210]Pb technique was first applied by Koide et al. (1972) to date marine sediments. Since then, a family of dating models has been used to interpret the excess [210]Pb depth distribution in marine and freshwater sediment cores, increasing in variety and complexity and involving a large diversity of post-depositional redistribution processes (Table 2). However, there are three models that are most widely used and described here: the Constant Flux : Constant Sedimentation (CF:CS) model (Krishnaswamy et al., 1971), the Constant Rate of Supply (CRS) model (Appleby and Oldfield, 1978) and the Constant Initial Concentration model (CIC) (Robbins, 1978). Although these three models each have specific assumptions, they share the following: (1) the deposition of excess [210]Pb is at steady state, (2) there is no post depositional mobility of [210]Pb, (3) the

deposition of excess [210]Pb is ideal, i.e., new radioactive inputs are deposited above the previously existing material, and (4) the sedimentary sequence is continuous. In the simplest of the cases, the CF:CS model assumes constant excess [210]Pb depositional flux and sedimentation rate and can be applied to downcore profiles to derive the mean accumulation rate. In this case the [210]Pb specific activity at the surface ($C_0$: Bq kg$^{-1}$) is constant and decreases exponentially with cumulative mass. The depth of burial $m$ is related to the elapsed time since burial through the rate of sedimentation ($MAR$) (Table 2). If there is mixing at the surface of the core, the mean MAR can be calculated from the excess [210]Pb concentration profile below the surface mixed layer (SML). If the concentrations of excess [210]Pb decline in sections, showing two or more exponentially decaying segments, then, a mean MAR can be derived for each segment (Goldberg et al., 1977). In this way the model is, to some degree, able to cope with temporal variations in sedimentation rate.

Variations in accumulation rate may occur in response to natural processes or anthropogenic influences. Under some such circumstances, the CRS or CIC models could be suitable. The CRS model assumes a constant flux of [210]Pb ($\Phi$) to the sediments over time (Table 2). The initial specific activity is variable and inversely related to MAR (higher MAR leads to lower excess [210]Pb concentrations and *vice versa*). The dating is based on the comparison of excess [210]Pb inventories ($A_m$; Bq m$^{-2}$) below a given depth (integration of excess [210]Pb specific activity as a function of the cumulative mass) with the overall excess [210]Pb inventory in the sediment core ($I$). Variations in $A_m/I$ are related to variations in $MAR$. The accurate determination of the [210]Pb inventories is of critical importance and required for the application of the CRS model (Appleby, 2001).

If the flux of excess [210]Pb is expected to vary with time, the CIC could be a better choice. This could be the case at locations where sediment focusing is a major factor, where event-deposit layers are present, or if significant hydrologic changes have occurred or there are hiatuses in the sediment record caused by erosion events (Appleby, 2008). The CIC model assumes that the initial concentration of excess [210]Pb at the sediment-water interface is constant with time irrespective of the sedimentation rate so that the excess [210]Pb flux co-varies with MAR. This model permits estimation of the age ($t$) at any depth where [210]Pb has been measured ($C_m$) if the initial specific activity $C_0$ is known (Table 2). However, the CIC model requires a monotonic decrease of excess [210]Pb concentrations down-core for age-reversals to be avoided, which is rare in most vegetated coastal sediments. In that event, the calculation of mean accumulation rates alone using the CF:CS model would be a more reasonable approach, as it might be too ambitious to calculate a detailed stepwise chronology based on often limited number of data points decreasing monotonically.

While the CIC or CF:CS models have been typically used in the marine environment, the CRS model is the most preferred in lake sediments and it is becoming widespread applied in estuarine environments and vegetated coastal ecosystems (Andersen, 2017; Breithaupt et al., 2014). Some of the reasons could be that it suffers less from problems associated with non-monotonic features in the [210]Pb record and is relatively insensitive to mixing (Appleby, 2008; Appleby et al., 1983; Appleby and Oldfield, 1992; Oldfield et al., 1978). The selection and use of a specific model should be based on the nature of the excess [210]Pb specific

activity and sediment accumulation. For further details on the main aspects relevant to the application of [210]Pb dating models in lake or estuarine environments we recommend two detailed and comprehensive papers by Appleby (2001) and Andersen (2017). Here, we focus specifically on analysis of [210]Pb dating of sediments in vegetated coastal ecosystems.

## 2 Methods

We performed a literature review of studies on sediment accumulation in vegetated coastal ecosystems in the Web of Science[TM] (accessed August 23, 2018) with the keywords mangrove sediment, salt marsh OR saltmarsh OR tidal marsh sediment, seagrass sediment AND [210]Pb OR Pb-210 OR lead-210. The search produced 86, 223 and 27 results, respectively, all of them using one or more of the three models described above, probably due to its simplicity, with the exception of Klubi et al. (2017) that additionally uses the TERESA model (Table 2). From the literature review we identified the most common sedimentary

processes that result in anomalous types of excess [210]Pb concentration profiles with depth (Fig. 2). These could be summarized in five main processes: mixing, increasing sedimentation, erosion, changes in sediment grain size, and decay of organic matter (OM). Then, we simulated the target processes on initial undisturbed seagrass, mangrove and tidal marsh [210]Pb sediment concentration profiles to determine the potential deviations in MAR (defined as the difference between the value which has been computed and the correct value) and analyse the limitations of the [210]Pb dating technique in these ecosystems to derive

CAR.

### 2.1 Numerical simulations

All simulations started from an ideal excess [210]Pb profile, complying with all assumptions, that was then manipulated to reflect the potential effect of each process. The ideal excess [210]Pb profile was modelled considering the following: (1) a constant flux of excess [210]Pb ($\Phi$) of 120 Bq m[-2] yr[-1] i.e., the average global atmospheric flux reported by Preiss et al. (1996); (2) a MAR of

0.2 g cm[-2] yr[-1] and dry bulk density (DBD) of 1.03 g cm[-3] to represent seagrass sediments; and (3) a MAR of 0.3 g cm[-2] yr[-1] and DBD of 0.4 g cm[-3] to represent mangrove/tidal marsh sediments based on typical values representative of these ecosystems (Duarte et al. 2013) (Table 1). Simulated surface activity per unit area of excess [210]Pb ($A_0$; in Bq m[-2]) in ideal profiles was estimated through equation 3. Then equation 4 was applied to estimate excess [210]Pb activities per unit area along the ideal profile (Supplementary, Table 1).

$$A_0 = \frac{\Phi}{\lambda}\left(1 - e^{-\lambda\, m_0/MAR}\right) \tag{3}$$

$$A_m = A_0 \cdot e^{-\lambda\, m/MAR} \tag{4}$$

Activities of excess [210]Pb per unit area ($A_m$) were then converted to concentrations, $C_m$ in Bq kg[-1], by dividing $A_m$ by the

cumulative mass ($m$) at each layer. Ideal profiles were then altered to simulate the following processes/scenarios: mixing (surface and deep mixing), increasing sedimentation (by 20%, 50%, 200% and 300%), erosion (recent and past), changes in

sediment grain size (coarse and heterogeneous) and OM decay (under anoxic and oxic conditions, and with labile OM contribution in sediments containing 16.5% and 65% OM) (Table 3). Refer to Appendix A for a detailed description of the methodology used to conduct each simulation.

The CF:CS and CRS dating models were applied to the simulated excess [210]Pb profiles to determine the average MAR for the last century (Table 2). The CIC model was excluded from the simulations presented in this study because in anomalous excess [210]Pb profiles: 1) the CRS model would lead to more reasonable approaches when the flux of excess [210]Pb is constant; and 2) when that is not the case (e.g., simulations of erosion or heterogeneous grain size), determination of mean accumulation rates alone by the CF:CS model would be a more reasonable approach. The models were applied in accordance with the simulated

process. For instance, MAR was determined below the surface mixed layer in mixing simulations using the CF:CS, and piecewise in those with a change in average MAR (Appendix A). However, the models were also applied considering that (1) excess [210]Pb profiles of mixing simulations were generated by increasing MAR and *vice versa*, and (2) erosion was not a factor in simulated scenarios (H-J). This was done to test the potential deviations in MAR and derived CAR if the incorrect process was assumed and dating models were applied. Once the dating model was established, the $C_{org}$ accumulation rate (CAR) was

estimated through equation 5 assuming average sediment $C_{org}$ contents of 2.5% in seagrass and 8% in mangrove/tidal marsh, in both ideal and simulated sediment profiles. Under ideal conditions, CAR rates were 50 g $C_{org}$ m$^{-2}$ yr$^{-1}$ and 240 g $C_{org}$ m$^{-2}$ yr$^{-1}$ in seagrass and mangrove/tidal marsh sediments, respectively. While this overall model structure was used in all simulated scenarios, MAR and CAR rates under ideal conditions varied from those reported above in increasing sedimentation and OM decay simulations to represent real increases in accumulation, changes in OM content and associated losses of sediment mass with depth (Table 3).

with depth (Table 3).

$$CAR = \frac{\sum_{n=i}^{t} (\%C_{org_i} \cdot m_i)}{m_t} \cdot MAR_t \qquad \text{(Eq. 5)}$$

where $(\%C_{org_i} \cdot m_i)$ is the mass per unit area of $C_{org}$ at layer $i$ (g $C_{org}$ m$^{-2}$), $m_t$ is the cumulative mass over the period $(t)$ (g m$^{-2}$) and $MAR_t$ is the mass accumulation rate of the period of interest $(n\text{-}t)$ (g m$^{-2}$ yr$^{-1}$). When *CAR* is examined over the last 100 years, $m_t$ is the cumulative mass down to the excess [210]Pb horizon (i.e., depth where excess [210]Pb concentrations approach zero)

and $MAR_t$ is the mean mass accumulation rate.

## 3 Results and Discussion

### 3.1 Types of excess [210]Pb concentration profiles

Seven distinct types of excess [210]Pb concentration profiles can be identified in vegetated coastal sediments based on examples

from the literature (Fig. 2). Type I is produced by constant sediment accumulation in steady state conditions (i.e. 'ideal' profiles). The other six types of excess [210]Pb concentration profiles summarize the most common disturbances encountered in

vegetated coastal sediments that are related to the presence of mixing (physical or bioturbation), increasing MAR, erosion, or alteration by intrinsic features of sediments such as heterogeneous grain size distribution and decay of OM.

- Type II illustrates a moderate decrease in the slope of excess [210]Pb concentrations in the upper part of the sediment core, which is often attributed to higher MAR (Cearreta et al., 2002; Haslett et al., 2003; Swales and Bentley, 2015), but can also be related to a mixing process (Gardner et al., 1987).

- Type III, showing constant excess [210]Pb concentrations along the upper part of the core overlaying an exponential decaying trend, is usually interpreted as the outcome of mixing as a result of bioturbation or sediment resuspension, re-deposition and reworking (Jankowska et al., 2016; Sanders et al., 2010a; Serrano et al., 2016a; Sharma et al., 1987; Smoak and Patchineelam, 1999). In some instances this profile type has also been related to rapid accumulation of homogeneous sediment (Walsh and Nittrouer, 2004).

- Type IV profiles show a reverse excess [210]Pb pattern at surface and have been attributed to a variety of factors. Similar to type III, these profile types can be caused by mixing processes in vegetated coastal ecosystems (Sanders et al., 2010b; Serrano et al., 2016a; Yeager et al., 2012) or by the deposition of allochthonous older material (Johannessen and Macdonald, 2018). However, they could also be produced by an acceleration of the sedimentation rate, as interpreted by Greiner et al. (2013), Smoak et al. (2013) and Bellucci et al. (2007) in seagrass, mangrove and tidal marsh, respectively, or by the decay of OM, as modelled by Chen and Twilley (1999) and Mudd et al. (2009), and observed by Church et al. (1981) in tidal marsh sediments containing > 30% OM in top layers. Additionally, type IV profiles could also be explained by non-ideal deposition (i.e. a fraction of the new excess [210]Pb input onto the sediment is not retained at the surface but penetrates to deeper layers), a process reported in peatlands and in sediments with very high porosities (> 90%) at the sediment-water interface (Abril and Gharbi, 2012; Olid et al., 2016).

- Type V profiles show scattered excess [210]Pb concentrations, which might reflect periodic occurrence of processes that can cause type III or IV profiles and often are interpreted as evidence of repetitive reworking in the overall mixed sediment column (Alongi et al., 2001; Serrano et al., 2016a; Smoak and Patchineelam, 1999). However, this profile form has also been explained by the deposition of excess [210]Pb outpacing its decay ($\lambda = 0.03111$ yr$^{-1}$) (Alongi et al., 2005) or by a heterogeneous grain-size sediment distribution with depth (Chanton et al., 1983; Kirchner and Ehlers, 1998; Sanders et al., 2010a), which could indicate varying excess [210]Pb fluxes due to flood events, major land use-changes or changes in vegetation cover (Appleby, 2001; Marbà et al., 2015).

- Types VI and VII represent low excess [210]Pb activities with depth, apparently showing low, negligible modern net accumulation of sediments. Such profiles are usually related to an abundance of coarse sediments or to erosion processes, as shown in tidal marsh sediments (Ravens et al., 2009) and bare sediments that were previously vegetated with seagrass in Greiner et al. (2013), Marbà et al. (2015) and Serrano et al. (2016c).

These examples identified from the literature reveal that various sedimentary processes might produce similar types of excess [210]Pb concentration profiles. Any particular excess [210]Pb concentration profile can accommodate a range of mathematical

modelling approaches (see below), which lead to development of differing chronologies and MAR estimates. Hence, the identification of the process driving accumulation and causing variation in the excess [210]Pb record aids in the determination of the $C_{org}$ accumulation rates.

## 3.2 Simulated sediment and $C_{org}$ accumulation rates (MAR and CAR)

We ran simulations for sedimentary processes (mixing, enhanced sedimentation, erosion) and heterogeneous sediment composition with depth (grain size distribution and OM decay). Results of the modelled excess [210]Pb profiles are summarized in Figures 3 and 4 and Supplementary Tables 1-7. We estimated mean 100-yr MAR and CAR for the simulated profiles by applying the CF:CS and CRS models, and results were compared with those from their respective ideal non-disturbed [210]Pb profiles. The estimated deviations in accumulation rates from those expected under ideal conditions are shown in Figure 5 for seagrass and mangrove/tidal marsh ecosystems. These deviations are driven by variations in MAR estimates caused by anomalies in [210]Pb concentration profiles as the $C_{org}$ fraction $\left(\frac{\sum_{n=i}^{t} (\%C_{org_i} \cdot m_i)}{m_t}\right)$ was considered to be the same in both ideal and simulated sediment profiles.

### 3.2.1 Mixing

Simulations of surface mixing (A and B in Fig. 3a) yielded [210]Pb concentrations profiles similar to types II and III (Fig. 2), while deep mixing (scenario C) led to stepwise excess [210]Pb profile forms similar to type V. Calculated MAR and CAR deviated by up to 80% from the expected value in seagrass sediments, while deviations were negligible (≤3%) in mangrove/tidal marsh sediments due to the smaller proportion (5 - 10%) of the excess [210]Pb profile affected by mixing (Fig. 5a and 5c). In both cases, higher deviations from the expected rates were associated with deep mixing and with the use of the CF:CS model, since this model interprets any divergence from the 'ideal' exponential decrease of the excess [210]Pb concentration with depth to reflect random variation. In contrast, the CRS model is based on the excess [210]Pb inventory ($I$), that is unaffected by vertical mixing.

Profiles of mixing in sediments could be equally explained by an increase in the sedimentation rate in recent years. If the incorrect process is assumed and inappropriate dating models are applied, mean MAR and CAR would be largely overestimated in seagrass sediments, by 20, 30 and 95%, using the CF:CS model in surface (scenario A, B) and deep mixing simulations, respectively (Fig. 5b). In mangrove/tidal marsh sediments, overestimation in mean MAR and CAR was substantial (30%) when deep mixing was considered to be caused by an increase in MAR (Fig. 5d). A process mismatch between mixing and increased sedimentation in recent years did not cause large deviations (between 2 and 5%) in MAR and CAR derived by the CRS model. The CRS model outputs are similar if mixing or changes in accumulation rates are present, albeit ages within the mixed layer cannot be reported if mixing occurs.

### 3.2.2 Increasing sedimentation rates

Simulated increases in sedimentation rates from 20% to 300% (scenarios D to G, Fig. 3b) resulted in similar profile forms as those simulated with surface mixing. Increases in sedimentation rates were modelled over the last 30 yr, a period over which more than a 2-fold increase was needed to produce a reversal of excess [210]Pb concentrations with depth under the conditions of this simulation (type IV profiles; Fig. 2). The influence of change in the sedimentation rate was better captured with the CRS model. The CF:CS model, in contrast, failed to account for rapid and large increases in MAR. Deviations from the expected value ranged from 0 to 15% in scenarios D and E (20% to 50% increase in MAR) and were up to 60% for a 100% increase in MAR (scenario F). Calculated MAR in scenario G (200% increase in MAR) was underestimated by a 30%, as piecewise dating is not applicable in profiles with constant or reversed concentrations of excess [210]Pb with depth. In such situations, additional tracers or times markers are required to estimate MAR and CAR in the layer of constant excess [210]Pb concentrations (see section 4.2). Deviations from the expected value ranged from 0 to 4% when using the CRS model (Fig. 5a and 5c). Results were similar for both ecosystem types. If the recent increase in MAR was interpreted as mixing, the mean MAR and CAR would be underestimated between 10 and 30% in both habitat types using the CF:CS model (Fig. 5b and 5d). In contrast, deviations from the ideal value were $\leq 5\%$ if the CRS model was applied.

### 3.2.3 Erosion

We ran three simulations (H, I and J) to represent recent (H) and past erosion events (I and J) (Fig. 3c). Simulations of erosion yielded lower excess [210]Pb concentrations than those of the 'ideal' reference profile (type VII, Fig. 2), and excess [210]Pb dating horizons were found at shallower depths in these simulations (Fig. 3c). Consequently, excess [210]Pb inventories ($I$) in eroded profiles were lower than expected (reference ideal profile $I_{ref}$: 3900 Bq m$^{-2}$). Inventories of simulated seagrass sediments had a deficit of 2,400 Bq m$^{-2}$ (60%), 1,250 Bq m$^{-2}$ (30%) and 600 Bq m$^{-2}$ (15%) in erosion scenarios H, I, and J, respectively, while these deficits were of 900 Bq m$^{-2}$ (22%), 700 Bq m$^{-2}$ (19%) and 600 Bq m$^{-2}$ (15%) in mangrove/tidal marsh sediments. Because seagrass ecosystems have lower sedimentation rates, a greater proportion of the excess [210]Pb inventory was comprised in the top 10 cm of the sediment column and thus missing because of erosion. Simulations of past erosion events, which can be identified deeper in the profile, produced breaks in the slope of excess [210]Pb concentrations (Fig. 3c) similar to those of type II, yet showing an increase in the slope (Fig. 2). Simulated erosion scenarios did not result in a large impact in MAR and CAR estimated by the CF:CS model under the conditions of this simulation (Fig. 5). The steeper gradient in excess [210]Pb concentrations produced by past erosion events resulted in a slight decrease in average MAR. Consequently, derived CAR decreased by only 7% and 2% in seagrass and mangrove habitats, respectively. The CRS model cannot be applied to eroded excess [210]Pb profiles unless the missing inventory is known and the total ($I$) and depth-specific ($A_m$) excess [210]Pb inventories can be corrected. Assuming erosion was not a factor, the application of the CRS model to our simulated profiles underestimated MAR and CAR by up to 25% in seagrass and by 10% in mangrove/tidal marsh sediments (Fig 5b and 5d). Therefore, we

caution against the use of the CRS model in profiles that show deviations from the expected inventory, such as those simulated for seagrass sediments here (Fig. 3c). The magnitude of erosion is better estimated by the deficit in inventories of excess $^{210}$Pb, rather than by sedimentation rates. The comparison between sediment records can provide information about the degree of erosion (Fig. 3c). In our simulations, the $C_{org}$ stocks over the last 100 yr were 20% and 5% lower in seagrass and in mangrove/tidal marsh sediments, respectively, compared to the corresponding 'ideal' profile under non-eroded conditions. Part of this is likely related to the fact that the concentration of $C_{org}$ is not changed, which in reality may actually change since fine sediments, where $C_{org}$ is more efficiently adsorbed, are more easily eroded and OM is remineralized when exposed to oxic conditions during resuspension (Burdige, 2007; Lovelock et al., 2017a; Serrano et al., 2016a) (see simulations 3.2.4 and 3.2.5). Consequently, losses of sediment $C_{org}$ could be significantly larger, as shown in some recent studies (Macreadie et al., 2013, 2015; Marbà et al., 2015; Serrano et al., 2016a).

### 3.2.4 Sediment grain size distribution

Coarse sediments are often unsuitable for $^{210}$Pb dating as they may lead to very low excess $^{210}$Pb concentrations. We simulated excess $^{210}$Pb concentration profiles in a coarse sand sediment (scenario K, Fig. 4a). This led to diluted excess $^{210}$Pb concentrations and thus, like erosion processes, produced profiles with lower specific activities of excess $^{210}$Pb. In contrast to erosion simulations, coarse but homogeneous grain size distribution with depth did not have any impact in MAR and CAR estimated by the CF:CS model, since the dilution effect did not cause any irregularity in the slope of the excess $^{210}$Pb concentration profile. However, the CRS model underestimated the sedimentation rate by 15% in both habitats (Fig. 5). The reduction of excess $^{210}$Pb specific activity may cause the limits of detection of excess $^{210}$Pb (0.35 Bq kg$^{-1}$ in our simulations) to be reached at shallower depths than in the ideal profile. In this simulation, the limits of detection were 5 and 7 cm shallower in seagrass and in mangrove/tidal marsh sediments, respectively (Supplementary, Table 5a and 5b). This conduces to the overestimation of the sediment age at bottom layers by the CRS model, and underestimated mean MAR, due to the omission of a higher fraction of the integrated excess $^{210}$Pb activity per unit area from $A_m$ and $I$ at depths greater than those at which the limit of detection was reached (MacKenzie et al., 2011). This effect is known as the "old-date error" of the CRS model and can be corrected as described in Binford (1990) and Appleby (2001). Because we have assumed the same $C_{org}$ content in ideal than in simulated profiles, CAR estimates vary similarly to MAR. However, $C_{org}$ content would likely co-vary with grain size, and we therefore expect lower $C_{org}$ content in coarser sediments (Dahl et al., 2016; Sanders et al., 2012).

Simulations of varying grain size distribution with depth (scenarios L M and N) led to stepwise excess $^{210}$Pb profile forms (Fig. 4b). A sharp increase in excess $^{210}$Pb concentrations in surface layers can be produced by the presence of finer sediments where $^{210}$Pb is preferentially associated (scenario L). As a result, sedimentation rates were 2 to 20% lower than those estimated for the ideal profile in both habitat types using the CF:CS and the CRS models, respectively. (Fig. 5). The CRS model assumes

that excess [210]Pb concentrations are inversely related to the sedimentation rate, and thus higher excess [210]Pb concentrations resulted in lower accumulation rates.

When coarser sediments dominate at the surface layers (scenario M), the simulated profiles obtained were similar to those with mixing and accelerated sediment accumulation in recent years (types II, III and IV). The dilution of the [210]Pb concentrations caused by the deposition of coarse sediments in surface layers was interpreted by the CRS model as an increase in the sedimentation rate, however, this effect was compensated in part by the "old-date error". With coarser sediments at surface layers, the CF:CS model applied piecewise overestimated average MAR and CAR by only 1% in both habitat types, while the CRS model resulted in a 5% overestimation (Fig. 5). If changes in grain size are considered throughout the entire excess [210]Pb profile (scenario N), deviation in accumulation rates increased for the CF:CS model and were up to 10% using both models in both habitat types. Indeed, the deposition of coarse sediments may indicate exceptional increases in sedimentation in the case of storm surge deposits or pulsed sediment deliveries. However, the presence of coarse sediments is often related to a reduction in the deposition of fine particles or to the transport and erosion of these in high energy environments, leading to a variation in the excess [210]Pb flux onto the sediment surface, considered constant through time by the two dating models. Where heterogeneous sediment layers are present, some corrections, such as the normalization of excess [210]Pb concentrations, are required before the application of any of the [210]Pb dating models to obtain more accurate estimates of MAR and CAR (see section 4.4).

### 3.2.5 Organic matter decay

Two different scenarios with low and high sediment organic matter (OM) content (16.5% and 65%, respectively) were modelled in relation to OM decay. In both scenarios simulated MAR and CAR were overestimated relative to those derived from ideal profiles that accounted for the loss of mass with depth due to OM decay. Variation in OM decay (from a starting level of 16.5%) only slightly affected the excess [210]Pb concentration profiles (Fig. 4c) causing a small overestimation of MAR and CAR of between 2 and 5% in both habitats and by both models, under any of the rates of decay considered in this simulation ($0.00005$ d$^{-1}$, $0.0005$ d$^{-1}$ and $0.01 - 0.03$ d$^{-1}$) (Fig. 5a and 5c). Organic matter decay in very rich organic sediments (65% OM) caused increased excess [210]Pb concentrations at surface (scenarios R and T) and subsurface sediments where decay of OM is greater, leading to reversal of excess [210]Pb concentrations (such as in type IV) in simulated scenario S. Derived CAR were 20 - 30% higher as estimated by the CF:CS model and 10 – 20% using the CRS model, in both habitat types (Fig. 5a and 5c). Mass accumulation in vegetated coastal ecosystems is the result of the balance between material accretion (detritus and sediment) from autochthonous and allochthonous sources, decomposition and erosion (e.g. Mateo et al., 1997). Assuming there is no erosion, the estimates of MAR and CAR by means of [210]Pb are the net result of mass accumulation with time, and hence integrate both burial and decomposition of organic matter over a centennial time scale. Therefore, these rates remain underestimated if compared with those at the time of deposition. If compared with the initial ideal MAR of 0.2 and 0.3 g cm$^{-2}$ yr$^{-1}$ in seagrass and mangrove/tidal marsh sediments, respectively, estimated MAR were from 16 to 65% lower if initial OM

content was 16.5 and 65%, respectively. CAR estimates, in contrast, were between 80% and 100% lower than those at the time of deposition in sediments with OM content of 16.5% and 65%, respectively. Because mean CAR rates are based on the $C_{org}$ presently available and not the amount originally deposited, their determination will be dependent on the time scale over which they are calculated.

### 3.2.6 General remarks

Among the various ecosystems considered here, average last 100-yr MAR and CAR derived from the CF:CS and the CRS models were less vulnerable to anomalies in mangrove/tidal marsh compared to seagrass sediments. Higher sedimentation rates lead to deeper excess $^{210}$Pb dating horizons and thus the fraction of $^{210}$Pb profile affected by anomalies was lower in

mangrove/tidal marsh than in seagrass sediments. Anomalies caused by deep mixing or 2- to 3-fold acceleration in sedimentation had larger effects on the CF:CS derived accumulation rates, while alterations caused by heterogeneous grain size composition primarily affected the CRS derived results (Fig. 5). Care must be taken in these cases and with the model choice as deviations in mean MAR were between 20% and 80%. Our simulations showed that the decay of OM results in an overestimation of the accumulation rates, which was most severe in very rich organic sediments regardless of the model used

(> 50% OM). However, this effect could reasonably be ignored in most cases since vegetated coastal ecosystems rarely contain OM concentrations >25% (Table 1), for which the deviation in computed MAR was below 10%. Overall, simulations showed that the variability in MAR and hence CAR due to sedimentary processes and differences in sediment composition was moderately low when appropriate dating models were applied and interpreted. Deviations in the determination of MAR and CAR, generally within 20%, confirmed that the $^{210}$Pb dating technique is secure (Fig. 5). However, failure to account for the

correct process affecting $^{210}$Pb concentration profiles could lead to deviations in mean MAR and CAR exceeding 20% (Fig. 5c, d).

MAR and CAR were most overestimated, from 20 to 95% in simulations with low accumulation rates, when acceleration was interpreted in mixed excess $^{210}$Pb profiles and the CF:CS model was applied piecewise. Deep mixing confounded with an

increase in MAR generated the largest overestimation of mean CAR in both habitat types. In contrast, if mixing was assumed in excess $^{210}$Pb profiles showing a recent increase in MAR, mean accumulation rates were underestimated by up to a 30% using the CF:CS model below the "surface mixed layer". Indeed, the CRS model was less sensitive to anomalies in excess $^{210}$Pb concentration profiles, however, its application requires accurate determination of the excess $^{210}$Pb inventory at each depth ($A_m$) and in the entire record (*I*), which can be problematic, for instance when all samples along a sediment core have

not been analysed or when sediment erosion has occurred at the core location. When the total excess $^{210}$Pb inventory is underestimated, be it through erosion, poor detection limits or insufficient core length, this generates erroneous dates and underestimation of average MAR and CAR. Underestimation of accumulation rates will depend largely on the proportion of the missing fraction of the excess $^{210}$Pb inventory from $A_m$ and *I*. In our simulations, MAR and CAR were underestimated by 10 to 25%. While uncertainties within a 20% might be acceptable for the determination of mean MAR and CAR over a

centennial time scale, they may not allow the determination of a detailed geochronology, historical reconstruction, or to ascertain rates of change and fluxes at specific times. In that event additional tracers or geochemical, ecological and historical data need to be used to validate the $^{210}$Pb-dreived results and reduce uncertainties caused by anomalies in excess $^{210}$Pb concentration profiles in vegetated coastal sediments.

**4 Approaches and Guidelines**

Retrieving reliable CAR depends on the correct determination of MAR and the diagnosis of the intervening sedimentary processes. However, similarities in simulation outcomes and variations associated with anomalies in excess $^{210}$Pb profiles point to the need for additional sources of evidence to discriminate between alternative processes and constrain $^{210}$Pb-derived estimates. $^{137}$Cs or other independent radioactive tracers can be used to corroborate $^{210}$Pb geochronologies. However, in its

absence, geochemical information combined with knowledge on events related to land-use and/or environmental changes (e.g. by means of aerial photographic evidence, Swales et al. 2015) can also be used as a tool to validate $^{210}$Pb geochronologies and interpret excess $^{210}$Pb profiles appropriately. In Figure 6 we have summarized the steps to characterize $^{210}$Pb profiles and the sedimentary processes most likely involved and suggest several techniques to complement the $^{210}$Pb dating method to obtain reliable MAR and CAR.

Prior to analysis, researchers can have control over some factors such as coring, sampling, or sample-handling, that can create artefacts in $^{210}$Pb profiles and therefore contribute to dating error. Guidelines for core sampling for the analysis of $^{210}$Pb and other radionuclides have been described in detail, for example, in Brenner and Kenney (2013) and in the technical report IAEA-TECDOC-1360 (2003). Some knowledge on the expected sedimentation rate is useful to decide how to section a

sediment core for $^{210}$Pb measurements, as well as the length that a core must have to reach the depth of the excess $^{210}$Pb horizon. Low sedimentation rates ($\sim$1-2 mm yr$^{-1}$) and/or coarse sediments may imply that the $^{210}$Pb datable part of sediment cores is limited to the very top centimetres. In such situation, fine sectioning intervals (0.5 - 1 cm) would be required. Longer cores (of about 100 cm) should be collected if high sedimentation rates are expected (several mm yr$^{-1}$) so that the entire excess $^{210}$Pb inventory is captured and the CRS model can be applied. These can be sliced at thicker intervals without compromising the

temporal resolution of the $^{210}$Pb record. If the order of magnitude of sedimentation rates are not known a priori, it is best to choose fine sampling intervals (e.g, at 0.5 cm along the upper 20 cm, at 1 cm from 20 to 50 cm, and at 2 cm below 50 cm) to ensure sufficient resolution.

After collection, a visual description (e.g., colour, sediment texture, presence of roots, organisms or layers) of the sediments and measurement of parameters such as water content, OM and grain size are relatively low-cost actions that provide

information to interpret $^{210}$Pb distribution and the pattern of accumulation. Indeed, the type of sediment (e.g., fine vs. coarse, rich in carbonates, homogeneous or with organic debris embedded) is a factor that should be considered (IAEA-TECDOC-1360, 2003). Coarse particles or coarse-grained carbonates where excess $^{210}$Pb is less preferentially adsorbed (Wan et al., 1993)

may hinder the detection of any excess $^{210}$Pb in vegetated coastal sediments. In such situations, the analysis of $^{210}$Pb in the smaller sediment fraction (i.e. < 63µm or < 125 µm) is recommended to concentrate $^{210}$Pb and reduce the dilution effect caused by coarse fractions. This methodology has been applied in mangrove ecosystems from arid regions where excess $^{210}$Pb flux is low (Almahasheer et al., 2017) and in Florida Bay carbonate-rich seagrass sediments (Holmes et al., 2001). Similarly, large organic material such as roots and leaves should be removed from the sediment samples prior to $^{210}$Pb analyses as these may contribute to the dilution of the excess $^{210}$Pb specific activity.

The analytical methods for $^{210}$Pb measurements can also be chosen depending upon the amount of sample available and its expected specific activity. While indirect determination of $^{210}$Pb by alpha spectrometry of its granddaughter $^{210}$Po requires little amount of sample (150 – 300 mg) and will provide a significant better limit of detection (< 1 Bq kg$^{-1}$), direct determination of $^{210}$Pb by gamma spectrometry can simultaneously provide data for supported $^{210}$Pb ($^{226}$Ra) and relevant radionuclides, such as $^{137}$Cs, $^{228}$Th, $^{7}$Be, $^{40}$K, to validate the $^{210}$Pb geochronologies. For a detailed description of the analytical methods and their advantages and disadvantages see for instance Corbett and Walsh, (2015) and Goldstein and Stirling, (2003).

## 4.1 General validation of $^{210}$Pb models

### 4.1.1 Artificial radionuclides

Independent validation of the chronology is essential to ensure a high level of confidence in the results (Smith, 2001). Varved sediments used to validate chronologies in lakes do not occur in vegetated coastal sedimentary sequences, and thus transient signals such $^{137}$Cs or $^{239+240}$Pu become the most commonly used option to validate $^{210}$Pb chronologies (Lynch et al., 1989; Sanders et al., 2010). $^{137}$Cs and $^{239+240}$Pu were released to the environment through the testing of high-yield thermonuclear weapons in 1950s to early 1960s and can be used as chronometers in sediments either by assuming that the peak in activity corresponds to the fallout peak in 1963 or the depth of its first detection corresponds to the onset of fallout in the early 1950s. In addition, $^{137}$Cs can also display a peak of elevated activity in sediment cores from Europe, corresponding to the emissions caused by the Chernobyl accident in 1986, which can also help to validate $^{210}$Pb chronologies (Callaway et al., 1996).

However, the use of $^{137}$Cs might have some limitations in vegetated coastal sediments. Two-thirds of the $^{137}$Cs activity released due to the tests in the atmosphere decayed after 5 decades, rendering the identification of peaks and its correspondence to the early 50's and 60's depths more difficult to determine. In addition, the absence of $^{137}$Cs signal is reportedly a problem in sediment cores from habitats located in the Southern hemisphere and near the Equator. The low $^{137}$Cs bomb-test fallout and Chernobyl inputs in these regions (Kelley et al., 1999; Ruiz-Fernández and Hillaire-Marcel, 2009), the greater solubility of $^{137}$Cs in seawater and the presence of sands and carbonates, particularly in seagrass sediments (Koch, 2001), are conditions that do not favour the adsorption of $^{137}$Cs (He and Walling, 1996a), and may lead to its mobility (Davis et al., 1984), due to its low partition coefficient in seawater ($K_d = 10^2$ to $10^3$, Bruland, 1983). This effect could be intensified in the intertidal zone, which is not permanently submerged, due to periodic changes in the water table. High contents of organic matter can also

affect the distribution of $^{137}$Cs in sediments as it is preferentially accumulated in leaf litter and may be absorbed by living roots (Olid et al., 2008; Staunton et al., 2002). In addition, decomposition of the organic phase in organic-rich sediments may cause mobility of this radionuclide (Davis et al., 1984). These factors together may compromise the use of $^{137}$Cs to validate the $^{210}$Pb geochronologies in vegetated coastal ecosystems. In contrast, Pu isotopes ($^{239}$Pu half-life = 24,100 yr and $^{240}$Pu half-life = 6,500 yr), although they are also dependent on the distribution of bomb-test fallout, would appear to offer several advantages over $^{137}$Cs in these environments, since $^{239+240}$Pu is relatively immobile under both freshwater and saltwater conditions (Crusius and Anderson, 1995). For instance, Sanders et al. (2016) determined sedimentation rates and $^{239+240}$Pu penetration depths to study nutrient and CAR in intertidal mangrove mudflats of Moreton Bay, Australia. Nevertheless, and because of the limitations to validate older $^{210}$Pb dates near the base of the core, and the low inventories of bomb-test fallout in coarse sediments and Southern Hemisphere latitudes, alternative tracers might need to be used.

### 4.1.2 Geochemical information of sediments

Besides the irregular shape of excess $^{210}$Pb profiles, the absence of a secondary radioactive tracer to validate $^{210}$Pb results can make interpretation even more complicated. However, geochemical information in the sediment column can provide the potential for an additional temporal frame and can also help to explain sedimentary processes that could be misinterpreted (e.g., mixing, increasing sedimentation rates, higher primary productivity or reduction of sediment supply). Analyses of additional proxies (pollen, diatom, nutrient concentrations, stable isotopes or trace metal records; López-Merino et al., 2017) that are based on well-described historical events at the study sites (e.g. pollution, crops and land-clearance) could be used in the absence of secondary radioactive tracers to corroborate $^{210}$Pb derived dates and accumulation rates. For instance, stable Pb isotopes or total Pb concentrations in sediments are related to the history of use of leaded gasoline in the area and can be used to identify age marks corresponding to peaks in its use or changes in lead sourcing. An example can be found in seagrass sediment cores from Florida Bay, USA (Holmes et al., 2001) or in Gehrels et al. (2005) that combines marsh elevation reconstructions with a precise chronology derived from pollen analysis, stable isotopes ($^{206}$Pb, $^{207}$Pb), $^{210}$Pb and artificial radionuclides ($^{137}$Cs, $^{241}$Am). Additionally, profiles of trace and heavy metals and of carbon $\delta^{13}$C and nitrogen $\delta^{15}$N isotopic composition of OM provide information about environmental changes for which historical information may be well known, i.e., human settlement, onset of tourism industry, temporal evolution of cropland areas or histories of variation in plant communities (Garcia-Orellana et al., 2011; Mazarrasa et al., 2017; Ruiz-Fernández and Hillaire-Marcel, 2009; Serrano et al., 2016c).

### 4.2 Mixing or Rapid sedimentation

The methods described above for the general validation of $^{210}$Pb models can also serve to discriminate between mixing or increasing MAR in recent years. $^{137}$Cs and $^{239+240}$Pu can also be used as tracers of bioturbation (Crusius et al., 2004) or acceleration of sedimentation during the past 50 years (Appleby, 1998; Cearreta et al., 2002; Lynch et al., 1989; Sharma et al.,

1987). For instance, demonstration of acceleration versus fast mixing could be supported when it is possible to find the distinct [137]Cs or [239+240]Pu peaks in the same zone where excess [210]Pb activities are constant (Appleby, 2001). Changes along the profiles of geochemical elements consistent with shifts in excess [210]Pb concentrations often can be associated with changes in sedimentation or erosion processes. For instance, instantaneous depositional event layers can be identified in the sedimentary

record as isolated minima of excess [210]Pb concentrations (Jaeger and Nittrouer, 2006; Smoak et al., 2013), but also as variations in grain size composition, OM, water content or dry bulk density (Smoak et al., 2013; Walsh and Nittrouer, 2004) (Box 1). Changes in sediment mineralogy can be discerned trough X-ray radiographs, X-ray fluorescence and CAT-scans (described below), but also through other radionuclides, like [226]Ra and [40]K, the profiles of which can be measured together with those of [210]Pb through gamma spectrometry. In particular [40]K is also part of the mineral matrix and is often used as a surrogate for the

lithogenic sediment fraction (Garcia-Orellana et al., 2006; Peterson, 2009; Xu et al., 2015).

### 4.2.1 Geophysical analyses

Prior to core sectioning and subsampling, non-destructive geophysical analyses such as X-ray radiographs, X-ray fluorescence (XRF), CAT-scans (Computerized Axial Tomography) or magnetic susceptibility can be conducted to identify changes in the

composition of sediments with depth, changes in MAR or provide evidence of mixing prior to the analysis of excess [210]Pb. For instance, using X-ray radiographs many features and physical sedimentary structures may be visible (Sun et al., 2017) and if preserved, could support the interpretation of a rapid increase in sedimentation (Walsh and Nittrouer, 2004). Pulsed sediment deliveries or erosion could be identified by discontinuous physical stratification, and sediment mixing by the presence of active burrows or the absence of sedimentary stratification (Chanton et al., 1983).

**4.2.2 Short-lived radionuclides ([234]Th, [228]Th, [7]Be)**

Radionuclides such as [234]Th, [7]Be and [228]Th with properties such as particle-reactivity and relatively short half-lives (24.1 days, 53.3 days and 1.9 years, respectively) are suitable to quantify sedimentation processes at scales from several months ([234]Th and [7]Be) to a decade ([228]Th), and are sensitive indicators of mixing in the zone of constant, scattered or reversed excess [210]Pb concentrations (Types II, III, IV, Fig, 2) (Cochran and Masqué, 2005; Sommerfield and Nittrouer, 1999). In addition,

demonstrating the presence of excess of a short-lived radionuclide can give confidence that there is little material missing from the top of the sediment record and no recent erosion, which is essential for the application of the CRS model. An example is documented by Smoak and Patchineelam (1999) for a [210]Pb concentration profile affected by bioturbation in a mangrove ecosystem in Brazil (Box 2).

Recent increases in sedimentation can be estimated from the slope of the best-fit lines of the plots of [7]Be, excess [234]Th and [228]Th concentrations against cumulative mass, as Alongi et al. (2005) showed in a mangrove ecosystem in Jiulongjiang Estuary, China (Box 3). However, the use of short-lived radionuclides to derive recent increases in sedimentation is restricted to habitats

with high accumulation rates (i.e. > 4 mm yr$^{-1}$, being the last 10 yr comprised in the upper centimetres) due to their relatively short half-lives. Indeed, excess $^{228}$Th ($T_{1/2}$ = 1.9 yr), might be the only suitable tracer to be used in mangrove/tidal marsh ecosystems where sedimentation rates are on average 5 - 7 mm yr$^{-1}$. A constraint on the use of excess $^{228}$Th is that sediments must contain a lithogenic/detrital fraction, but this is often the case in vegetated coastal sediments. The other short-lived radionuclides might only be applied to assess the magnitude of mixing or recent erosion in vegetated coastal sediments.

Mixing, either due to bioturbation or hydrodynamic energy, is the most common process affecting vegetated coastal sediment records. Although the presence of vegetation and anoxic sediments tends to reduce the depth of sediment mixing (Duarte et al., 2013), the mixed layer can extend to depths of 10-15 cm in marine sediments (Boudreau, 1994). If surface mixing occurs, valid estimates of sedimentation rates (within 5% variability as shown in section 3.2.1) can still be obtained using the dating models described above, however this can only be possible in sediments where excess $^{210}$Pb is buried below the mixed layer prior to decay, i.e., the residence time of sediments in the mixed layer must be shorter than the effective dating time scale (~100 yr) (Crusius et al., 2004). In the example from Smoak and Patchineelam (1999) (Box 2), where mixing extends to a depth of 11 cm, the sedimentation rate had to be higher than 1.1 mm yr$^{-1}$ in order for $^{210}$Pb to be a useful chronometer (residence time in the mixed layer = 110 mm / 1.1 mm yr$^{-1}$ = 100 yr, which is within the effective dating time scale of $^{210}$Pb).

### 4.2.3 Maximum penetration depth of excess $^{210}$Pb

A chronology cannot be estimated if mixing affects the whole or the vast majority of the sediment record, as in the simulation of deep mixing in seagrass sediments in this study. However, information such as the total historical inventory of elements, like nutrients accumulated at a site, and the maximum conservative sedimentation rate can still be estimated. The penetration-depth method (Goodbred and Kuehl, 1998; Jaeger et al., 2009) uses the maximum penetration depth of excess $^{210}$Pb (depth of disappearance) as a marker horizon for sediments that are ~100 yr old. Low surface excess $^{210}$Pb concentrations can greatly restrict the age of the $^{210}$Pb dating horizon, therefore this is an issue that should be considered when establishing the age of the excess $^{210}$Pb horizon. For surficial concentrations less than ~100 Bq kg$^{-1}$ this could be as little as 3–4 $^{210}$Pb half-lives, i.e., 65–90 years. By locating the dating horizon, independently of subsequent alteration of sedimentary processes and of assumptions of the CF:CS or CRS models, an upper estimate of the average sedimentation rate can be derived. It is important to highlight that by using this method, the rates of change or fluxes cannot be estimated and these types of excess $^{210}$Pb profiles may be of little use in establishing chrono-stratigraphies since are unlikely to have good records of other environmental parameters.

### 4.3 Erosion: Excess $^{210}$Pb inventories (*I*)

Assessing the extent of erosion requires the comparison of the excess $^{210}$Pb inventories between reference i.e., undisturbed locations (*I$_{ref}$*) and eroded sites (*I*). Because excess $^{210}$Pb is particle reactive, once deposited in sediments, its subsequent lateral redistribution is primarily controlled by resuspension and transport processes, and thus a deficit in excess $^{210}$Pb inventories

relative to undisturbed sediments may indicate loss or mobilisation of sediment particles. This approach has been used in terrestrial soils (Martz and Jong, 1991; Walling et al., 2003) and more recently to assess erosion of seagrass sediments (Greiner et al., 2013; Marbà et al., 2015; Serrano et al. 2016b) (Box 4). Because the excess [210]Pb inventories at a reference undisturbed location may be spatially variable, we recommend the use of a reference inventory value based on several cores (i.e., mean ± 2SE). The consistency of the resulting reference inventory value can then be assessed by comparing it with that expected from the local atmospheric flux of excess [210]Pb $\Phi$ ($\Phi = I_{ref} \cdot \lambda$), which might have been reported by others (for global and regional ranges see Preiss et al. 1996) or with the excess [210]Pb inventory measured in a terrestrial undisturbed soil characterized by minimal slope.

## 4.4 Heterogeneous sediment composition

### 4.4.1 Normalization of excess [210]Pb concentrations

Dating models assume rapid and non-discriminatory removal of radionuclides from the water column regardless of major changes in grain size or OM content along a sediment record. Radionuclide adsorption onto sediments is strongly governed by the binding capacity of the settling particles (Cremers et al., 1988; Loring, 1991), thus its scavenging is increased by fine-grained texture (He and Walling, 1996) and OM particles (Yeager and Santschi, 2003). Variations in the influx of these particles into vegetated coastal sediments may proportionally affect the influx of particle bound excess [210]Pb (as long as it is still available), thus violating the assumption of constant flux of the CRS model and leading to subsections and irregularities of excess [210]Pb profiles. Constant or reversed patterns in excess [210]Pb concentrations, which could be easily mistaken for reworked deposition, could be caused, for instance, by vertical fluctuations of grain size due to seasonal variations of sediment discharge or reoccurring tidal currents. Sediment studies often attempt to minimize these effects by normalizing radionuclide concentrations to granulometric or geochemical parameters that reduce the influence of preferential adsorption by fine sediments and OM (Álvarez-Iglesias et al., 2007; Loring, 1991; Wan et al., 2005), allowing to obtain excess [210]Pb concentration profiles showing an exponential decreasing trend with depth (Kirchner and Ehlers, 1998; Sun et al., 2017). Radiometric applications in coastal sediments have traditionally opted for grain size normalizers such as the < 4 μm, < 63 μm fraction or Al content (Álvarez-Iglesias et al., 2007; Sanders et al., 2010; Sun et al., 2017; Walsh and Nittrouer, 2004), while in dynamic, sandy-rich coastal systems where the mud fraction is small, normalization by OM content has been shown to be also effective (Van Eaton et al., 2010). Equation 6 can be used to normalize excess [210]Pb concentrations ([210]Pb$_{xs}$-NORM in Bq kg$^{-1}$) by grain size fractions, OM content or other geochemical parameters that control the variation of the input of excess [210]Pb and play an important role in the distribution of excess [210]Pb concentrations.

$$^{210}Pb_{xs\text{-}NORM} = {}^{210}Pb_{xs\text{-}MEAS}(NP_{AVG}/NP_m) \hspace{3cm} (Eq.\ 6)$$

where $^{210}Pb_{xs\text{-}MEAS}$ is the measured specific activity of the bulk sample at depth $m$, and $(NP_{AVG}/NP_m)$ is the ratio between the core average normalizing parameter to its content at depth $m$. For instance, multiplication by this ratio corrects measured [210]Pb

activities for variations in OM with respect to an average core value. In addition, excess [210]Pb can be analysed in clay, silt and sand fractions to determine the [210]Pb partitioning among the three size fractions to then correct bulk sediment excess [210]Pb concentrations for dilution by sands or silts, if clay is the main excess [210]Pb carrying phase (Chanton et al., 1983).

### 4.4.2 [226]Ra concentration profiles

Excess [210]Pb concentrations are determined by subtracting supported [210]Pb, assuming it is in equilibrium with [226]Ra, to total [210]Pb concentrations. This is straightforward when gamma spectrometry is employed since the total [210]Pb and supported [210]Pb (i.e., [226]Ra) can be quantified simultaneously. On occasions, particularly when [210]Pb is determined by alpha spectrometry, [226]Ra is not measured, and supported [210]Pb is most often determined from the region of constant and low [210]Pb concentrations at depth, or alternatively, from a number of determinations of [226]Ra via gamma spectrometry or liquid scintillation counting (LSC) along the core. This method assumes that [226]Ra or supported [210]Pb are constant throughout the sediment core (Binford, 1990). However, this might not be always the case, especially in heterogeneous profiles consisting of a variety of sediment types (Aalto and Nittrouer, 2012; Armentano and Woodwell, 1975; Boyd and Sommerfield, 2016) or in records containing episodes of rapid sedimentation (Chanton et al., 1983). In addition, equilibrium of [210]Pb supported with [226]Ra might be compromised in surface sediments, where [222]Rn is deficient (Appleby, 2001). Although variations in [226]Ra concentrations with depth are small in most cases, accurate determination of [226]Ra might be crucial in sediments with low total [210]Pb concentrations (e.g., due to the presence of coarse sediments), where slight variations in the supported [210]Pb may result in significant errors in the estimation of excess [210]Pb concentrations (Diemer et al., 2011). To avoid deviations in excess [210]Pb concentration profiles associated with variations in supported [210]Pb, it is recommended to measure [226]Ra concentration profiles or to, at least, use depth-specific [226]Ra values at several depths along a sediment profile to estimate excess [210]Pb.

### 5 Conclusions

[210]Pb dating techniques provides crucial information for the study of carbon sequestration in vegetated coastal ecosystems and can also provide accurate geochronologies for the reconstruction of environmental processes based on the study of the sedimentary sequences found in the sediments of these habitats. However, [210]Pb reconstruction studies may be difficult to conduct in mangrove, tidal marsh and seagrass ecosystems, where unaltered sedimentary records are rare.

Shallow vegetated coastal sediments are often affected by a number of processes such as mixing and bioturbation, accelerated sedimentation or erosion and might be composed of heterogeneous sediments. These factors may lead to anomalies in the excess [210]Pb concentration profiles, and thus produce erroneous geochronologies and deviations in mean last-century MAR and CAR. Simulated irregular [210]Pb profiles in this study show that the deviations, relative to ideal undisturbed [210]Pb profiles, in MAR and CAR are within 20% if a correct diagnosis of the intervening sedimentary processes is made. Otherwise,

deviations may range between 20% and 100%, with higher errors associated with the application of CF:CS model. Additional tracers or geochemical, ecological or historical data can be used to identify the process causing anomalies in excess $^{210}$Pb profiles and reduce uncertainties in derived accumulation rates. Model choice is another important factor that should be considered to reduce deviations in CAR. Using the procedures in section 4, researchers have been able to obtain credible

chronologies in vegetated coastal sediments and reliable mean CAR. This, however, might be particularly challenging in seagrass sediments because of their relatively low sedimentation rates and high sand content where $^{210}$Pb is less adsorbed because of the low specific surface area of sands. Special caution should be applied in those sites where sediments might be altered by multiple processes (leading to profile types V or VI shown in this study) and where other chronological tools or time markers are not available (e.g., $^{137}$Cs). Sites that have slow accumulation rates and/or intense mixing may unlikely be

datable and derived CAR estimates may be largely overestimated. Mistakes would include assigning discrete ages in mixed sediments or extrapolating an age-depth model for a core that should be considered undatable to depths down the core or to nearby sites. While attention should be paid to the limitations of $^{210}$Pb-derived results in vegetated coastal ecosystems, the guidelines provided here should help to understand the limitations that arise from anomalous $^{210}$Pb profiles retrieved from vegetated coastal sediments and to develop a strategy to strengthen the evaluation of MAR and CAR.

## Appendix A: Simulation methods

### Mixing

To simulate surface mixing (scenarios A and B), we estimated the accumulated excess $^{210}$Pb activity per unit area over the top 5 cm of the ideal excess $^{210}$Pb profile ($I_{5cm}$: 2126 Bq m$^{-2}$ in seagrass and 723 Bq m$^{-2}$ in mangrove/tidal marsh sediments) (Supplementary, Tables 1a and 1b). We split this inventory within the 5 upper centimetres using a random function, the outputs

of which fell within the standard deviation ($\pm SD$) of the mean of the excess $^{210}$Pb activities in the upper 5 cm ($\pm 107$ Bq m$^{-2}$ in seagrass; and $\pm 9$ Bq m$^{-2}$ in mangrove/tidal marsh sediments). To simulate deep mixing (scenario C), we followed the same methodology but we split randomly the excess $^{210}$Pb inventory within the upper 15 cm, which is a depth reported as deep mixing in seagrass (Serrano et al., 2016a), mangroves and tidal marshes (Nittrouer et al., 1979; Smoak and Patchineelam, 1999) and is characteristic for marine sediments globally (Boudreau, 1994). We ran the simulation several times until we

obtained three scenarios (A, B, C) of mixing encompassing a range of surface mixed layers (SML) (Supplementary, Tables 2a and 2b). Mixing A ($k_m$: $\infty$ g$^2$ cm$^{-4}$ yr$^{-1}$) consisted of constant excess $^{210}$Pb concentrations with depth in surface layers; mixing B ($k_m$: 20 - 23 g$^2$ cm$^{-4}$ yr$^{-1}$) was characterised by a decrease in the slope of excess $^{210}$Pb concentrations in top layers; and mixing C represented deep mixing from the sediment surface down to 15 cm ($k_m$: 6 - 25 g$^2$ cm$^{-4}$ yr$^{-1}$). Excess $^{210}$Pb concentrations per

unit area ($A$) were converted to excess $^{210}$Pb concentrations ($C$) in Bq kg$^{-1}$, which we averaged every two layers to represent smooth transitions. Sedimentation and derived CAR were estimated from the modelled profiles using the CF:CS and the CRS models. The CF:CS model was applied below the depth of the visually apparent SML (3 cm) in scenarios A and B to avoid

overestimation of MAR. The CF:CS model was applied to the entire profile in deep mixing scenario C in seagrass sediments and below the apparent mixed layer (13 cm) in mangrove sediments. Deep mixing affected 10% of the entire excess [210]Pb profile of mangrove/tidal marsh sediments and 45% of seagrass sediments. To account for the deviations in mean MAR and CAR associated with a process mismatch (i.e., as if considering that the actual mixing was caused by an increase in MAR), we applied the CF:CS model piecewise (scenarios B and C) and to the entire profile (scenario A). In the case of the CRS model, ages were determined at each layer and average centennial MAR was estimated dividing the mass of sediment accumulated (g cm$^{-2}$) down to 100 yr-depth by its age (i.e., 100 yr) in all cases.

**Increasing sedimentation**

We simulated an enhancement of the MAR that could result, for instance, from increased sediment run-off due to coastal development, by increasing the basal MAR (0.2 g cm$^{-2}$ yr$^{-1}$ and 0.3 g cm$^{-2}$ yr$^{-1}$ in seagrass and mangrove/tidal marsh, respectively) by different magnitudes (20%, 50%, 100% and 200%). Increases in MAR were simulated over the top 6 cm and 23 cm of the idealized excess [210]Pb concentration profiles, which represent the last 30 years of accumulation in seagrass and mangrove/tidal marsh sediments, respectively. Last century mass accumulation rates expected for ideal profiles were estimated by dividing the accumulated mass down to a 100yr-depth (derived from gradual increases in MAR) by its age (Supplementary, Tables 3a and 3b). Excess [210]Pb concentrations *($C_m$)* as a result of increased MAR were estimated through equation A1 for each layer. Simulations of increasing MAR generated four profiles per habitat type (scenarios D, E, F and G) (Fig. 3b). Average MAR and CAR were estimated from the modelled profiles using the CF:CS and CRS models. The CF:CS model was applied piecewise in scenarios D, E and F, and below the layer of constant excess [210]Pb concentrations in scenario G.

$$C_m = \frac{\lambda \cdot I_m}{MAR \cdot 10} \tag{A1}$$

where $\lambda$ is the decay constant of [210]Pb (0.0311 yr$^{-1}$) and $I_m$ is the excess [210]Pb inventory accumulated at layer m. 10 allows unit conversion to Bq kg$^{-1}$.

We also estimated mean MAR and CAR assuming that the process causing scenarios D, E, F and G was mixing. For this, we applied the CF:CS model below the surface mixed layer (6 and 23 cm in seagrass and mangrove/tidal marsh sediments, respectively). The shift in the slope of excess [210]Pb concentrations in scenario D in seagrass sediments was minimal, hence we applied the CF:CS model to the entire excess [210]Pb profile, as this would likely be the method applied by most researchers in a real case. The CRS model was run similarly if mixing or changes in accumulation rates are expected, ages were determined at each layer and average centennial MAR was estimated dividing the mass of sediment accumulated (g cm$^{-2}$) down to 100 yr-depth by its age (i.e., 100 yr). If mixing is expected, ages within the mixed layer cannot be reported.

**Erosion**

Erosion in vegetated coastal sediments can occur due to high-energy events (Short et al., 1996), vegetation loss and subsequent destabilization of sediments (Marbà et al., 2015) or mechanical disturbances (e.g. Serrano et al., 2016c). We ran three simulations to represent recent (H) and past erosion events (I and J) (Fig. 3c). We started with an ideal excess [210]Pb profile with a total initial excess [210]Pb inventory of 3,900 Bq m[-2]. To simulate erosion, we removed the excess [210]Pb inventory accumulated in the top 0 - 5 cm (H), middle 5 – 10 cm (I) and 10 – 15 cm sections (J) in sediments from both habitat types (mangrove/tidal marsh and seagrass). Resulting excess [210]Pb activity per unit area (Bq m[-2]) were converted to excess [210]Pb concentrations (Bq kg[-1]) by dividing by the corresponding mass depth (g cm[-2]) at each section after correcting the latter for the loss of sediment layers (Supplementary, Tables 4a and 4b). [210]Pb concentrations were averaged every two layers to simulate smooth transitions rather than a sharp discontinuity after and erosion event. We estimated the resulting average MAR and CAR using the CF:CS model (applied piecewise in erosion scenarios I and J). The CRS model should not be applied in simulated erosion scenarios since the overall core inventories *(I)* are incomplete. However, we ran the CRS model to test the errors associated with its application in eroded sediments assuming that erosion is not a factor.

**Changes in sediment grain size**

We simulated various excess [210]Pb concentration profiles with changes in sediment grain size distribution using the approach described by He & Walling (1996), where the specific surface area of particles exerts a primary control on the excess [210]Pb concentrations adsorbed:

$$C\left(S_{sp}\right) = \mu \cdot S_{sp}^{0.67} \qquad \text{(Eq. A2)}$$

where $C$ is excess [210]Pb concentration (mBq g[-1]), $S_{sp}$ is the specific surface area of the sediment particles (m[2] g[-1]), and $\mu$ is a constant scaling factor depending upon the initial excess [210]Pb activity per unit area (mBq m[-2]). The excess [210]Pb concentration in bulk sediments can also be represented by equation A2 replacing $S_{sp}$ by the mean specific surface area $S_{mean}$ (m[2] g[-1]) of the bulk sample. In this work, we estimated $\mu$ at each layer of an ideal excess [210]Pb profile in seagrass and mangrove/tidal marsh sediments if ideally $S_{sp}$ throughout the core is 0.07 m[2] g[-1], corresponding to a mean particle size of 63 μm. The surface area can be estimated as (Jury and Horton, 2004):

$$S_{sp} = \frac{3}{\rho \cdot r} \qquad \text{(Eq. A3)}$$

where $\rho$ is the density of the sediment particles and $r$ is the mean radius of sediment particles, which are considered spherical. We estimated the weighted mean specific surface area of a very coarse sediment composed of 70% coarse sand (500 – 1000 μm) and 30% medium sand (250 – 500 μm) ($S_{mean}$ = 0.0103 m[2] g[-1]), through equation A3 (size scale: Wentworth, 1922). Bulk density ($\rho$) of sediment fractions were considered: 1.6 g cm[-3] for medium sand and 1.8 g cm[-3] for coarse sand. Then, we simulated excess [210]Pb concentration profiles as a function of the specific surface area applying equation A2 to an ideal excess [210]Pb concentration profile (scenario K) (Supplementary, Tables 5a and 5b). Second, we simulated a shift to sandy and clayey

sediments in surface layers, as could result after the restoration or loss of vegetated coastal ecosystems. The percentages of sands and clay along the core were changed using a random function (from $60 \pm 20\%$ in surface to $15 \pm 5\%$ in bottom layers; scenarios L and M) (Supplementary, Tables 6a and 6b). The shift was simulated at the same age depth (30 yr before collection) in all scenarios and habitat types. Finally, we simulated a heterogeneous grain size distribution along the entire sediment profile

intercalating sand and clay layers randomly with depth (scenario N) (Supplementary, Tables 6a and 6b). The mass depth term was corrected in each case for changes in grain size, which lead to variations in DBD with depth. Bulk density ($\rho$) of sediment fractions was considered: $0.4$ g cm$^{-3}$ for clays and $1.6$ g cm$^{-3}$ for medium sands. In addition, the value of $\mu$ was readjusted at each sediment depth of the ideal profile to represent non-monotonic variations in cumulative dry mass. Excess $^{210}$Pb concentration profiles were estimated as a function of the specific surface area that was estimated at each layer according to

the various proportions of clay and sand. The average MAR was estimated using the CF:CS and CRS models. The CF:CS model was applied piecewise in simulated scenarios L and M.

## Organic matter decay

Excess $^{210}$Pb in vegetated coastal sediments is deposited in association with mineral particles but also with organic particulates (Krishnaswamy et al., 1971; Yeager and Santschi, 2003). Once buried, sediment organic matter (OM) content usually decays

with sediment depth and aging due to remineralization of labile fractions, leading to an enrichment of excess $^{210}$Pb concentrations. We simulated the resultant excess $^{210}$Pb concentration profiles derived from this process in two sediments with different OM contents (16.5% and 65%). The first value (16.5% OM) is within the usual range of tidal marsh, mangrove and in the high range for seagrass sediments (Fourqurean et al., 2012) (Table 1). The second value (65% OM) represents an extreme scenario based in existing studies in seagrass and mangrove ecosystems (Callaway et al., 1997; Serrano et al., 2012). The

simulations were run under three OM decay constants assuming: (1) the whole pool of OM is refractory under anoxic conditions, decaying at a rate of $0.00005$ d$^{-1}$ in seagrass and in mangrove/tidal marsh sediments (Lovelock et al., 2017b); (2) 50% of the refractory pool is exposed to oxic conditions, decaying at a rate of $0.0005$ d$^{-1}$ in mangrove/tidal marsh sediments; and (3) 50% of the OM pool is labile, decaying fast, although exposed to anoxic conditions, at $0.01$ d$^{-1}$ and $0.03$ d$^{-1}$ in seagrass and mangrove/tidal marsh sediments, respectively (Lovelock et al., 2017b).

The $^{210}$Pb enrichment factor ($\eta$) can be determined for a given time after deposition as:

$$\eta(t) = \frac{\chi_s + \chi_{org} \cdot e^{-k_{org} \cdot t}}{\chi_s + \chi_{org}} \quad\quad\quad\quad \text{(Eq. A4)}$$

where $\chi_s$ is the mineral fraction of sediments, $\chi_{org}$ is the organic fraction of sediments at time 0, $k_{org}$ is the decay constant of the OM in sediments and $t$ is time and can be estimated as $^m/_{MAR}$. As time ($t$) increases the exponential term tends to zero, hence the OM stored in the sediment reaches a constant value, where it is no longer decomposed. We assume that the

remineralized OM leaves the sediment as $CO_2$, but in fact a fraction ($f$) would transform to mineral matter as $\chi_s(t) = \chi_{s(0)} + f \cdot \chi_{org(0)} \cdot \left(1 - e^{-k_{org} \cdot t}\right)$. In our simulations $f = 0$ was assumed.

Then, the excess $^{210}$Pb concentration of a sample of age $t$ with initial concentration $C_0$ is:

$$C_t = \frac{C_0 \cdot e^{-\lambda t}}{\eta(t)} \qquad \text{(Eq. A5)}$$

and the total mass accumulated with depth ($M$) above a layer of age $t$ is:

$$M = MAR \cdot \chi_S \cdot t + MAR \cdot \chi_{org} \cdot e^{-k_{org} \cdot t} \cdot t \qquad \text{(Eq. A6)}$$

MAR was estimated using the CF:CS and CRS models. The CF:CS model was applied below the excess $^{210}$Pb reversed
concentrations in scenario S. CAR was estimated through eq. 5. Organic matter (%OM) in mangrove/tidal marsh sediments was transformed to %C$_{org}$ using equation A7 (Kauffman and Donato, 2012). In seagrass sediments we applied the relationship reported by Fourqurean et al. (2012) (Eq. A8) (Supplementary, Table 7a and 7b).

$$\%C_{org} = 0.415 \; \%OM + 2.89 \qquad \text{(Eq. A7)}$$

$$\%C_{org} = 0.43 \; \%OM - 0.33 \qquad \text{(Eq. A8)}$$

For this simulation new MAR and CAR were estimated derived from ideal $^{210}$Pb profiles to represent changes in organic matter content due to decay and associated losses of sediment mass with depth. This resulted in lower ideal MAR in seagrass and mangrove/tidal marsh sediments (seagrass: 0.17 g cm$^{-2}$ yr$^{-1}$ and 0.07 g cm$^{-2}$ yr$^{-1}$ ; mangrove/tidal marsh: 0.25 g cm$^{-2}$ yr$^{-1}$ and 0.10 g cm$^{-2}$ yr$^{-1}$ in OM decay simulations starting at 16.5% and 65% OM, respectively) (Table 3) (Supplementary, Table 7).

**Competing Interest**

The authors declare that they have no conflict of interest.

**Author Contribution**

All authors contributed to the design and data acquisition and/or interpretation. In addition, A. Arias-Ortiz analysed the data and wrote the draft of the manuscript. All authors provided critical review of the manuscript and approved its final version.

**Acknowledgements**

This work was funded by the CSIRO Flagship Marine & Coastal Carbon Biogeochemical Cluster (Coastal Carbon Cluster), the Spanish Ministry of Economy and Competitiveness (Projects EstresX CTM2012- 32603, MedShift CGL2015-71809-P), the Generalitat de Catalunya (MERS 2017 SGR – 1588), the Australian Research Council LIEF Project (LE170100219), the Edith Cowan University Faculty Research Grant Scheme and King Abdullah University of Science and Technology (KAUST)

through baseline funding to CMD. This work contributes to the ICTA 'Unit of Excellence' (MinECo, MDM2015-0552). AAO was funded by a PhD fellowship from Obra Social "laCaixa". OS was supported by an ARC DECRA (DE170101524). IM was funded by a post-doctoral grant (Juan de la Cierva-Formación) from the Spanish Ministry of Economy, Industry and Competitiveness.

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

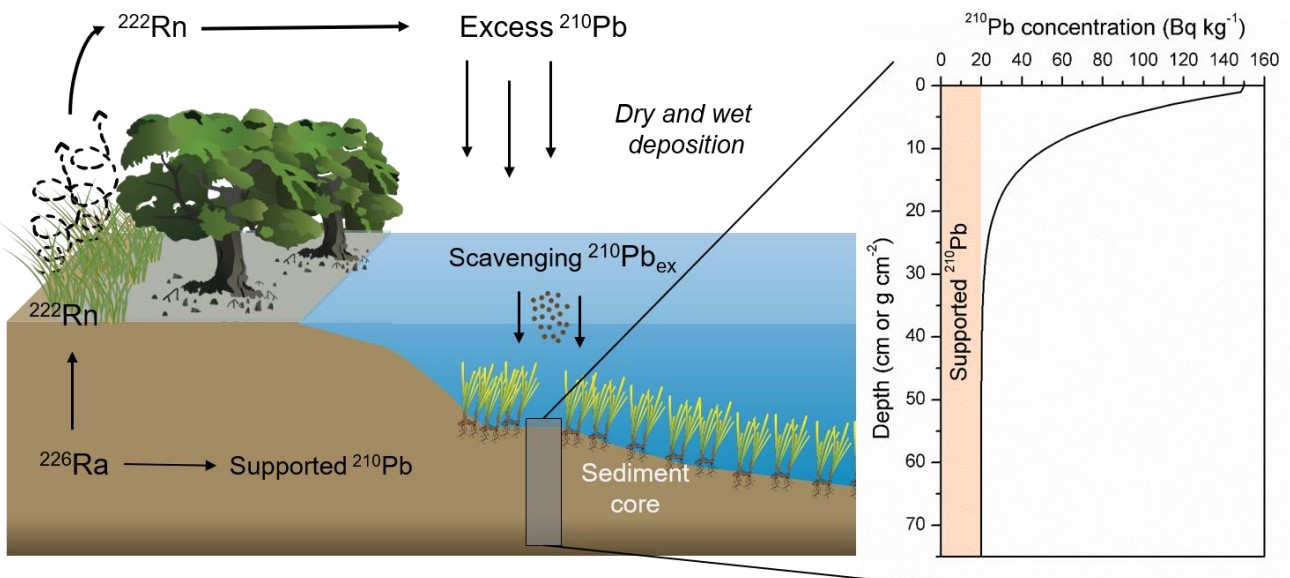

**Figure 1.** $^{210}$Pb cycle and idealized $^{210}$Pb concentration profile in sediments. Images of vegetated coastal habitats: Tracey Saxby, Integration and Application Network, University of Maryland Center for Environmental Science (http://ian.umces.edu/imagelibrary/).

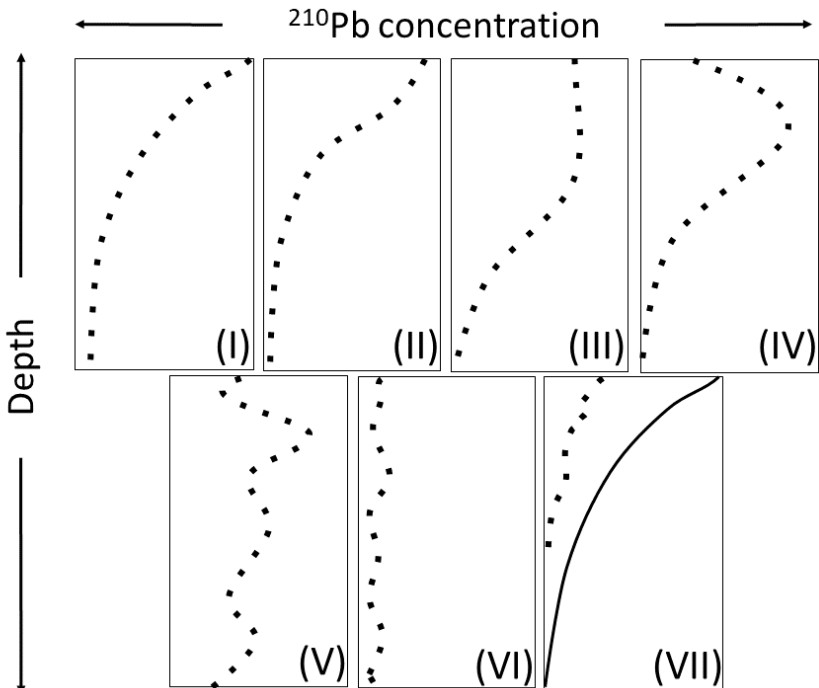

**Figure 2.** Sketch of seven sedimentary types of excess [210]Pb concentration profiles in sediments from vegetated coastal habitats identified from the literature (see references included). Characteristics of each profile type are explained in the text and summarized in Figure 6. The continuous line in Type VII represents the excess [210]Pb concentration profile at a reference undisturbed site. Type II (Cearreta et al., 2002; Gardner et al., 1987; Haslett et al., 2003; Swales and Bentley, 2015; Mazarrasa et al., 2017); Type III (Church et al., 1981; Jankowska et al., 2016; Sanders et al., 2010b, 2010a; Serrano et al., 2016a; Sharma et al., 1987; Smoak and Patchineelam, 1999; Walsh and Nittrouer, 2004); Type IV (Chen and Twilley, 1999; Greiner et al., 2013; Mudd et al., 2009; Sanders et al., 2010b; Serrano et al., 2016c; Smoak et al., 2013; Yeager et al., 2012); Type V (Alongi et al., 2001, 2005; Chanton et al., 1983; Kirchner and Ehlers, 1998; Serrano et al., 2016a; Smoak and Patchineelam, 1999); Type VI (Greiner et al., 2013; Serrano et al., 2016c; 2016d); Type VII (Marbà et al., 2015; Ravens et al., 2009).

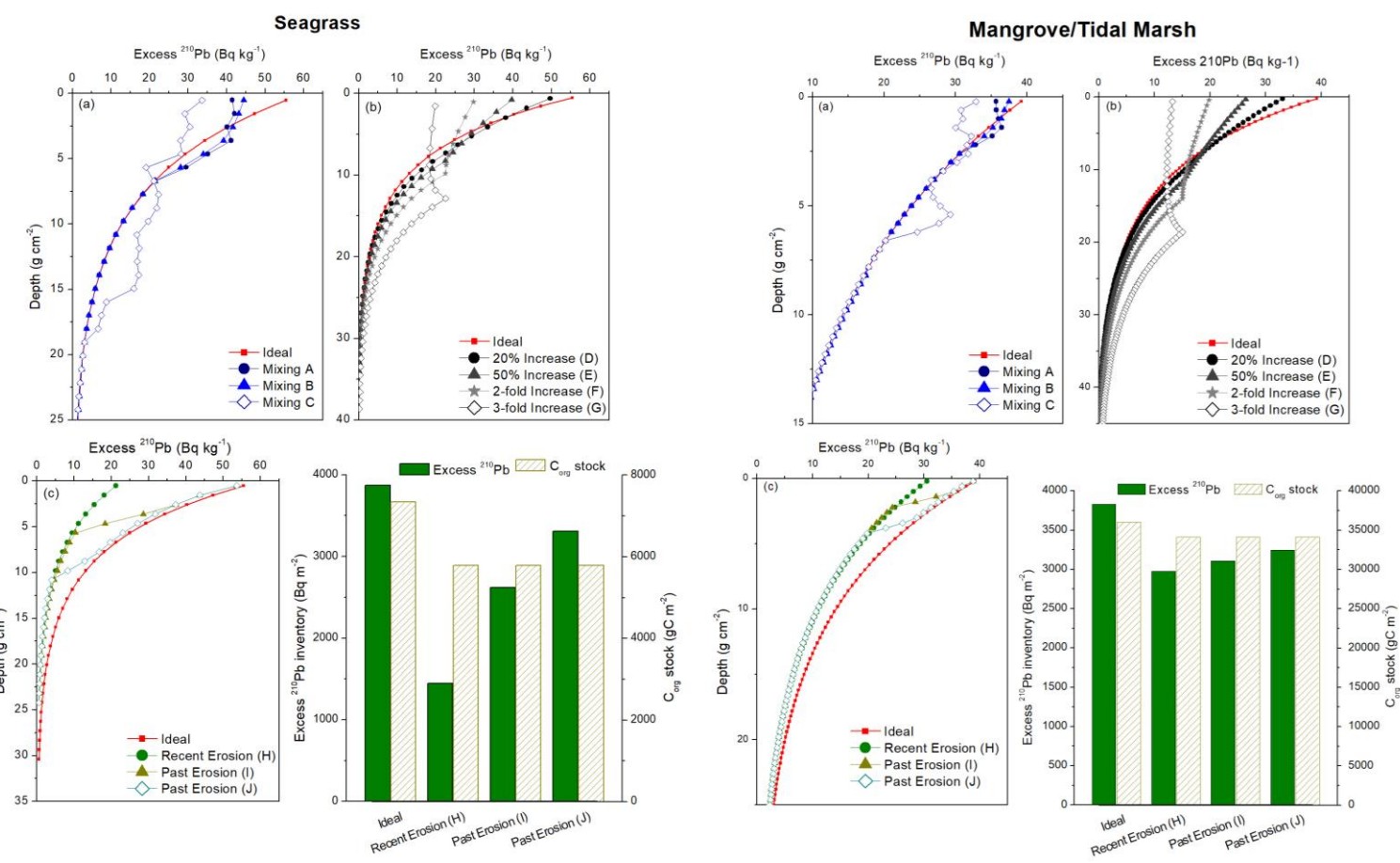

**Figure 3.** Simulated excess $^{210}$Pb concentration profiles of mixing (a), increase in sedimentation rates (b) and erosion processes (c) in vegetated coastal sediments. Several dry bulk density (DBD) and mass accumulation rates (*MAR*) are used to represent the effects of these processes in seagrass sediments (Left: DBD 1.03 g cm$^{-3}$; *MAR* = 0.2 g cm$^{-2}$ yr$^{-1}$; $C_{org}$ = 2.5%) and in mangroves and tidal marsh sediments (Right: DBD: 0.4 g cm$^{-3}$; *MAR* = 0.3 g cm$^{-2}$ yr$^{-1}$; $C_{org}$ = 8%). Bar charts illustrate the deficits in excess $^{210}$Pb inventories and $C_{org}$ stocks after erosion events. See appendix A for detailed description of each scenario.

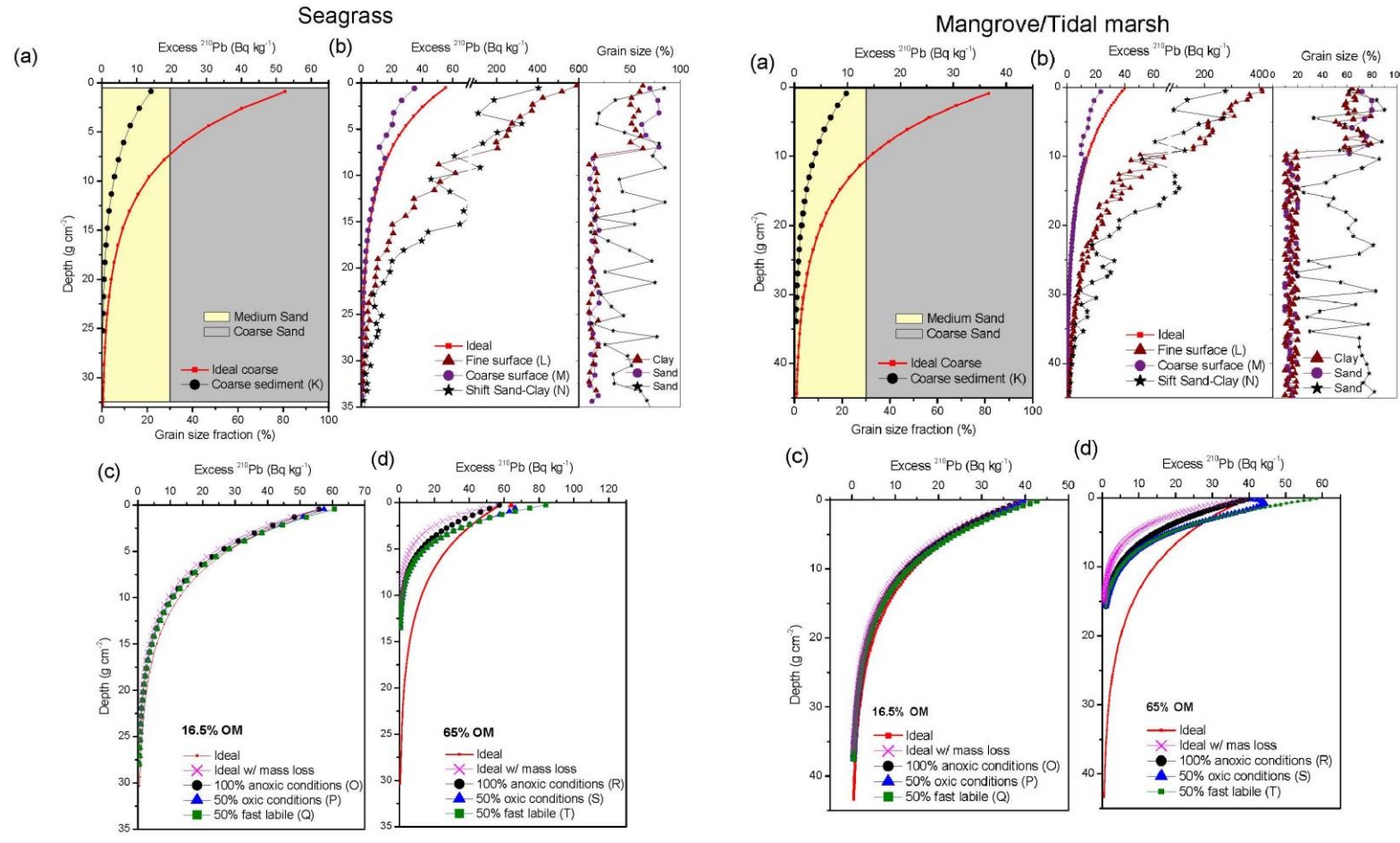

**Figure 4.** Simulated excess [210]Pb concentration profiles resulting from changes in sediment composition and organic matter decay. Coarse homogeneous grain size (a); heterogeneous grain size with depth (b), were triangles and dots represent an excess [210]Pb profile in sediments consisting of fines (< 63 μm) or sands (> 125 μm) at surface layers, respectively; (c and d) organic matter decay from starting level of 16.5% and 65%, respectively (considering different scenarios described in appendix A) in seagrass (Left: DBD 1.03 g cm$^{-3}$; MAR = 0.2 g cm$^{-2}$ yr$^{-1}$) and mangrove/tidal marsh sediments (Right: DBD: 0.4 g cm$^{-3}$; MAR = 0.3 g cm$^{-2}$ yr$^{-1}$).

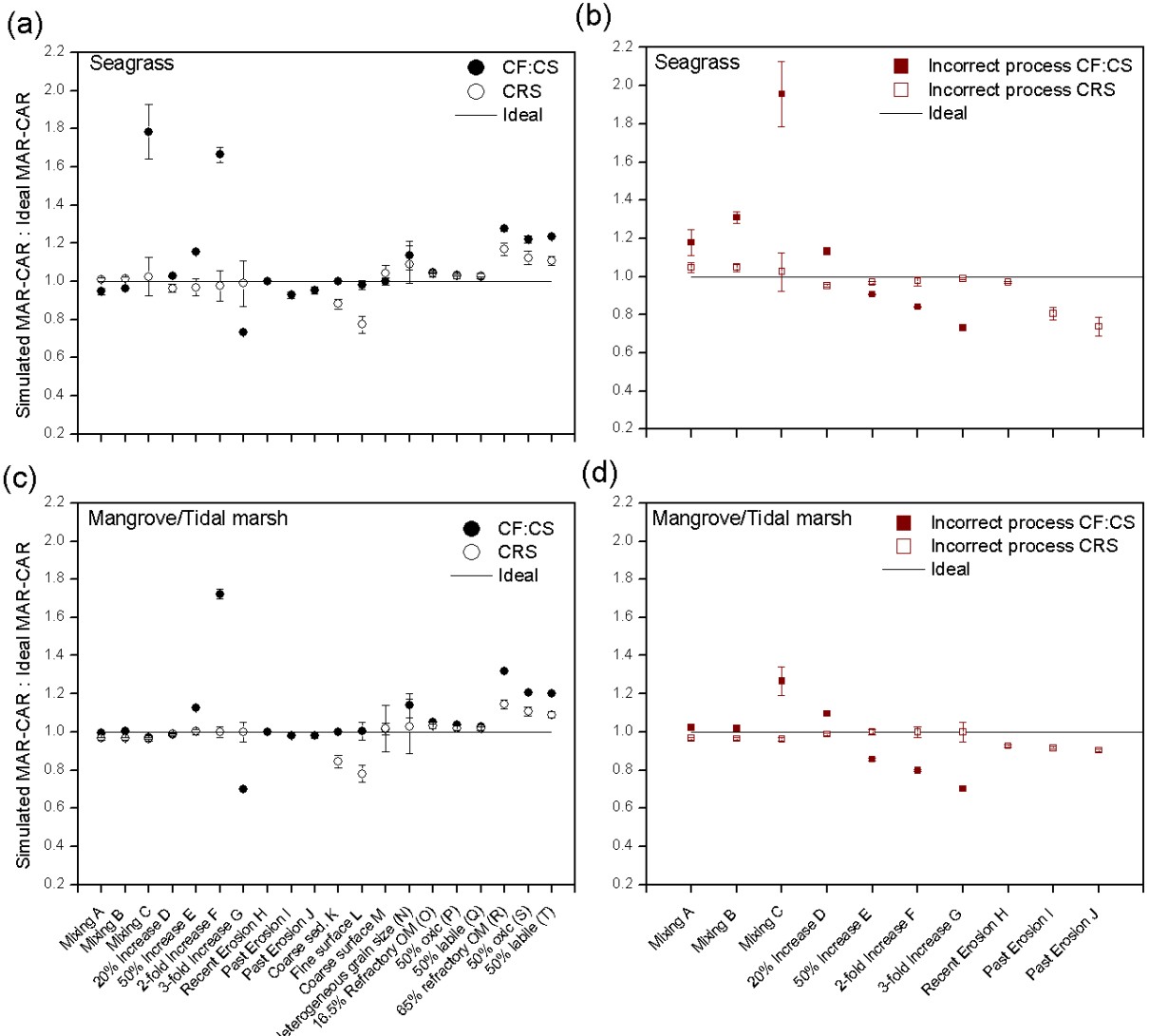

**Figure 5.** Ratio of average 100-yr $C_{org}$ accumulation rates (CAR) between simulated and ideal $^{210}$Pb profiles produced by various sedimentary processes in seagrass (a,b) and mangrove/tidal marsh habitats (c,b). Left: the correct process is assumed and models are applied: Right: incorrect process is assumed and models are applied accordingly. Error bars represent the result of error propagation. Uncertainties for mean MAR were derived from SE of the regression and SE of the mean using the CF:CS and CRS models, respectively. Ratios of simulated/ideal sedimentation rates (MAR) are equal to those for CAR, determined from multiplying MAR by the fraction of $C_{org}$ in sediments (Eq. 5), which was considered constant between ideal and simulated profiles. In simulations of increasing sedimentation and organic matter decay, new MAR and CAR were estimated for ideal $^{210}$Pb profiles to represent real changes in accumulation, organic matter decay and associated changes in sediment mass with depth.

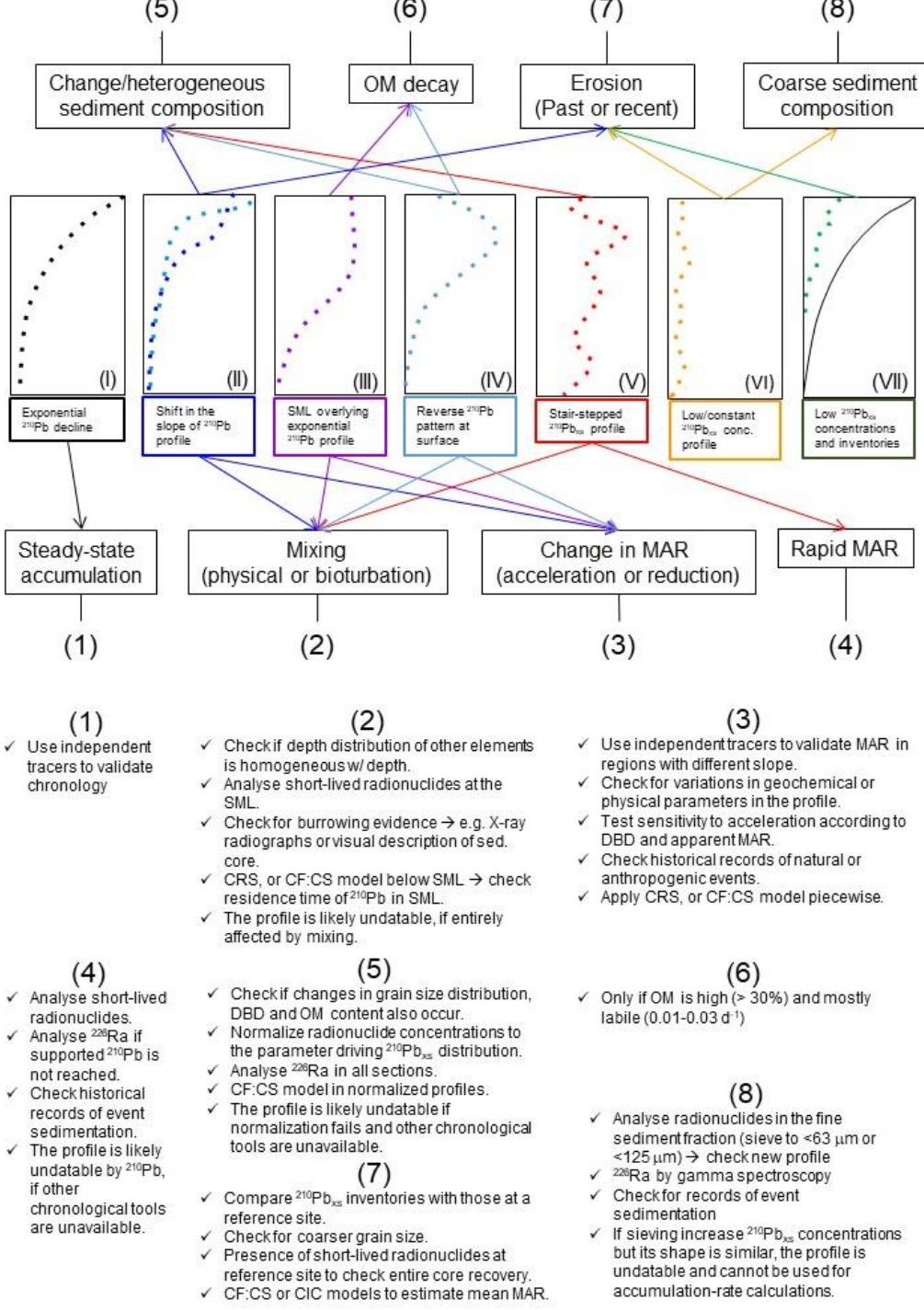

**Figure 6.** Diagnostic features for seven distinct types of sediment accumulation in vegetated coastal sediments (based on excess [210]Pb concentration profiles as shown in Figure 2) and recommended actions to interpret the [210]Pb profiles and the sedimentary processes most likely involved.

**Table 1.** Common values of main parameters of vegetated coastal sediments (seagrass, mangrove and tidal marshes): average dry bulk density (DBD), average sedimentation rates, range of organic matter (OM) content, median organic carbon ($C_{org}$) contents, and decay rate of buried $C_{org}$ (from above ground biomass to refractory sediment $C_{org}$).

| Habitat Type | DBD[a] | Sediment and mass accumulation rate [b] | | OM[c] | $C_{org}$[d] | Decay rate of buried $C_{org}$[e] |
|---|---|---|---|---|---|---|
| | (g cm$^{-3}$) | SAR (mm yr$^{-1}$) | MAR (g cm$^{-2}$ yr$^{-1}$) | (%) | (%) | (d$^{-1}$) |
| Seagrass | 1.03 | 2.0 ± 0.4 | 0.21 ± 0.04 | 0.5-16.5 | 2.5 | 0.01- 0.00005 |
| Mangrove | 0.45 | 5.5 ± 0.4 | 0.25 ± 0.02 | 7-25 | 7.0 | 0.03 – 0.00005 |
| Tidal marsh | 0.43 | 6.7 ± 0.7 | 0.29 ± 0.03 | 5-80 | 9.0 | 0.005 - 0.00005 |

[a] Seagrass (Fourqurean et al., 2012); Mangrove (Donato et al., 2011) and Tidal marsh (Craft, 2007; Hatton et al., 1983).

[b] Seagrass and mangrove (Duarte et al., 2013), and tidal marsh (Kirwan and Megonigal, 2013).

[c] Seagrass (Koch, 2001); Mangrove (Breithaupt et al., 2012); Tidal marsh (Cochran et al., 1998; Ember et al., 1987).

[d] Seagrass (Fourqurean et al., 2012); Mangrove (Breithaupt et al., 2012); Tidal marsh (Chmura et al., 2003).

[e] Seagrass, mangrove and tidal marsh (Lovelock et al., 2017b).

**Table 2**. Summary of the main [210]Pb-based models for sediment dating (adapted from Mabit et al., 2014)

| Model | Assumptions | Analytical Solutions | References |
|---|---|---|---|
| CIC: Constant Initial Concentration | [1], $\Phi(t)/MAR(t) = Cte$ | $C_m = C_0 \cdot e^{-\lambda t}$ | (Robbins, 1978; Robbins and Edgington, 1975) |
| CF:CS: Constant Flux: Constant Sedimentation | [1], [2], [3] | $C_m = C_0 \cdot e^{-\lambda m/MAR}$; $t = \dfrac{m}{MAR}$ | (Krishnaswamy et al., 1971) |
| CRS: Constant Rate of Supply | [1], [2] | $A_m = I \cdot e^{-\lambda t}$; $MAR = \dfrac{\lambda A_m}{C_m}$ | (Appleby, 2001; Appleby and Oldfield, 1978) |
| CMZ:CS Complete Mixing Zone with constant SAR | [2], [3], $k_m = \infty, m \geq m_a$ $k_m = 0, m < m_a$ | $C_m = C = \dfrac{\Phi}{MAR + \lambda m_a}, m \geq m_a$ $C_m = C \cdot e^{-\lambda(m-m_a)/MAR}, m < m_a$ | (Robbins and Edgington, 1975) |
| CF:CS-Constant Diffusion | [2], [3], $k_m = Cte$ | $C_m = \dfrac{\Phi}{MAR - k_m\beta} e^{-\beta m}$; $\beta = \dfrac{MAR - \sqrt{MAR^2 + 4\lambda k_m}}{2k_m}$ | (Laissaoui et al., 2008; Robbins, 1978) |
| CF:CS-depth dependent diffusion and/or translocational mixing | [2], [3], $k_m = f_m$; may include local sources and sinks | General numerical solution | (Abril, 2003; Abril and Gharbi, 2012; Robbins, 1986; Smith et al., 1986) |
| IMZ: Incomplete Mixing Zone | [2], [3] | A linear combination of solutions for CF-CS and CMZ-CS with coefficients g and $(1 - g)$, being $g \in [0, 1]$ | (Abril et al., 1992) |
| SIT: Sediment Isotope Tomography | [1] | $C_m = C_0 \cdot e^{-B \cdot m} \cdot$ $\cdot e^{\sum_{n=1}^{N} a_n sin\left(\frac{n\pi m}{m_{max}}\right) + \sum_{n=1}^{N} b_n\left(1 - cos\frac{n\pi m}{m_{max}}\right)}$ | (Carroll and Lerche, 2003) |
| NID-CSR: Non-Ideal-Deposition, Constant Sedimentation Rate | [1], [2], [3], fractioning of fluxes, depth distribution | $C_m = C_1 \cdot e^{-\lambda m/MAR} + C_2 \cdot e^{-\alpha m}$; $C_2 = \dfrac{-\alpha g\Phi}{\alpha MAR - \lambda}$; $C_1 = \dfrac{(1 - g)\Phi}{MAR} - C_2$ | (Abril and Gharbi, 2012) |
| CICCS: constant initial concentration and constant sedimentation rate | [1], [2] | $MAR = \lambda \dfrac{I - I_{ref}}{C_r}$; $I_{ref}$ = local fallout [210]Pb inventory; $C_r$ = Initial excess [210]Pb in catchment-derived sediment. | (He and Walling, 1996b) |
| IP-CRS: Initial Penetration-Constant Rate of Supply | [2], initial mobility of excess [210]Pb downward; two compartments 0 to $z_k$ and $z_k$ to $\infty$ | $C_{i\,(z)} = A_i e^{\theta + (i)z} + B_i e^{\theta - (i)z}$; $from\ 0\ to\ z_k$ $C_{i\,(z)} = A_i e^{\sigma + (i)z} + B_i e^{\sigma - (i)z} + \dfrac{F_i}{\lambda}$; $from\ z_k\ to\ \infty$ $F_i = \dfrac{f_i}{(z_i - z_{i-1})}\sum_{m=1}^{k}\int_{z_{m-1}}^{z_m} r_m C_m\, dz$; $\sum f_i = 1$ *See reference for constants* | (Olid et al., 2016) |
| TERESA: Time estimates from random entries of | [1], excess [210]Pb fluxes are governed by horizontal inputs, correlation with MAR | $C_1 = C_0 \cdot e^{-\lambda T_0} \cdot \dfrac{1 - e^{-\lambda\Delta T_1}}{\lambda\Delta T_1}$ $C_m = C_0 \cdot e^{-\lambda\left(T_0 + \frac{\Delta_{m-1}}{MAR_{m-1}}\right)} \cdot \dfrac{1 - e^{-\lambda\Delta T_m}}{\lambda\Delta T_m}$ | (Abril, 2016; Botwe et al., 2017) |

| | sediments and activities |
|---|---|

[1] Non post-depositional redistribution; [2] constant excess $^{210}$Pb fluxes at the SWI; [3] constant MAR. All models assume continuity of the sediment sequence.

$C_m$: excess $^{210}$Pb activity concentration in sediments at mass depth m

I: total inventory of excess $^{210}$Pb

5    $A_m$: excess inventory accumulated below depth m

$k_m$: effective mixing coefficient ($D\rho^2$)

$m_a$: mass thickness of top sediment zone

$\Phi$: Flux of excess $^{210}$Pb onto the sediment

g: fraction of excess $^{210}$Pb flux distributed within a certain mass depth

10    $F_i$: additional supply of excess $^{210}$Pb to layer $i$

**Table 3.** Summary description of the numerical simulations conducted to test for the effects of sedimentary processes on excess $^{210}$Pb concentration profiles in seagrass and mangrove/tidal marsh sediments. MAR and CAR results derived from simulated profiles were compared with MAR and CAR estimates derived from the ideal excess $^{210}$Pb profiles reported here. $k_s$ is the decay rate of the refractory sediment organic matter (OM) under anoxic conditions and $k_{ox}$ is that in oxic conditions. $K_{lb}$ is the decay constant of the labile OM derived from seagrass and mangrove/tidal marsh ecosystems (0.01 yr$^{-1}$ and 0.03 yr$^{-1}$, respectively).

| Influencing Factor | Scenario | Description | MAR Ideal profile (g cm$^{-2}$ yr$^{-1}$) | | CAR Ideal profile (g C$_{org}$ m$^{-2}$ yr$^{-1}$) | |
|---|---|---|---|---|---|---|
| | | | Seagrass | Mangrove/Tidal marsh | Seagrass | Mangrove/Tidal marsh |
| Mixing | A | Random upper 5 cm | 0.20 | 0.30 | 50 | 240 |
| | B | Random upper 5 cm | 0.20 | 0.30 | 50 | 240 |
| | C | Random upper 5-10 cm | 0.20 | 0.30 | 50 | 240 |
| Increasing MAR in recent years | D | Increased basal MAR by 20% | 0.21 | 0.31 | 52 | 248 |
| | E | Increased basal MAR by 50% | 0.22 | 0.32 | 54 | 259 |
| | F | Increased basal MAR by 100% | 0.23 | 0.35 | 59 | 278 |
| | G | Increased basal MAR by 200% | 0.27 | 0.40 | 67 | 317 |
| Erosion | H | Removal of excess $^{210}$Pb inventory from 0-5 cm | 0.20 | 0.30 | 50 | 240 |
| | I | Removal of excess $^{210}$Pb inventory from 5-10 cm | 0.20 | 0.30 | 50 | 240 |
| | J | Removal of excess $^{210}$Pb inventory from 10-15 cm | 0.20 | 0.30 | 50 | 240 |
| Grain size | K | Coarse sediment (70% coarse, 30% medium) | 0.20 | 0.30 | 50 | 240 |
| | L | Fine surface sediments (50 - 80% of clays at surface) | 0.20 | 0.30 | 50 | 240 |
| | M | Coarse surface sediments (50 - 80% of sands at surface) | 0.20 | 0.30 | 50 | 240 |
| | N | Heterogeneous grain size (alternated sand layers with clay layers) | 0.20 | 0.30 | 50 | 240 |
| Organic matter decay | | 16.5% OM | | | | |
| | O | 100% with: ks = 0.00005 d$^{-1}$ | 0.17 | 0.25 | 34 | 150 |
| | P | 50% with $kox$ = 0.0005 d$^{-1}$ | 0.17 | 0.25 | 16 | 116 |
| | Q | 50% with $k_{lb}$ = 0.01 d$^{-1}$ or 0.03 d$^{-1}$ | 0.17 | 0.25 | 14 | 111 |
| | | 65% OM | | | | |
| | R | 100% with: ks = 0.00005 d$^{-1}$ | 0.07 | 0.10 | 62 | 156 |
| | S | 50% with $kox$ = 0.0005 d$^{-1}$ | 0.07 | 0.10 | 33 | 100 |
| | T | 50% with $k_{lb}$ = 0.01 d$^{-1}$ or 0.03 d$^{-1}$ | 0.07 | 0.10 | 30 | 94 |

| Box 1. Case study of a sedimentation event |
|---|

Hurricanes and cyclones can lead to the sudden delivery of large amounts of sediments and nutrients to mangroves and tidal marshes, which in turn can result in enhanced production (Castañeda-Moya et al., 2010; Lovelock et al., 2011). Smoak et al. (2013) obtained an excess $^{210}$Pb concentration profile consistent with a large pulse of sediment delivered to fringing mangroves in the Everglades, Florida (Panel A). The concentration of excess $^{210}$Pb was vastly different (several times lower) in sediments accumulated during the event. The sediment accumulation rate estimated by the CRS model for the upper part of the sediment record was six times that of background levels, resulting in a doubled accretion rate, due to the high bulk density of the delivered sediments (Castañeda-Moya et al., 2010). $C_{org}$ concentrations in the abruptly accumulated sediments were lower (5%) than those of the sediments beneath the event layer (20-25%). In fact, event-deposits could consist of coarse sediments (for instance sand, shell and carbonate sediment layers deposited during storm events characteristic of offshore environments; (Swindles et al., 2018)), but also of fine sediments that could present lower excess $^{210}$Pb specific activity compared to surrounding layers (e.g., siltation events due to clearing of the catchment area; Cambridge et al., 2002; Serrano et al., 2016d). Indeed, if the initial excess $^{210}$Pb concentration ($C_0$) is known, the CIC model could be useful to constrain dating when it is difficult to precisely define the thickness of such deposits.

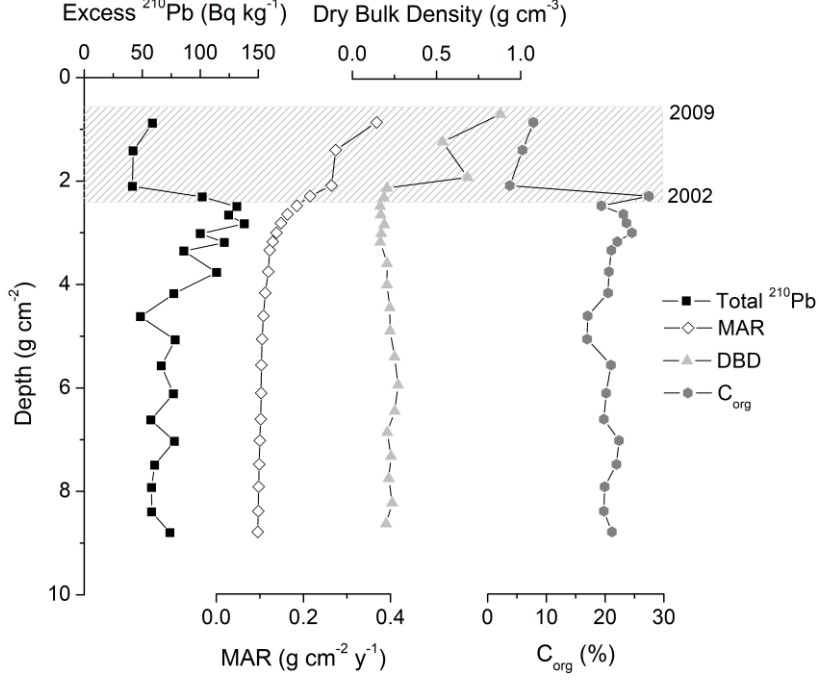

**Panel A.** Excess $^{210}$Pb, mass sedimentation rates (MAR), dry bulk density and $C_{org}$ content in a mangrove sediment core at the Everglades, Florida. The gridded area represents the period 2002 - 2009, when Hurricane Wilma (2005) delivered a large pulse of sediment (Adapted from Smoak et al., 2013).

| Box 2. Case Study of Mixing |
| --- |

An example of bioturbation processes is documented by Smoak and Patchineelam (1999) where they showed a mixed excess $^{210}$Pb profile down to 11 cm depth in a mangrove ecosystem in Brazil evidenced from the $^{210}$Pb, $^{234}$Th and $^{7}$Be concentration profiles. The excess $^{210}$Pb concentration decreases exponentially below the surface mixed layer, resulting in an estimated accumulation rate of 1.8 mm yr$^{-1}$. In the upper layers the excess $^{210}$Pb follows a complex pattern, with alternate relative maxima and minima, which could be representative of varying conditions of fluxes and sediment accumulation rates, presence of coarse sediments or physical or biological mixing. However, $^{7}$Be penetrated down to 4 cm depth and excess $^{234}$Th was detected only in the surface layer. Sediments that are buried for a period of more than 6 months will have undetectable $^{7}$Be, hence its presence at 4 cm depth indicated that the activity of benthic communities had remobilised it downwards to a much greater degree than sedimentation.

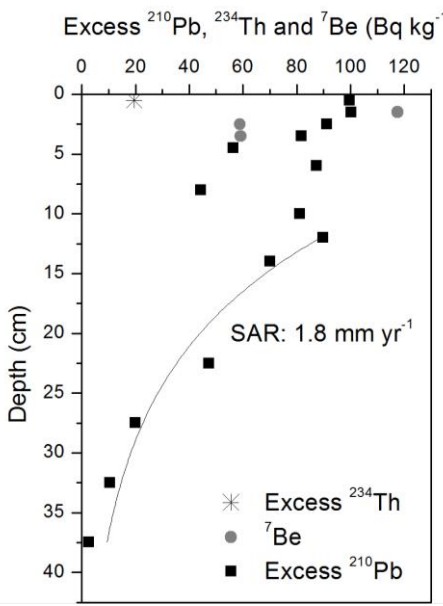

**Panel B.** Excess $^{210}$Pb concentration profile affected by bioturbation. Short-lived $^{7}$Be and excess $^{234}$Th concentration profiles are indicators of mixing in the zone of constant excess $^{210}$Pb concentrations (0 - 5 cm). (Adapted from Smoak and Patchineelam, (1999).

| Box 3. Case Study of rapid sedimentation rates |
|---|

Alongi et al. (2005) studied the rates of sediment accumulation at three mangrove forests spanning the intertidal zone along the south coastline of the heavily urbanized Jiulongljiang Estuary (China). Mass accumulation rates (MAR) were rapid and one of the excess [210]Pb concentration profiles showed scattered concentrations with depth. This could be related to either a very high MAR during the last decades or a very intense mixing down core. However, the excess [228]Th concentration profile, determined from the difference between the total [228]Th and [228]Ra concentrations in the sediment, showed a clearly decaying trend down to 15 cm (Panel C). The exponential decay curve fitted to the excess [228]Th concentrations yielded an accumulation rate of 10 cm yr$^{-1}$, which was consistent with the [210]Pb concentration profile. Therefore, the evidence provided by excess [228]Th indicated that a very high MAR was the most plausible processes responsible for the sediment record.

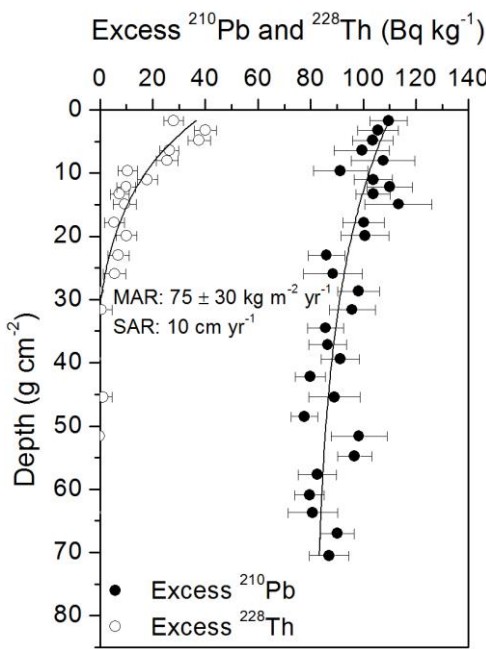

**Panel C.** Vertical concentration profiles of excess [210]Pb and [228]Th in core 3564 fom Alongi et al. (2005), produced by a rapid mass accumulation rate.

| Box 4. Case Study of Erosion |
| --- |

Incomplete inventories of excess [210]Pb indicative of erosion can be illustrated by the measured [210]Pb concentration profiles in sediments from Oyster Harbor (Albany, Western Australia), some of which were devoid of seagrass vegetation since the 1980s due to eutrophication (Marbà et al., 2015). The measured excess [210]Pb concentrations in the unvegetated sediments were relatively low, and the horizons of excess [210]Pb were detected at shallower sediment depths than in neighbouring sediments, where seagrass meadows persisted (Panel D). The inventory of excess [210]Pb in the unvegetated sediment exhibited a deficit of 722 Bq m$^{-2}$ compared to that in the vegetated site. This deficit could not solely result from the lack of accumulation of excess [210]Pb while sediments were unvegetated (30 years; atmospheric flux of 25 Bq m$^{-2}$ yr$^{-1}$), but also to the subsequent sediment erosion. These results, combined with C$_{org}$ analyses, showed that unvegetated sediments had an average deficit in accumulated C$_{org}$ stocks of 2.3 kg C$_{org}$ m$^{-2}$ compared to vegetated sediments over the last ca. 100 years. This deficit was produced since seagrass loss in 1980, but was equivalent to a loss of approximately 90 years of C$_{org}$ accumulation.

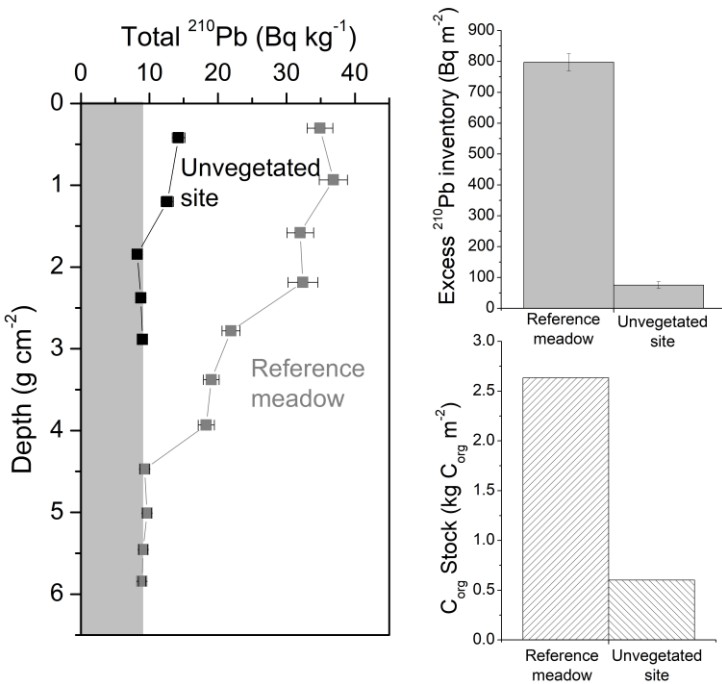

**Panel D.** Comparison of [210]Pb concentration profiles and inventories of excess [210]Pb and organic carbon (C$_{org}$) between vegetated and unvegetated site. The grey area indicates supported [210]Pb concentrations (Adapted from Marbà et al., 2015).