# Peer review of "Reviews and syntheses: $^{210}$Pb-derived sediment and carbon accumulation rates in vegetated coastal ecosystems - setting the record straight"

_Biogeosciences, 2018_

## Referee Comment (RC1) · Anonymous Referee #3 · 14 Jun 2018

General comments Overall, a very valuable contribution to the literature. This is a helpful synthesis of the literature that will be a go-to for those in the field, and it is also an interesting modeling exercise that sheds light on the processes producing various 210Pb patterns. My main concern is that the manuscript provides an overly optimistic view of the errors associated with complex 210Pb profiles, for reasons explained below.

Specific comments As the authors note in Table 4, patterns II, III, and IV can have multiple causes. Especially common, and especially problematic, is the difficulty in distinguishing between mixing and an increase in MAR. The simulation studies in this paper don't address this adequately because they separate the mixing simulations from

the increased sedimentation simulations. For the mixing simulations, for example, "the CF:CS model was applied below the depth of the visually apparent SML (3 cm) in scenarios A and B to avoid overestimation of MAR" (Appendix). But if you didn't know this profile was created by mixing, how would you know that you would be overestimating MAR rather than accurately estimating an increase in MAR? In other words, in the real world, how would you know whether it was mixing (so leave out the SML) or increased MAR? True, the mixing and increased sedimentation profiles in Figure 3 do look somewhat different, but I am not convinced that in the real world they are so easily distinguished. Bottom line: I am concerned that if the authors tested the error in non-ideal profiles without knowing what caused them, they would find higher errors than those shown in Figure 5.

Related to the above: the authors choose not to create a CRS estimate for the profiles with erosion. That is fine as long as one knows that erosion is a factor. In the real world, minor deviations from the ideal inventory (especially the small ones shown in the tidal marsh half of Figure3c) do not generally preclude investigators from applying the CRS method. I would strongly encourage the authors to apply CRS to these profiles to get a sense of how large the associated errors are. At a minimum, they should caution others not to use the CRS method with profiles that show deviations from the expected inventory.

The authors use their results to suggest in Figure 5 and Table 4 that pretty much any 210Pb profile is date-able (except those with extreme OM concentrations). However, in the real world, some profiles are likely to be altered in more complex ways than the simulations shown here – by mixing and erosion and different grain sizes. I believe that some profiles may just be too altered to be retrievable, and would suggest using extreme caution in interpreting Types V, VI, and VII. Section 4 of the paper is very helpful in suggesting alternative approaches that can help disentangle various factors, but it is in tension with Figure 5 (and the abstract), which suggest that those are not necessary, since maximum error is only 20% anyway.

It would be helpful if the Supplementary Tables in Excel had formulas rather than just values, to make it easier to understand how the simulations were done.

I think the authors could emphasize more strongly that they are looking at the 100-year average MAR and Corg-MAR, not the patterns over time. For example, the y-axis in Figure 5 (or at least the figure caption) could say "100-year Corg burial."

Does this analysis only apply to Corg burial? There will be an audience interested in the equivalent of Figure 5 for the MAR itself, which presumably would be easy to make.

Table 4 is too long and repetitive; there must be a way to condense it, since the options for each outcome are the same.

I found the boxes helpful, except for Box 4, which is different from the others and not necessary in my opinion.

I understand the logic of including the methods in an appendix – mostly because they are quite long and detailed. But it is important for the reader to understand what the authors are doing. The authors might consider including in the methods a more detailed description than what is there now (but still less detailed than in the appendix).

Section 2.1 doesn't seem like it should be in the methods.

The authors mention a literature review several times, but the only detail is provided on p. 4 line 27ff. in establishing that CIC, CRS, and CFCS are the most commonly used approaches. Is this the same literature review that was used to construct Figure 2? Please clarify. Also, they probably missed some of the literature by not including the term Pb-210, which is sometimes used instead of 210Pb. (There are almost certainly more than 150 uses of 210Pb in the salt marsh literature.)

The reason for excluding the CIC method – the absence of ideal profiles – is not persuasive as currently expressed. The other methods also suffer when there are deviations from the ideal profile, which is exactly what the authors explore. Perhaps more of a justification for excluding CIC could be given?

I'm not sure the distinction between Types VI and VII is necessary. They are both characterized by low inventories, regardless of profile shape.

---

## Referee Comment (RC2) · Anonymous Referee #4 · 27 Jun 2018

The review paper presented by Arias-Ortiz discuss the use of the 210Pb dating technique to estimate the rate of mass accumulation in vegetated coastal ecosystems. Such information is indeed very important in considering the significant role of vegetated coastal habitats (tidal marsh, mangrove, seagrass) as sinks of carbon. Over the last 150 years, 210Pb is the only tool that permits to calculate sediment and carbon accumulation rates (SAR/CAR) in such environments. However, the application of the 210Pb-based method is not tricky in these environments. The authors aim to illustrate the models usually applied to calculate SAR or MAR in these setting. This article is extremely timely as there is a growing interest in better estimate C source/sink. The authors are presenting in a correct way the principle and the conditions of the 210Pb

method. Although the article is mostly dedicated to the models, there are some recommendation on the 210Pb determination and a comment of the interest of additional time marker (like 137Cs) or normalisation. In fact I regrets that the authors do not develop the experimental section. Indeed, it would be of great interest to provide recommendations about sampling: core description, porosity determination etc. It is also important to precise more clearly the advantage of gamma counting compared to alpha counting. In addition to avoid chemistry step, gamma spectrometry has the major advantage to determined simultaneously 210Pb and its supported parent (226Ra), 137Cs, 228Th, 7Be, 40K among others I am surprised that the authors mentioned 228Th as a potential dating/bioturbation tracer. In such coastal environment, I usually use 228TH as 232Th its grandfather to trace the detrital fraction. It is a good way to normalize also radionuclide activities. I think it is also important to point out the need to well consider the samples. In the case of sediments presenting coarse fraction or vegetal debris, it could be useful to separate the fine sediment fraction, that supports 210Pb, from the other fractions (that dilute its activity). In fact it is the first step to do : how to obtain the best 210Pbxs profile depending of the sediment. It could help to reduce variability in the 210Pbxs profile. The authors need to develop this aspect. In fact I am convinced that some model adaptions are not required if sampling and measurements are done in an appropriate way (see figure panel D why measure with the sandy fraction).

Other comments: - the authors need to check the manuscrit in order to verify the terms and acronyms (like Db and not D for bioturbation) . - Page 2 line 24: "210Pb is not affected by interannual variability" : to moderate 210Pbxs fluxes could have some variability although moderate - Page 3 line 3: and subsequent fallout - Page 4 (and in all the text): be careful to use correctly concentration and activity - CIC model/ I disagree with the statement CIC is not appropriate. This model could be useful in some sediment core presenting event-deposit (like flood). Such deposits could be sand, but also fine sediments that could present lower 210Pbxs (compared to surrounding layers). In fact case, CIC could be useful to check dating when it is difficult to precisely define the thickness of such deposits. - page 7 type II: lower activities could be also

explained by dilution by roots for example, so it is important as indicated previously to provide recommendations for sampling. - 13 line 30-34: the presence of large OC concentration or vegetal (like leaves) could promote high concentration of Cs due to mobility. So care is required with 137Cs - page 15 line5-7: not clear, it seems there is a confusion between alpha (that requires to assume the rather constant 210Pb activities correspond to the supported 210Pb) and gamma (that determines both 210Pb and 226ra)).

---

## Author Comment (AC1) · 17 Jul 2018

Dear Dr. Sarin,

Below, please find our response to the comments raised by the reviewers to the manuscript entitled "Reviews and syntheses: 210Pb-derived sediment and carbon accumulation rates in vegetated coastal ecosystems: setting the record straight", along with a description of the changes we suggest to improve the manuscript. We are very grateful to the reviewers for their thoughtful and constructive comments and we address below each of the points they raised.

[Figure]

Ariane Arias-Ortiz on behalf of the authors

RESPONSE TO COMMENTS BY ANONYMOUS REFEREE #3

General comments: Overall, a very valuable contribution to the literature. This is a helpful synthesis of the literature that will be a go-to for those in the field, and it is also an interesting modeling exercise that sheds light on the processes producing various 210Pb patterns. My main concern is that the manuscript provides an overly optimistic view of the errors associated with complex 210Pb profiles, for reasons explained below.

We sincerely thank the reviewer for acknowledging the interest of our work as well as for his/her constructive comments, which were very helpful in improving the manuscript. We will include in the revised manuscript a more throughout discussion showing the implications of estimating accumulation rates in complex 210Pb profiles.

1. Specific comments: As the authors note in Table 4, patterns II, III, and IV can have multiple causes. Especially common, and especially problematic, is the difficulty in distinguishing between mixing and an increase in MAR. The simulation studies in this paper don't address this adequately because they separate the mixing simulations from the increased sedimentation simulations. For the mixing simulations, for example, "the CF:CS model was applied below the depth of the visually apparent SML (3 cm) in scenarios A and B to avoid overestimation of MAR" (Appendix). But if you didn't know this profile was created by mixing, how would you know that you would be overestimating MAR rather than accurately estimating an increase in MAR? In other words, in the real world, how would you know whether it was mixing (so leave out the SML) or increased MAR? True, the mixing and increased sedimentation profiles in Figure 3 do look somewhat different, but I am not convinced that in the real world they are so easily distinguished. Bottom line: I am concerned that if the authors tested the error in non-ideal profiles without knowing what caused them, they would find higher errors than those shown in Figure 5.

RESPONSE: We agree that mixing and increasing mass accumulation rate processes

are difficult to distinguish, however there are some complementary analyses that can help to distinguish each of these processes. Such actions/analyses are explained in section 4, particularly in subsections 4.1, 4.2 and 4.3 of the original version of the manuscript (pages 12-15). We think it is indeed interesting to show to the reader the consequences of mismatching the process and how they translate in higher errors in sediment and Corg accumulation rates (CAR).

ACTIONS after Editor's consideration: In the revised version of the manuscript we will run the 210Pb models in mixing simulations assuming that the observed anomalies were caused by increased mass accumulation rates and vice versa in increasing sedimentation simulations. We will add a discussion in section 3.2.6 "General remarks" (page 11, following line 20) related to the resulting MAR and CAR if the incorrect process is assumed and dating models are applied. CAR results of incorrect process interpretation will be also plotted in Figure 5 so the reader can easily be aware of the potential errors associated if the processes causing anomalies in 210Pb concentration profiles are not well identified. In addition, we will reorganize section 4 according to the processes simulated in section 3 (mixing, increasing MAR, erosion, changes in gran size and OM decay). Thus, the alternative analyses and potential actions will be merged into a single section so it is straightforward for the reader to pick actions to identify each of these processes. Section 4 will be restructured as follows:

"4. Approaches and guidelines

4.1 General validation of 210Pb models

- Artificial radionuclides

- Geochemical information of sediments

4.2 Mixing or Rapid sedimentation

- Geophysical analyses (i.e, X-ray radiographies, CAT scans)

- Short-lived radionuclides

- Geochemical element profiles

4.3 Erosion

- Excess 210Pb inventories in reference and disturbed sites

4.4 Heterogeneous sediment composition

- Normalization of excess 210Pb profiles

- 226Ra profiles"

2. Related to the above: the authors choose not to create a CRS estimate for the profiles with erosion. That is fine as long as one knows that erosion is a factor. In the real world, minor deviations from the ideal inventory (especially the small ones shown in the tidal marsh half of Figure3c) do not generally preclude investigators from applying the CRS method. I would strongly encourage the authors to apply CRS to these profiles to get a sense of how large the associated errors are. At a minimum, they should caution others not to use the CRS method with profiles that show deviations from the expected inventory.

RESPONSE and ACTIONS: We agree and we will also apply the CRS model to the simulated eroded profiles and plot the results in Figure 5. Additionally, in section 3.2.6 "General remarks" (page 11), text will be added to emphasize those problems associated to the application of the CRS model in incomplete sediment records, and the even older ages for deeper sections and the bias in calculated MAR and CAR as explained by MacKenzie et al. (2011).

REFERENCES:

-MacKenzie, A. B., Hardie, S. M. L., Farmer, J. G., Eades, L. J. and Pulford, I. D.: Analytical and sampling constraints in 210Pb dating, Sci. Total Environ., 409(7), 1298–1304, doi:10.1016/j.scitotenv.2010.11.040, 2011.

3. The authors use their results to suggest in Figure 5 and Table 4 that pretty much any

210Pb profile is dateable (except those with extreme OM concentrations). However, in the real world, some profiles are likely to be altered in more complex ways than the simulations shown here – by mixing and erosion and different grain sizes. I believe that some profiles may just be too altered to be retrievable, and would suggest using extreme caution in interpreting Types V, VI, and VII. Section 4 of the paper is very helpful in suggesting alternative approaches that can help disentangle various factors, but it is in tension with Figure 5 (and the abstract), which suggest that those are not necessary, since maximum error is only 20% anyway.

RESPONSE: This is correct, and we probably failed to capture this point in the original version of the manuscript. Although some research reports extremely altered sediment profiles, these are few since a literature bias exists towards those profiles where dating or MAR estimates could be achieved. However, as the reviewer comments, more often than not, some profiles are likely to be altered by a composite of processes, leading to types V and VI. This is especially true in seagrass ecosystems that present lower sedimentation rates and can occur in sand-dominated substrates, where 210Pb is less preferentially adsorbed. For instance, Saderne et al. (2018) collected 9 and 11 sediment cores in seagrass and mangroves of the Red Sea, respectively, but none of the seagrass and only 4 of the mangrove sediment cores were useful for the determination of MAR and CAR. In the revised version of the manuscript we will capture this point raised by the reviewer through 3 main actions (see below)

ACTIONS after Editor's consideration:

- We will modify the deep mixing simulation (scenario C) so mixing influences the entire excess 210Pb profile to simulate type V profile forms. Both results, assuming the process causing this anomaly in the 210Pb profile is mixing or fast accumulation rate, can be plotted in Figure 5, which indeed will increase the errors associated to the estimation of mean MAR and CAR. Text in the abstract and in the results section "3.2.6 General remarks" of the original version of the manuscript will be modified accordingly.

- Section "4. Approaches and Guidelines" will include a short text recommending being critical with the data and acknowledging when a profile is not datable. Text will read: "Using the above procedures, it has been possible to determine mean accumulation rates and, in some instances, also obtain credible chronologies. However, we should be especially cautious in those sites where sediments might be altered by multiple processes (leading to profile types V or VI) and where other chronological tools are not available (e.g., 137Cs). Sites/Cores that have slow accumulation rates and/or intense mixing are unlikely datable, since the excess 210Pb concentration profiles may be unsolvable and overprinted by other post-depositional processes. Environments with complex sedimentation that results in significant variations in grain size and irregular deposition (e.g., hiatal surfaces or erosion) may also be difficult or impossible for calculating accumulation rates. Mistakes would include assigning discrete ages in mixed sediments or extrapolate an age-depth model for a core that should be considered undatable to depths down the core or to nearby sites."

- In Table 4, a new recommended action will be added for profile types V, VI provided the other recommended actions fail: "Acknowledge that the core cannot be used for geochronology/ MAR and CAR cannot be estimated accurately".

REFERENCES:

- Saderne, V., Cusack, M., Almahasheer, H., Serrano, O., Masqué, P., Ariasâ̆ŘOrtiz, A., ... & Duarte, C. M. (2018). Accumulation of carbonates contributes to coastal vegetated ecosystems keeping pace with sea level rise in an arid region (Arabian Peninsula). Journal of Geophysical Research: Biogeosciences, 134, 1498-1510.

4. It would be helpful if the Supplementary Tables in Excel had formulas rather than just values, to make it easier to understand how the simulations were done.

RESPONSE: We agree and we will add formulas in the supplementary Tables.

5. I think the authors could emphasize more strongly that they are looking at the 100-

year average MAR and Corg-MAR, not the patterns over time. For example, the y-axis in Figure 5 (or at least the figure caption) could say "100-year Corg burial."

RESPONSE: We agree with the reviewer and in the revised version of the manuscript we will make this point clearer, not only in Figure 5 but also in the Methods and Results sections of the revised version of the manuscript.

ACTIONS after Editor's consideration:

- In the Methods section "2.2 Numerical simulation" (page 6) text will be added (line 22) "The CF:CS and CRS dating models were applied to altered excess 210Pb profiles to determine the average MAR for the last century."

- In the Results section "3.2 Simulated sediment and Corg accumulation rates (MAR and CAR)" (page 8) text will be added in line 7 "We estimated mean 100-yr MAR and CAR for the simulated profiles by applying the CF:CS and CRS models, and results were compared with those from their respective ideal non-disturbed 210Pb profiles."

- In the Results section "3.2.6. General remarks" (page 11; line 11), text will be added: "Among ecosystems, average last 100-yr MAR and CAR derived from both the CF:CS and the CRS models were less vulnerable to anomalies in mangrove/tidal marsh compared to seagrass sediments" and in lines 16-17 "The decay of OM in very rich organic sediments (> 50% OM) was the process that caused the largest deviations in average 100-yr MAR and CAR in all ecosystems".

- In Figure 5, the figure caption will be modified to read as "Figure 5. Ratio of average 100-yr Corg accumulation rates (CAR) between simulated and ideal 210Pb profiles produced by various sedimentary processes. (a) seagrass and (b) mangrove/tidal marsh habitats."

6. Does this analysis only apply to Corg burial? There will be an audience interested in the equivalent of Figure 5 for the MAR itself, which presumably would be easy to make.

RESPONSE: The ratio between ideal vs. disturbed CAR (Fig. 5) mostly represents

variations in MAR, therefore Figure 5 would look similar for MAR ratios between ideal and disturbed profiles as it is explained in the Figure 5 caption in the original version of the manuscript. For our simulations the C content was considered to be the same in both the disturbed and the ideal excess 210Pb profiles, meaning that the mixed sediments or the newly deposited ones had same Corg (%DW) as those in the ideal non-disturbed profile (2.5% in seagrass sediments and 8% in mangrove/tidal marsh sediments). While any disturbance of the sedimentary record would also affect Corg concentrations due to changes in biogeochemical processes within sediments, the potential and magnitude of such effects is unclear, and therefore, they were not considered here. The aim of the manuscript is to estimate how errors in the estimation of MAR using 210Pb would affect resulting CAR rates and how these errors can be minimized.

ACTIONS after Editor's considerations:

-We will modify Figure 5 caption to make the point above clearer: "Figure 5. Ratio of average 100-yr Corg accumulation rates (CAR) between simulated and ideal 210Pb profiles produced by various sedimentary processes. (a) seagrass and (b) mangrove/tidal marsh habitats. Error bars represent SE of the regression and SE of the mean using the CF:CS and CRS models, respectively. Ratios of simulated/ideal sedimentation rates (MAR) are equal to those of CAR, determined from multiplying MAR by the fraction of Corg in sediments (Eq. 3), which was considered constant between ideal and simulated profiles. In simulations of increasing sedimentation and organic matter decay, new MAR and CAR were estimated for ideal 210Pb profiles to represent real changes in accumulation, organic matter decay and associated changes in sediment mass with depth."

- We also will add text in the results section where Figure 5 is referenced (page 8, lines 8-10) "The estimated deviations in accumulation rates from those expected under ideal conditions are shown in Figure 5 for seagrass and mangrove/tidal marsh ecosystems. These deviations are driven by variations in MAR estimates caused by anomalies in the 210Pb concentration profiles."

- And in the Methods section (page 6; lines 21-27): "...the Corg accumulation rate (CAR) was estimated through equation 3 assuming average sediment Corg contents of 2.5% and 8% in seagrass and mangrove/tidal marsh, respectively, in both ideal and simulated sediment profiles".

7. Table 4 is too long and repetitive; there must be a way to condense it, since the options for each outcome are the same.

RESPONSE: We agree with the reviewer. Actions after Editor's consideration: We can present the information in Table 4 using a diagram rather than a table. Please, see the figure attached to the author's response.

8. I found the boxes helpful, except for Box 4, which is different from the others and not necessary in my opinion.

RESPONSE: We agree and will remove Box 4 from the current version of the manuscript after Editor's consideration.

9. I understand the logic of including the methods in an appendix – mostly because they are quite long and detailed. But it is important for the reader to understand what the authors are doing. The authors might consider including in the methods a more detailed description than what is there now (but still less detailed than in the appendix).

RESPONSE: Since the manuscript is already long and dense, the addition of a description of each simulation would be repetitive to what is in the appendix. Table 3 summarizes each simulation, while also being included in the Methods section.

ACTIONS after Editor's consideration: We will add further details in section "2.2 Numerical simulations" in the original version of the manuscript (page 6, lines 21-22). Text will be modified to read: "Ideal profiles were then altered to simulate the following processes/scenarios: mixing (surface and deep mixing), increasing sedimentation (by 20%, 50%, 200% and 300%), erosion (recent and past), changes in sediment grain size (coarse and heterogeneous) and OM decay (under anoxic and oxic conditions,

and with labile OM contribution in sediments containing 16.5% and 65% OM). See Table 3 for a summary description of the modelled scenarios and refer to Appendix A for a detailed description of the methodology used to perform each simulation."

10. Section 2.1 doesn't seem like it should be in the methods.

RESPONSE: We agree with the reviewer that most of the information would be best located in the introduction section.

ACTIONS after Editor's consideration: We will move section 2.1 to the introduction as a new section 1.1. 210Pb dating models. The equation and methods to estimate Corg accumulation rates (page 5, lines 24-28), however, will be kept in the Methods section.

11. The authors mention a literature review several times, but the only detail is provided on p. 4 line 27ff. in establishing that CIC, CRS, and CFCS are the most commonly used approaches. Is this the same literature review that was used to construct Figure 2? Please clarify. Also, they probably missed some of the literature by not including the term Pb-210, which is sometimes used instead of 210Pb. (There are almost certainly more than 150 uses of 210Pb in the salt marsh literature.)

RESPONSE: The publications we used to construct Figure 2 are cited in section "3.1 Types of excess 210Pb concentration profiles" (page 6-7) and in the caption of Figure 2. These examples are part of the literature review but more cases could be cited, especially for mixing types II, III and IV in all vegetated coastal ecosystems. We believe that the examples provided are representative of the diversity of 210Pb concentration profiles encountered by researchers. The web of Science search was a simple search meant to identify the dating models generally used in vegetated coastal ecosystems, while showcasing examples of the sedimentary processes driving 210Pb distribution. We agree with the reviewer that we missed some tidal marsh and mangrove studies by not including the term Pb-210 or lead-210. Using the keywords mangrove sediment, salt marsh/saltmarsh/tidal marsh sediment, seagrass sediment AND 210Pb/Pb-210/lead-210 produces 85, 198 and 26 results, respectively for each ecosystem.

ACTIONS after Editor's consideration:

- We will update our search in the Web of Science for all ecosystems also including the term Pb-210 and lead-210.

- In section "3.1 Types of excess 210Pb concentration profiles" we will modify the statement in page 7 lines 32-33 "Our literature review reveals that various sedimentary processes might produce similar types of excess 210Pb concentration profiles" to "These examples identified from the literature reveal that various sedimentary processes might produce similar types of excess 210Pb concentration profiles".

- A clarification will be added in Figure 2 caption: "Figure 2. Sketch of seven sedimentary types of excess 210Pb concentration profiles in sediments from vegetated coastal habitats identified from the literature (see references included) . . .".

12. The reason for excluding the CIC method – the absence of ideal profiles – is not persuasive as currently expressed. The other methods also suffer when there are deviations from the ideal profile, which is exactly what the authors explore. Perhaps more of a justification for excluding CIC could be given?

RESPONSE: In most sediment systems, variations in accumulation rate may occur in response to natural processes or anthropogenic influences. Under some such circumstances, the CRS or CIC models could be suitable, but more often than not, the CRS model is applied as it usually yields more reasonable results than that of the CIC in both fresh and marine environments (Appleby, 2008; Appleby et al., 1983; Breithaupt et al., 2014; Oldfield et al., 1978). The application of the CIC model requires a monotonic decrease in excess 210Pb concentrations with depth that usually occur in lakes but rarely occur in coastal environments. Sediment disturbances like mixing or changes in the sedimentation rate may result in excess 210Pb activities leading to age reversals that prevent the construction of an age model. The CRS model, in contrast, suffers less with non-monotonic features in the 210Pb record and is relatively insensitive to mixing (Appleby and Oldfield, 1992). Because of the general preference and widely application of the CRS model over the CIC model under varying sediment accumulation rates we excluded the CIC model in our simulations.

In some cases, the CIC model could be preferred over CRS model. These are locations where primary sedimentation rate has remained relatively constant (we still observe a monotonic decrease in excess 210Pb specific activities) but there is a hiatus in the sediment record caused by an erosion event, where sediment focusing is a major factor or where there have been major hydrological changes. However, these approaches are site specific and each data set must be evaluated independently for consistency with one or other of the dating models.

ACTIONS after Editor's consideration:

- We will remove the statement in page 5, lines 6-8 that points to CIC as not being appropriate to date vegetated coastal sediments and we will change it for: "The variability of accumulation rates, the progressive loss of matter because of organic matter degradation, and the post-depositional sediment disturbances causing variations in the initial 210Pb concentration have resulted in the CRS model being the most widely used means of deriving 210Pb chronologies".

- Then, text will be added after the description of the CRS model (page 5, lines 16-21) to highlight the situations where the CIC model might be preferred "Although the CRS model often yields more reasonable results than that of the CIC in both fresh and marine environments (Appleby, 2008; Breithaupt et al., 2014; Oldfield et al., 1978; Sanders et al., 2016), the CIC model might be preferred at locations where sediment focusing is a major factor, significant hydrologic changes have occurred or there are hiatus in the sediment record caused by erosion events (Appleby, 2008). These approaches are site specific and each data set must be evaluated independently for consistency with one or other of the dating models, and therefore the CIC model has been excluded from the simulations presented in this study."

REFERENCES:

- Appleby, P. G.: Three decades of dating recent sediments by fallout radionuclides: a review, The Holocene, 18, 83–93, doi:10.1177/0959683607085598, 2008.

- Appleby, P. G. and Oldfield, F.: Applications of lead-210 to sedimentation studies, in Uranium-series disequilibrium: applications to earth, marine, and environmental sciences, edited by M. Ivanovich and R. Harman, Clarendon Press, Oxford., 1992.

- Appleby, P. G., Oldfield, F. and Physics, T.: The assessment of 210Pb data from sites with varying sediment accumulation rates, Hydrobiologia, 103, 29–35, doi:10.1007/BF00028424, 1983.

- Breithaupt, J. L., Smoak, J. M., Smith, T. J., Sanders, C. J., Smoak, J. M., Smith, T. J. and Sanders, C. J.: Temporal variability of carbon and nutrient burial, sediment accretion, and mass accumulation over the past century in a carbonate platform mangrove forest of the Florida Everglades, J. Geophys. Res. Biogeosciences, 119, 2032–2048, doi:10.1002/2014JG002715, 2014.

- Oldfield, F., Appleby, P. G. and Battarbee, R. W.: Alternative 210Pb dating: results from the New Guinea Highlands and Lough Erne, Nature, 271(5643), 339–342, doi:10.1038/271339a0, 1978.

13. I'm not sure the distinction between Types VI and VII is necessary. They are both characterized by low inventories, regardless of profile shape.

RESPONSE: We do not agree, type VI show an extreme situation with almost negligible excess 210Pb concentrations that in most of the cases will be undatable. Type VII, because of sediment erosion, might be undatable too, but some researchers may not consider erosion and date it anyhow. We would like to keep the distinction of the two profile types in the revised version of the manuscript after Editor's consideration.

RESPONSE TO COMMENTS BY ANONYMOUS REFEREE #4

The review paper presented by Arias-Ortiz discuss the use of the 210Pb dating technique to estimate the rate of mass accumulation in vegetated coastal ecosystems.

Such information is indeed very important in considering the significant role of vege-tated coastal habitats (tidal marsh, mangrove, seagrass) as sinks of carbon. Over the last 150 years, 210Pb is the only tool that permits to calculate sediment and carbon accumulation rates (SAR/CAR) in such environments. However, the application of the 210Pb-based method is not tricky in these environments. The authors aim to illustrate the models usually applied to calculate SAR or MAR in these setting. This article is extremely timely as there is a growing interest in better estimate C source/sink. The authors are presenting in a correct way the principle and the conditions of the 210Pb method. Although the article is mostly dedicated to the models, there are some rec-ommendation on the 210Pb determination and a comment of the interest of additional time marker (like 137Cs) or normalisation.

We sincerely thank the reviewer for acknowledging the interest of our work as well as for his/her comments, which were very helpful in improving the paper.

1.In fact I regret that the authors do not develop the experimental section. Indeed, it would be of great interest to provide recommendations about sampling: core descrip-tion, porosity determination etc.

RESPONSE to comment 1 and 4: Since questions 1 and 4 of reviewer #4 target the same issue (i.e. development of an experimental section prior to 210Pb analyses), we addressed them together below.

Discussing sampling and sample-handling is not a simple task that should include plenty of aspects if it is done properly. For instance, estimation of porosity, dry bulk density, which types of corers to use, how to extrude or slice the sediment, preserva-tion of the interface or a discussion of the analytical methods. Some available manu-als/chapters dealing with all these aspects already exist, such as Brenner and Kenney, (2013) or IAEA-TECDOC-1360, (2003), and we will cite them in the revised version of the manuscript to provide the reader with additional guidelines for coring, sampling and sample-handling. However, developing the above-mentioned aspects goes beyond the

scope of the manuscript. We, however, agree in adding some guidelines to consider the type of sample, which is related to reviewer's comment 4.

ACTIONS after Editor's consideration: We will modify the text in section 4. Approaches and Guidelines (page 12, lines 1-15) to briefly develop some basic sampling and sample-handling procedures to achieve good 210Pb profiles. We also will provide some references that the reader could use to expand on the topic such as Brenner and Kenney (2013) and IAEA-TECDOC-1360 (2003). The suggested text will read as:

"Researchers can have control over some factors such as coring, sampling, or sample-handling, that can create artifacts in 20Pb profiles and contribute to dating error. Guidelines for core sampling for the analysis of 210Pb and other radionuclides have been described in detail, for example, in Brenner and Kenney (2013) and in IAEA technical report IAEA-TECDOC-1360 (2003). Some knowledge on the expected sedimentation rate is useful to decide how to section a sediment core for 210Pb measurements, as well as the length that a core must have to reach the excess 210Pb horizon. Core sectioning should be planned such appropriate resolution in terms of 210Pb dating is achieved, reconciling what is technically feasible while securing enough material to conduct analyses of 210Pb and other parameters (e.g., grain size, other radionuclides, metals, organic matter or nutrients) (IAEA-TECDOC-1360, 2003). Low sedimentation rates (âĹij1-2 mm yr-1) mean that the entire excess 210Pb inventory will be captured in the uppermost 20 cm of the sediment. In such situation, fine sectioning intervals (0.5 - 1 cm) are required. Longer cores (of about 100 cm) should be collected if high sedimentation rates are expected (several mm yr-1) and these can be sliced at thicker intervals, without compromising the temporal resolution of the 210Pb record. If sedimentation rates are not known a priori, it is best to choose fine sampling intervals (e.g, at 0.5 cm along the upper 20 cm and at 1 cm below 20 cm) to ensure sufficient resolution. After collection, a visual description (e.g., colour, texture, presence of roots, organisms or layers) of the sediments contained in the corer and measurement of parameters such as water content, organic matter or grain size are relatively low-cost actions that

provide information to interpret 210Pb distribution. Indeed, the type of sediment (e.g., fine vs. coarse, rich in carbonates, homogeneous or with organic debris embedded) is a factor that should be considered (IAEA-TECDOC-1360, 2003). Reliable sedimentation histories are difficult to obtain in vegetated coastal sediments consisting of coarse particles or coarse-grained carbonates where excess 210Pb is less preferentially adsorbed (Wan et al., 1993). In such situations, the analysis of 210Pb in the smaller sediment fraction (i.e. < 63um or < 125 um) is recommended to concentrate 210Pb and reduce the dilution effect caused by coarse fractions. This methodology has been applied in mangrove ecosystems from arid regions (Almahasheer et al., 2017) where excess 210Pb flux is very low, and in Florida Bay carbonate-rich seagrass sediments (Holmes et al., 2001). Similarly, large organic material such as roots and leaves should be removed from the sediment samples prior to 210Pb analyses as these will contribute to the dilution of the excess 210Pb specific activities."

REFERENCES:

- Brenner, M. and Kenney, W. F.: Dating Wetland Sediment Cores, in Methods in Biogeochemistry of Wetlands, edited by R. D. DeLaune, K. R. Reddy, C. J. Richardson, and J. P. Megonigal, pp. 879–900, Soil Science Society of America, Madison., 2013.

- IAEA-TECDOC-1360: Collection and preparation of bottom sediment samples for analysis of radionuclides and trace elements., International Atomic Energy Agency, IAEA, Vienna., 2003.

- Almahasheer, H., Serrano, O., Duarte, C. M., Arias-Ortiz, A., Masque, P. and Irigoien, X.: Low Carbon sink capacity of Red Sea mangroves, Sci. Rep., 7(1), 9700, doi:10.1038/s41598-017-10424-9, 2017.

- Holmes, C. W., Robbins, J., Halley, R., Bothner, M., Brink, M. Ten and Marot, M.: Sediment dynamics of Florida Bay mud banks on decadal time scale, Bull. Am. Paleontol., 361, 31–40, 2001.

- Wan, G. J., Liu, J. and Li, B.: The Isotopic Character and the Remobilization of Lead at the Top of Sediment in Erhai, Chinese Sci. Bull., 38(2), 139–142, doi:10.1360/sb1993-38-2-139, 1993.

2.It is also important to precise more clearly the advantage of gamma counting compared to alpha counting. In addition to avoid chemistry step, gamma spectrometry has the major advantage to determined simultaneously 210Pb and its supported parent (226Ra), 137Cs, 228Th, 7Be, 40K among others.

RESPONSE: Both gamma and alpha counting have pros and cons. Gamma counting avoids the radiochemical step, allows determining the concentrations of various radionuclides simultaneously and sample preparation is nondestructive. However, it has higher limits of detection compared to alpha spectrometry, requires relatively large amount of sample and requires correction for self-adsorption at low energies (i.e. for Pb-210). Indeed, the efficiency calibration is not straightforward. In addition, gamma detectors are costly compared to alpha detectors, and this fact can limit the number of detectors a laboratory can have and thus the sample throughput. Most often, the analysis employed is dependent on the instrument availability of the laboratory, therefore in our manuscript we did not provide with details about the measuring techniques of 210Pb. However, in the revised version of the manuscript, we will add some text about the advantage of using gamma spectrometry in the determination of 226Ra, as the referee suggested in comment 12, below.

ACTIONS after Editor's consideration:

- We will state more clearly where to find the information about measuring approaches in the revised manuscript. Text that is in page 14, lines 11-12 in the current version will be modified to read: "(for a detailed description of the laboratory analysis of these radioisotopes and advantages and disadvantages of each method see Corbett and Walsh (2015) and Goldstein and Stirling (2003))".

- Then we will highlight the advantage of gamma compared to alpha spectroscopy in

section 4.4. 226Ra concentration profiles (page 15, lines 4-6) of the current version of the manuscript, text will be modified to read: "Excess 210Pb concentrations are determined by subtracting supported 210Pb to total 210Pb concentrations assuming it is in equilibrium with 226Ra. This is straightforward when gamma spectrometry is employed since the total 210Pb and supported 210Pb (i.e., 226Ra) can be quantified simultaneously. In occasions, particularly when 210Pb is determined by alpha spectrometry, 226Ra is not measured, and supported 210Pb is most often determined from the region of constant and low 210Pb concentrations at depth, or alternatively, from a number of determinations of 226Ra via gamma spectrometry or liquid scintillation counting (LSC)".

REFERENCES: - Corbett, D. R. and Walsh, J. P.: 210Lead and 137Cesium: establishing a chronology for the last century, in Handbook of Sea-Level Research, edited by I. Shennan, A. J. Long, and B. P. Horton, pp. 361–372, John Wiley & Sons, Ltd., 2015.

- Goldstein, S. J. and Stirling, C. H.: Techniques for measuring uranium-series nuclides: 1992-2002, Rev. Mineral. Geochemistry, 52, 23–57, doi:10.2113/0520023, 2003.

3. I am surprised that the authors mentioned 228Th as a potential dating/bioturbation tracer. In such coastal environment, I usually use 228Th as 232Th its grandfather to trace the detrital fraction. It is a good way to normalize also radionuclide activities.

RESPONSE: The use of 228Th as indicated by the reviewer is a possibility, indeed. But then, excess 228Th has been also used to determine sedimentation and mixing rates in coastal sediments (Hancock and Hunter, 1999; Huh et al., 1987). In vegetated coastal ecosystems some researchers have used excess 228Th to estimate fast rates of particle deposition in mangroves (e.g., Alongi et al., 2005) or, together with 7Be and 234Th, mixing (Sharma et al., 1987; Smoak and Patchineelam, 1999). In section "4.6 Normalization of excess 210Pb concentrations and sieving of sediments" of the original version of the manuscript we suggest the normalization of 210Pb profiles to organic matter content, grain size, or aluminum that traces the lithogenic fraction as

well, as these are the most common variables used normalize 210Pb profiles in the literature.

ACTIONS after Editor's consideration: we will add text in section 4.1 Short-lived radionuclides (234Th, 228Th, 7Be) (page 12) to explain briefly the origin of excess 228Th. "A constrain to the use of excess 228Th is that sediments must contain a lithogenic/detrital fraction, as is mostly the case in vegetated coastal sediments."

REFERENCES:

- Alongi, D. M., Pfitzner, J., Trott, L. a., Tirendi, F., Dixon, P. and Klumpp, D. W.: Rapid sediment accumulation and microbial mineralization in forests of the mangrove Kandelia candel in the Jiulongjiang Estuary, China, Estuar. Coast. Shelf Sci., 63, 605–618, doi:10.1016/j.ecss.2005.01.004, 2005.

- Hancock, G. J. and Hunter, J. R.: Use of excess 210Pb and 228Th to estimate rates of sediment accumulation and bioturbation in Port Phillip Bay, Australia, Mar. Freshw. Res., 50(6), 533, doi:10.1071/MF98053, 1999.

- Huh, C.-A., Zahnle, D. L., Small, L. F. and Noshkin, V. E.: Budgets and behaviors of uranium and thorium series isotopes in Santa Monica Basin sediments, Geochim. Cosmochim. Acta, 51(6), 1743–1754, doi:10.1016/0016-7037(87)90352-8, 1987.

- Sharma, P., Gardner, L. R., Moore, W. S. and Bollinger, M. S.: Sedimentation and bioturbation in a salt marsh as revealed by 210Pb, 137Cs, and 7Be studies, Limnol. Oceanogr., 32(2), 313–326, doi:10.4319/lo.1987.32.2.0313, 1987.

- Smoak, J. M. and Patchineelam, S. R.: Sediment mixing and accumulation in a mangrove ecosystem : evidence from 210Pb , 234Th and 7Be, Mangroves Salt Marshes, 3, 17, 1999.

4. I think it is also important to point out the need to well consider the samples. In the case of sediments presenting coarse fraction or vegetal debris, it could be useful to separate the fine sediment fraction, that supports 210Pb, from the other fractions (that

dilute its activity). In fact it is the first step to do: how to obtain the best 210Pbxs profile depending of the sediment. It could help to reduce variability in the 210Pbxs profile. The authors need to develop this aspect. In fact I am convinced that some model adaptions are not required if sampling and measurements are done in an appropriate way (see figure panel D why measure with the sandy fraction).

RESPONSE: We agree in adding some guidelines to consider the type of sample, since this was also raised in comment 1, we paste here the relevant text

ACTIONS after Editor's consideration: We will modify the text in section 4. Approaches and Guidelines (page 12, lines 1-15) to briefly develop some basic sampling and sample-handling procedures to achieve good 210Pb profiles. The suggested text will read as: "After collection, a visual description (e.g., colour, texture, presence of roots, organisms or layers) of the sediments contained in the corer and measurement of parameters such as water content, organic matter or grain size are relatively low-cost actions that provide information to interpret 210Pb distribution. Indeed, the type of sediment (e.g., fine vs. coarse, rich in carbonates, homogeneous or with organic debris embedded) is a factor that should be considered (IAEA-TECDOC-1360, 2003). Reliable sedimentation histories are difficult to obtain in vegetated coastal sediments consisting of coarse particles or coarse-grained carbonates where excess 210Pb is less preferentially adsorbed (Wan et al., 1993). In such situations, the analysis of 210Pb in the smaller sediment fraction (i.e. < 63um or < 125 um) is recommended to concentrate 210Pb and reduce the dilution effect caused by coarse fractions. This methodology has been applied in mangrove ecosystems from arid regions (Almahasheer et al., 2017) where excess 210Pb flux is very low, and in Florida Bay carbonate-rich seagrass sediments (Holmes et al., 2001). Similarly, large organic material such as roots and leaves should be removed from the sediment samples prior to 210Pb analyses as these will contribute to the dilution of the excess 210Pb specific activities."

Other comments: 5.the authors need to check the manuscript in order to verify the terms and acronyms (like Db and not D for bioturbation).

RESPONSE: The following acronyms will be revised and unified throughout the manuscript: D for Db: Bioturbation Corg-MAR for CAR: Carbon accumulation rate

6.Page 2 line 24: "210Pb is not affected by interannual variability": to moderate 210Pbxs fluxes could have some variability although moderate

RESPONSE: text will be corrected as suggested - "210Pb flux is moderately affected by interannual variability".

7.Page 3 line 3: and subsequent fallout

RESPONSE: Text will be added as suggested.

8.Page 4 (and in all the text): be careful to use correctly concentration and activity

RESPONSE: We agree with the reviewer that some of these terms were not correctly used throughout. Concentration is equally used as specific activity throughout the manuscript, these both refer to activity per unit of mass, while the single term activity should be used when referred to decays per unit of time. Inventory refers to activity per unit of area.

ACTIONS after Editor's consideration: We will revise all the entries for activity, specific activity and concentration in the current version of the manuscript and check for its correct use based on the above.

9.CIC model/ I disagree with the statement CIC is not appropriate. This model could be useful in some sediment core presenting event-deposit (like flood). Such deposits could be sand, but also fine sediments that could present lower 210Pbxs (compared to surrounding layers). In fact, CIC could be useful to check dating when it is difficult to precisely define the thickness of such deposits.

RESPONSE: We agree with the reviewer; indeed, the CIC model can be more appropriate than the CRS model and useful in situations where there is a hiatus in the sediment record caused by an erosion event, there are significant hydrologic changes

or sediment focusing is a major factor.

ACTIONS after Editor's consideration:

- We will remove the statement in page 5, lines 6-8 in the current version of the manuscript that points to the CIC model as not being appropriate to date vegetated coastal sediments and we will change it for: "The variability of accumulation rates, the progressive loss of matter because of organic matter degradation, and the sediment disturbances causing variations in the initial 210Pb concentration have resulted in the CRS model being the most widely used means of deriving 210Pb chronologies".

- Then, text will be added after the description of the CRS model (page 5, lines 16-21) to highlight the situations where the CIC model might be preferred "Although the CRS model often yields more reasonable results than that of the CIC in both fresh and marine environments (Appleby, 2008; Breithaupt et al., 2014; Oldfield et al., 1978; Sanders et al., 2016), the CIC model might be preferred at locations where sediment focusing is a major factor, significant hydrologic changes have occurred or there are hiatus in the sediment record caused by erosion events (Appleby, 2008)."

10. page 7 type II: lower activities could be also explained by dilution by roots for example, so it is important as indicated previously to provide recommendations for sampling.

RESPONSE: Large organic debris like roots should be removed from the sediment samples prior the sample preparation for alpha or gamma spectrometry analyses. This aspect will be included in the revised version of the manuscript in section 4. Approaches and Guidelines. Since this was also raised in comment 1 and 4, we paste here the relevant text:

ACTIONS after Editor's consideration: We will modify the text in section "4. Approaches and Guidelines" (page 12, lines 1-15) to briefly develop some basic sampling and sample-handling procedures to achieve good 210Pb profiles. The suggested text will

read as: " Similarly, large organic material such as roots and leaves should be removed from the sediment samples prior to 210Pb analyses as these will contribute to the dilution of the excess 210Pb specific activities."

11. 13 line 30-34: the presence of large OC concentration or vegetal (like leaves) could promote high concentration of Cs due to mobility. So care is required with 137Cs.

RESPONSE: We agree that, in addition to the mobility of 137Cs due to the reasons we indicated in the manuscript, 137Cs concentration profiles can also be affected by the presence of organic matter, and can be accumulated in leaf litter and living roots, this will be included in the revised manuscript.

ACTIONS after Editor's consideration: text will be added to also take into consideration this aspect in page 4 lines 1-3: "High contents of organic matter can also affect the distribution of 137Cs in sediments as it is preferentially accumulated in leaf litter and may be absorbed by living roots (Staunton et al., 2002)."

References:

- S. Staunton, Camille Dumat, A. Zsolnay. Possible role of organic matter in radiocaesium adsorption in soils. 2002 Journal of Environmental Radioactivity, 58, 163-173.

12. page 15 line5-7: not clear, it seems there is a confusion between alpha (that requires to assume the rather constant 210Pb activities correspond to the supported 210Pb) and gamma (that determines both 210Pb and 226Ra)).

RESPONSE: This will be clarified in the revised version of the manuscript.

ACTIONS after Editor's consideration: Text in page 15 line 5-7 will be modified as: "Excess 210Pb concentrations are determined by subtracting supported 210Pb to total 210Pb concentrations assuming it is in equilibrium with 226Ra. This is straightforward when gamma spectrometry is employed since the total 210Pb and supported 210Pb (i.e., 226Ra) can be quantified simultaneously. In occasions, particularly when 210Pb is determined by alpha spectrometry, 226Ra is not measured, and supported 210Pb

is most often determined from the region of constant and low 210Pb concentrations at depth, or alternatively, from a number of determinations of 226Ra via gamma spectrometry or liquid scintillation counting (LSC) along the core"

[Figure]

[Figure]

**Fig. 1.** Table 4: new diagram

---

## Author Response (AR1)

Perth, WA, 13th of September 2018

Dear Dr. Sarin,

Below, please find our response to the comments raised by the reviewers to the manuscript entitled "Reviews and syntheses: [210]Pb-derived sediment and carbon accumulation rates in vegetated coastal ecosystems: setting the record straight", along with a description of the changes included in the revised version of the manuscript. The revised version of the manuscript is attached after the response letter. We are very grateful to the reviewers for their thoughtful and constructive comments and below we address each of the points they raised.

Ariane Arias-Ortiz on behalf of the authors

Response to comments by Anonymous Referee #3

General comments: Overall, a very valuable contribution to the literature. This is a helpful synthesis of the literature that will be a go-to for those in the field, and it is also an interesting modeling exercise that sheds light on the processes producing various [210]Pb patterns. My main concern is that the manuscript provides an overly optimistic view of the errors associated with complex [210]Pb profiles, for reasons explained below.

We sincerely thank the reviewer for acknowledging the interest of our work as well as for his/her constructive comments, which were very helpful in improving the manuscript. We will include in the revised manuscript a more throughout discussion showing the implications of estimating accumulation rates in complex [210]Pb profiles.

1. Specific comments: As the authors note in Table 4, patterns II, III, and IV can have multiple causes. Especially common, and especially problematic, is the difficulty in distinguishing between mixing and an increase in MAR. The simulation studies in this paper don't address this adequately because they separate the mixing simulations from the increased sedimentation simulations. For the mixing simulations, for example, "the CF:CS model was applied below the depth of the visually apparent SML (3 cm) in scenarios A and B to avoid overestimation of MAR" (Appendix). But if you didn't know this profile was created by mixing, how would you know that you would be overestimating MAR rather than accurately estimating an increase in MAR? In other words, in the real world, how would you know whether it was mixing (so leave out the SML) or increased MAR? True, the mixing and increased sedimentation profiles in Figure 3 do look somewhat different, but I am not convinced that in the real world they are so easily distinguished.
Bottom line: I am concerned that if the authors tested the error in non-ideal profiles without knowing what caused them, they would find higher errors than those shown in Figure 5.

Response: We agree that mixing and increasing mass accumulation rate processes are difficult to distinguish from the [210]Pb concentration profile, however there are some complementary analyses that can help to distinguish each of these processes. Such actions/analyses are explained in section 4.2 of the revised version of the manuscript (pages 16-18). We think it is interesting to show to the reader the consequences of mismatching the process and how they translate in higher errors in sediment and $C_{org}$ accumulation rates (CAR).

Actions: In the revised version of the manuscript we have run the [210]Pb models in mixing simulations assuming that the observed anomalies were caused by increased mass accumulation rates and vice versa in increasing sedimentation simulations. In the case of assuming mixing in increasing MAR simulations, mean accumulation rates were underestimated between 10 and 30% in both habitats using the CF:CS model. Assuming increasing MAR in recent years in mixing simulations, this yield an overestimation of the accumulation rate ranging between 20 and 95%

in seagrass sediments and between 3 and 30% in mangrove/tidal marsh sediments using the CF:CS model. A process mismatch between mixing and increased MAR in recent years did not cause large deviations (between 2 and 7%) in accumulation rates derived by the CRS model. The CRS model is run similarly if mixing or changes in accumulation rates are expected albeit ages within the mixed layer cannot be reported if mixing occurs.

We have added these results in section *3.2.1 Mixing* (page 9) and section *3.2.2 Increasing sedimentation rates* (page 10). Additionally, text was added in section *3.2.6 General remarks* (page 13) discussing the resulting MAR and CAR if the incorrect process is assumed and dating models are applied.

Text added in lines 19-27; page 13: *"However, failure to account for the correct process affecting $^{210}$Pb concentration profiles could lead to deviations in mean MAR and CAR exceeding 20% (Fig. 5c, d).*

*MAR and CAR were most overestimated, from 20 to 95% at simulations with low accumulation rates when acceleration was interpreted in mixed excess $^{210}$Pb profiles and the CF:CS model was applied piecewise. Deep mixing confounded with an increase in MAR generated the largest overestimation of mean CAR in both habitat types. On the contrary if mixing was assumed in excess $^{210}$Pb profiles showing a recent increase in MAR, mean accumulation rates were underestimated by up to a 30% using the CF:CS model below the "surface mixed layer".*

CAR results of incorrect process interpretation have also been plotted in Figure 5b and 5d and we have reorganized section 4 according to the processes simulated in section 3 (mixing, increasing MAR, erosion, changes in gran size and OM decay). Thus, it is straightforward for the reader to pick actions to identify each of these processes. Section 4 has been restructured as follows:

4. Approaches and guidelines
4.1 General validation of $^{210}$Pb models
- Artificial radionuclides
- Geochemical information of sediments
4.2 Mixing or Rapid sedimentation
- Geophysical analyses (i.e, X-ray radiographies, CAT scans)
- Short-lived radionuclides
- Maximum penetration depth of excess $^{210}$Pb
4.3 Erosion
- Excess $^{210}$Pb inventories in reference and disturbed sites
4.4 Heterogeneous sediment composition
- Normalization of excess $^{210}$Pb profiles
- $^{226}$Ra profiles

2. Related to the above: the authors choose not to create a CRS estimate for the profiles with erosion. That is fine as long as one knows that erosion is a factor. In the real world, minor deviations from the ideal inventory (especially the small ones shown in the tidal marsh half of Figure3c) do not generally preclude investigators from applying the CRS method. I would strongly encourage the authors to apply CRS to these profiles to get a sense of how large the associated errors are. At a minimum, they should caution others not to use the CRS method with profiles that show deviations from the expected inventory.

Response: We agree, and we have also applied the CRS model to the simulated eroded profiles, as a result accumulation rates were underestimated by 25% in seagrass and by 10% in mangrove/tidal marsh sediments. Differences between habitats are explained by the different proportion of the eroded excess $^{210}$Pb inventory. Because seagrass ecosystems have lower sedimentation rates, a greater proportion of the excess $^{210}$Pb inventory was comprised in the top

10 cm of the sediment column and thus eroded. We have plotted the results in Figure 5b and 5d. Text has been added in section *3.2.3 Erosion* (page 10-11) describing the outputs of the CRS model.

Text added (page 10, lines 29-32; page 11, lines 1-2): *"The CRS model cannot be applied to eroded excess $^{210}$Pb profiles unless the missing inventory is known and the total (I) and depth-specific ($A_m$) excess $^{210}$Pb inventories can be corrected. Assuming erosion was not a factor, the application of the CRS model to our simulated profiles underestimated MAR and CAR by up to 25% in seagrass and by 10% in mangrove/tidal marsh sediments (Fig 5b and 5d). Therefore, we caution against the use of the CRS model in profiles that show deviations from the expected inventory, such as those simulated for seagrass sediments here (Fig. 3c)."*

Additionally, in section *3.2.6 General remarks* (page 13), text has been added to emphasize those problems associated to the application of the CRS model in incomplete sediment records, and the bias in calculated MAR and CAR.

Text added (page 13, lines 27-34): *"Indeed, the CRS model was less sensitive to anomalies in excess $^{210}$Pb concentration profiles, however, its application requires accurate determination of the excess $^{210}$Pb inventory at each depth ($A_m$) and in the entire record (I), which can be problematic, for instance when all samples along a sediment core have not been analysed or when sediment erosion has occurred at the core location. When the total excess $^{210}$Pb inventory is underestimated, be it through erosion, poor detection limits or insufficient core length, this generates erroneous dates and underestimation of average MAR and CAR. Underestimation of accumulation rates will depend largely on the proportion of the missing fraction of the excess $^{210}$Pb inventory from $A_m$ and I. In our simulations, MAR and CAR were underestimated by 10 to 25%."*

3. The authors use their results to suggest in Figure 5 and Table 4 that pretty much any 210Pb profile is dateable (except those with extreme OM concentrations). However, in the real world, some profiles are likely to be altered in more complex ways than the simulations shown here – by mixing and erosion and different grain sizes. I believe that some profiles may just be too altered to be retrievable, and would suggest using extreme caution in interpreting Types V, VI, and VII. Section 4 of the paper is very helpful in suggesting alternative approaches that can help disentangle various factors, but it is in tension with Figure 5 (and the abstract), which suggest that those are not necessary, since maximum error is only 20% anyway.

Response: This is correct, and we probably failed to capture this point in the original version of the manuscript. Although some research reports extremely altered sediment profiles, these are few since a literature bias exists towards those profiles where dating or MAR estimates could be achieved. However, as the reviewer comments, more often than not, some profiles are likely to be altered by a composite of processes, leading to types V and VI. This is especially true in seagrass ecosystems that present lower sedimentation rates and can occur in sand-dominated substrates, where $^{210}$Pb is less preferentially adsorbed. For instance, Saderne et al. (2018) collected 9 and 11 sediment cores in seagrass and mangroves of the Red Sea, respectively, but none of the seagrass and only 4 of the mangrove sediment cores were useful for the determination of MAR and CAR. In the revised version of the manuscript we have captured this point raised by the reviewer through 3 main actions (see below).

We would like to mention that we have re-assessed OM decay simulations and now the revised version of the manuscript contain both the results for the deviations in mean MAR and CAR if rates are compared with 1) ideal profiles with no decomposition of OM, and 2) ideal profiles that take into account the loss of sediment material with depth (page 12, lines 18-33; page 13, lines 1-4).

Actions:

1. We modified the deep mixing simulation (scenario C) to better represent profile type V. In the new simulation scenario C mixing influences the upper 15 cm, which is a depth reported as deep mixing in seagrass (Serrano et al., 2016), mangroves and tidal marshes (Nittrouer et al., 1979; Smoak and Patchineelam, 1999) and is characteristic for marine sediments globally (Boudreau, 1994). Both results, assuming the process causing this anomaly in the $^{210}$Pb profile is mixing or fast accumulation rate, were plotted in Figure 5. This change indeed increased the errors associated to the estimation of mean MAR and CAR up to 80% in seagrass sediments and up to 30% in mangrove sediments, which have a higher accumulation rate, hence a lower proportion of their entire excess $^{210}$Pb profiles were affected by mixing. Text in the abstract of the current version of the manuscript has been modified accordingly.

   Text in the abstract now reads (page 1; lines 31-35): *"Our results show that the deviations in sediment and derived $C_{org}$ accumulation rates relative to those estimated at undisturbed profiles are within 20% if the process causing the anomalies in $^{210}$Pb profiles is well understood. While these uncertainties might be acceptable for the determination of mean sediment and $C_{org}$ accumulation rates over the last century, they may not always allow the determination of a credible geochronology, or historical reconstruction. Calculations of accumulation rates, however, might be difficult or impossible at sites with slow accumulation rates and intense mixing, and errors in the identification of the processes responsible may lead to deviations of up to 30 to 100%."*

2. Text was included in the Conclusions recommending critical evaluation of the data and acknowledging when a profile is not datable. Text reads (page 20; lines 29-31; page 21 lines 1-11): *"Simulated irregular $^{210}$Pb profiles in this study show that the deviations, relative to ideal undisturbed $^{210}$Pb profiles, in MAR and CAR are within 20% if a correct diagnosis of the intervening sedimentary processes is made. Otherwise, deviations may range between 20% and 100%, with higher errors associated with the application of CF:CS model. Additional tracers or geochemical, ecological or historical data can be used to identify the process causing anomalies in excess $^{210}$Pb profiles and reduce uncertainties in derived accumulation rates. Model choice is another important factor that should be considered to reduce deviations in CAR. Using the procedures in section 4, researchers have been able to obtain credible chronologies in vegetated coastal sediments and reliable mean CAR. This, however, might be particularly challenging in seagrass sediments because of their relatively low sedimentation rates and high sand content, where $^{210}$Pb is less adsorbed because of the low specific surface area of sands. Special caution should be applied in those sites where sediments might be altered by multiple processes (leading to profile types V or VI shown in this study) and where other chronological tools or time markers are not available (e.g., $^{137}$Cs). Sites that have slow accumulation rates and/or intense mixing may unlikely be datable and derived CAR estimates may be largely overestimated. Mistakes would include assigning discrete ages in mixed sediments or extrapolating an age-depth model for a core that should be considered undatable to depths down the core or to nearby sites."*

3. In Figure 6 (former Table 4), a new recommended action has been added for profile types V, VI provided the other recommended actions fail: *"the profile is likely undatable by $^{210}$Pb, if other chronological tools are unavailable or if it is in its majority affected by mixing".*

References:
- Boudreau, B. P.: Is burial velocity a master parameter for bioturbation?, Geochim. Cosmochim. Acta, 58(4), 1243–1249, doi:10.1016/0016-7037(94)90378-6, 1994.
- Nittrouer, C. A., Sternberg, R. W., Carpenter, R. and Bennett, J. T.: The use of Pb-210 geochronology as a sedimentological tool: Application to the Washington continental shelf, Mar. Geol., 31(3–4), 297–316, doi:10.1016/0025-3227(79)90039-2, 1979.

- Saderne, V., Cusack, M., Almahasheer, H., Serrano, O., Masqué, P., Arias-Ortiz, A., ... & Duarte, C. M. (2018). Accumulation of carbonates contributes to coastal vegetated ecosystems keeping pace with sea level rise in an arid region (Arabian Peninsula). Journal of Geophysical Research: Biogeosciences, 134, 1498-1510.
- Serrano, O., Ruhon, R., Lavery, P. S., Kendrick, G. A., Hickey, S., Masqué, P., Arias-Ortiz, A., Steven, A. and Duarte, C. M.: Impact of mooring activities on carbon stocks in seagrass meadows, Sci. Rep., 6, 23193, doi:10.1038/srep23193, 2016.
- Smoak, J. M. and Patchineelam, S. R.: Sediment mixing and accumulation in a mangrove ecosystem: evidence from 210Pb, 234Th and 7Be, Mangroves Salt Marshes, 3, 17–27. Doi:10.1023/A:1009979631884, 1999.

4. It would be helpful if the Supplementary Tables in Excel had formulas rather than just values, to make it easier to understand how the simulations were done.

Response: We agree and we have added formulas in the supplementary Tables.

5. I think the authors could emphasize more strongly that they are looking at the 100-year average MAR and Corg-MAR, not the patterns over time. For example, the y-axis in Figure 5 (or at least the figure caption) could say "100-year Corg burial."

Response: We agree with the reviewer and in the revised version of the manuscript we have made this point clearer, not only in Figure 5 but also in the Methods and Results sections of the revised version of the manuscript.

Actions:
- In the Methods section *"2.1 Numerical simulation"* text has been added (page 7, line 5-6) *"The CF:CS and CRS dating models were applied to the simulated excess $^{210}$Pb profiles to determine the average MAR for the last century."*
- In the Results section *"3.2 Simulated sediment and $C_{org}$ accumulation rates (MAR and CAR)"* text has been added (page 9, lines 8) *"We estimated mean 100-yr MAR and CAR for the simulated profiles by applying the CF:CS and CRS models, and results were compared with those from their respective ideal non-disturbed $^{210}$Pb profiles."*
- In the Results section *"3.2.6. General remarks"* text has been added (page 13; line 7-8): *"Among the various ecosystems considered here, average last 100-yr MAR and CAR derived from both the CF:CS and the CRS models were less vulnerable to anomalies in mangrove/tidal marsh compared to seagrass sediments".*
- In Figure 5 (page 41), the figure caption has been modified to read as *"Ratio of average 100-yr $C_{org}$ accumulation rates (CAR) between simulated and ideal $^{210}$Pb profiles produced by various sedimentary processes in seagrass (a,b) and mangrove/tidal marsh habitats (c,b)."*

6. Does this analysis only apply to $C_{org}$ burial? There will be an audience interested in the equivalent of Figure 5 for the MAR itself, which presumably would be easy to make.

Response: The ratio between ideal vs. disturbed CAR (Fig. 5) mostly represents variations in MAR, therefore Figure 5 would look similar for MAR ratios between ideal and disturbed profiles. This was explained in the Figure 5 caption in the original version of the manuscript, but we acknowledge this was not clear enough. Therefore, we have modified the Figure 5 caption and title on Y axis to make the point clearer, and text has also been added in the Methods and Results section where Figure 5 is referenced.

For our simulations the $C_{org}$ content was considered to be the same in both the disturbed and the ideal excess [210]Pb profiles, meaning that the mixed sediments or the newly deposited ones had same $C_{org}$ (%DW) as those in the ideal non-disturbed profile (2.5% in seagrass sediments and 8% in mangrove/tidal marsh sediments). While any disturbance of the sedimentary record would also affect $C_{org}$ concentrations due to changes in biogeochemical processes within sediments, the potential and magnitude of such effects is unclear, and therefore, they were not considered here. In addition, the $C_{org}$ content is a parameter readily measurable in sediments hence should not lead to errors in the estimation of $C_{org}$ accumulation rates. The aim of the manuscript is to estimate how errors in the estimation of MAR using [210]Pb would affect resulting CAR rates and how these errors can be minimized.

Actions: We have modified the Figure 5 caption (page 41, lines 8-12): *"Figure 5...Ratios of simulated/ideal sedimentation rates (MAR) are equal to those of CAR, determined from multiplying MAR by the fraction of $C_{org}$ in sediments (Eq. 5), which was considered constant between ideal and simulated profiles. In simulations of increasing sedimentation and organic matter decay, new MAR and CAR were estimated for ideal [210]Pb profiles to represent real changes in accumulation, organic matter decay and associated changes in sediment mass with depth."*

Title of Y axis in Figure 5 now reads: *Simulated MAR-CAR: Ideal MAR-CAR*

Text has been added in the Methods section (page 7; lines 14-20): *"...the $C_{org}$ accumulation rate (CAR) was estimated through equation 5 assuming average sediment $C_{org}$ contents of 2.5% in seagrass and 8% in mangrove/tidal marsh, in both ideal and simulated sediment profiles. Under ideal conditions, CAR rates were 50 g $C_{org}$ m$^{-2}$ yr$^{-1}$ and 240 g $C_{org}$ m$^{-2}$ yr$^{-1}$ in seagrass and mangrove/tidal marsh sediments, respectively. While this overall model structure was used in all simulated scenarios, MAR and CAR rates under ideal conditions varied from those reported above in increasing sedimentation and OM decay simulations to represent real increases in accumulation, changes in OM content and associated losses of sediment mass with depth (Table 3)".*

Text in the Results section where Figure 5 is referenced (page 9, lines 11-13) now reads: *"The estimated deviations in accumulation rates from those expected under ideal conditions are shown in Figure 5 for seagrass and mangrove/tidal marsh ecosystems. These deviations are driven by variations in MAR estimates caused by anomalies in [210]Pb concentration profiles as the $C_{org}$ fraction $\left(\frac{\sum_{n=i}^{t} (\%C_{org_i} \cdot m_i)}{m_t}\right)$ was considered to be the same in both ideal and simulated sediment profiles."*

7. Table 4 is too long and repetitive; there must be a way to condense it, since the options for each outcome are the same.

Response: We agree with the reviewer.
Actions: We have presented the information in Table 4 using a diagram rather than a table. The diagram can be found in Figure 6 of the revised version of the manuscript (page 42).

8. I found the boxes helpful, except for Box 4, which is different from the others and not necessary in my opinion.

Response: We agree and we have removed Box 4 from the current version of the manuscript.

9. I understand the logic of including the methods in an appendix – mostly because they are quite long and detailed. But it is important for the reader to understand what the authors are doing.

The authors might consider including in the methods a more detailed description than what is there now (but still less detailed than in the appendix).

Response: Since the manuscript is already long and dense, the addition of a description of each simulation would only repeat what is in the appendix. We have added more information on Table 3 that summarizes each simulation, while also including this in the Methods section. We also agree on adding a short text to mention the simulations that were conducted and how dating models were applied.

Actions: We have added further details in section *2. 1 Numerical simulations* in the revised version of the manuscript (page 6, lines 30-31; page 7, lines 1-14). Text now reads:
*"Ideal profiles were then altered to simulate the following processes/scenarios: mixing (surface and deep mixing), increasing sedimentation (by 20%, 50%, 200% and 300%), erosion (recent and past), changes in sediment grain size (coarse and heterogeneous) and OM decay (under anoxic and oxic conditions, and with labile OM contribution in sediments containing 16.5% and 65% OM) (Table 3). Refer to Appendix A for a detailed description of the methodology used to conduct each simulation.*
*The CF:CS and CRS dating models were applied to the simulated excess $^{210}Pb$ profiles to determine the average MAR for the last century (Table 2). The CIC model was excluded from the simulations presented in this study because in anomalous excess $^{210}Pb$ profiles: 1) the CRS model would lead to more reasonable approaches when the flux of excess $^{210}Pb$ is constant; and 2) when that is not the case (e.g., simulations of erosion or heterogeneous grain size), determination of mean accumulation rates alone by the CF:CS model would be a more reasonable approach. The models were applied in accordance with the simulated process. For instance, MAR was determined below the surface mixed layer in mixing simulations using the CF:CS, and piecewise in those with a change in average MAR (Appendix A). However, the models were also applied considering that (1) excess $^{210}Pb$ profiles of mixing simulations were generated by increasing MAR and vice versa, and (2) erosion was not a factor in simulated scenarios (H-J). This was done to test the potential deviations in MAR and derived CAR if the incorrect process was assumed and dating models were applied."*

10. Section 2.1 doesn't seem like it should be in the methods.

Response: We agree with the reviewer that most of the information would be best located in the introduction section.

Actions: We have moved section 2.1 to the introduction as a new section *1.1. $^{210}Pb$ dating models*. The equation and methods to estimate $C_{org}$ accumulation rates, however, have been kept in the Methods section.

11. The authors mention a literature review several times, but the only detail is provided on p. 4 line 27ff. in establishing that CIC, CRS, and CFCS are the most commonly used approaches. Is this the same literature review that was used to construct Figure 2? Please clarify. Also, they probably missed some of the literature by not including the term Pb-210, which is sometimes used instead of 210Pb. (There are almost certainly more than 150 uses of 210Pb in the salt marsh literature.)

Response: The publications we used to construct Figure 2 are cited in section *3.1 Types of excess $^{210}Pb$ concentration profiles* (page 7-8) and in the caption of Figure 2. These examples are part of the literature review but more cases could be cited, especially for mixing types II, III and IV in all vegetated coastal ecosystems. We believe that the examples provided are representative of the diversity of $^{210}Pb$ concentration profiles encountered by researchers.
The web of Science$^{TM}$ search was a simple search meant to identify the dating models generally

used in vegetated coastal ecosystems, while showcasing examples of the sedimentary processes driving $^{210}$Pb distribution. We agree with the reviewer that we missed some tidal marsh and mangrove studies by not including the term Pb-210 or lead-210.

Actions:

(1) We have updated our search in the Web of Science for all ecosystems also including the term Pb-210 and lead-210. Using the keywords mangrove sediment, salt marsh/saltmarsh/tidal marsh sediment, seagrass sediment AND 210Pb/Pb-210/lead-210 produces 86, 223 and 27 results, respectively for each ecosystem.

(2) In section *3.1 Types of excess $^{210}$Pb concentration profiles* we have modified the statement in page 8 line 32 *"Our literature review reveals that various sedimentary processes might produce similar types of excess $^{210}$Pb concentration profiles."* to *"These examples identified from the literature reveal that various sedimentary processes might produce similar types of excess $^{210}$Pb concentration profiles"*.

(3) A clarification has been added in Figure 2 caption *"Figure 2. Sketch of seven sedimentary types of excess $^{210}$Pb concentration profiles in sediments from vegetated coastal habitats identified from the literature (see references included) ...".*

12. The reason for excluding the CIC method – the absence of ideal profiles – is not persuasive as currently expressed. The other methods also suffer when there are deviations from the ideal profile, which is exactly what the authors explore. Perhaps more of a justification for excluding CIC could be given?

Response: The application of the CIC model requires a monotonic decrease in excess $^{210}$Pb concentrations with depth that usually occur in lakes but rarely occur in coastal environments. Sediment disturbances like mixing or changes in the sedimentation rate may result in excess $^{210}$Pb activities leading to age reversals that prevent the construction of an age model. When initial concentrations are expected, the CF:CS model would be a better suggested approach as it might be too ambitious to calculate a detailed stepwise chronology based on often limited numbers of data points decreasing monotonically. If on the contrary, a non-monotonic decrease is caused by changes in accumulation rates and the excess $^{210}$Pb flux is constant, the best approach would be to use the CRS model, since it suffers less with non-monotonic features in the $^{210}$Pb record and is relatively insensitive to mixing (Appleby and Oldfield, 1992). Because of the general preference and widely application of the CRS model over the CIC model under varying sediment accumulation rates and because the CF:CS model would be preferred for the purpose of estimating mean sediment and $C_{org}$ accumulation rates in anomalous $^{210}$Pb profiles, we excluded the CIC model in our simulations. However, in the revised version of the manuscript we have highlighted some situations where the CIC may be preferred.

Actions:

1. Text has been added to the description of the CIC model (page 5, lines 24-28) to highlight the point explained above: *"However, the CIC model requires a monotonic decrease of excess $^{210}$Pb concentrations down-core for age-reversals to be avoided, which is rare in most vegetated coastal sediments. In that event, the calculation of mean accumulation rates alone using the CF:CS model would be a more reasonable approach, as it might be too ambitious to calculate a detailed stepwise chronology based on often limited number of data points decreasing monotonically.*

2. Text has been added in the Methods section explaining why we have excluded the CIC model (page 7; lines 6-9): *"The CIC model was excluded from the simulations presented*

*in this study because in anomalous excess $^{210}Pb$ profiles 1) the CRS model would lead to more reasonable approaches when the flux of excess $^{210}Pb$ is constant, 2) when that is not the case (e.g., simulations of erosion or heterogeneous grain size), determination of mean accumulation rates alone by the CF:CS model would be a more reasonable approach."*

Response: We do not agree, type VI show an extreme situation with almost negligible excess $^{210}Pb$ concentrations that in most of the cases will be undatable. Type VII, because of sediment erosion, might be undatable too, but some researchers may not consider erosion and date it anyhow. Therefore, we have kept the distinction of the two profile types in the revised version of the manuscript.

Response to comments by Anonymous Referee #4

The review paper presented by Arias-Ortiz discuss the use of the 210Pb dating technique to estimate the rate of mass accumulation in vegetated coastal ecosystems. Such information is indeed very important in considering the significant role of vegetated coastal habitats (tidal marsh, mangrove, seagrass) as sinks of carbon. Over the last 150 years, 210Pb is the only tool that permits to calculate sediment and carbon accumulation rates (SAR/CAR) in such environments. However, the application of the 210Pb-based method is not tricky in these environments. The authors aim to illustrate the models usually applied to calculate SAR or MAR in these setting. This article is extremely timely as there is a growing interest in better estimate C source/sink. The authors are presenting in a correct way the principle and the conditions of the 210Pb method. Although the article is mostly dedicated to the models, there are some recommendation on the 210Pb determination and a comment of the interest of additional time marker (like 137Cs) or normalisation.

We sincerely thank the reviewer for acknowledging the interest of our work as well as for his/her comments, which were very helpful in improving the paper.

1. In fact I regret that the authors do not develop the experimental section. Indeed, it would be of great interest to provide recommendations about sampling: core description, porosity determination etc.

Response to comment 1 and 4: Since questions 1 and 4 of reviewer 2 target the same issue (i.e. development of an experimental section prior to $^{210}$Pb analyses), we addressed them together below.

Discussing sampling and sample-handling is not a simple task and includes numerous aspects if it is done properly. For instance, estimation of porosity, dry bulk density, which types of corers to use, how to extrude or slice the sediment, preservation of the interface or a discussion of the analytical methods. Developing the above-mentioned aspects goes beyond the scope of the manuscript. However, some manuals/chapters dealing with all these aspects already exist, such as Brenner and Kenney, (2013) or IAEA-TECDOC-1360, (2003), and we have cited them in the revised version of the manuscript to provide the reader with additional guidelines for coring, sampling and sample-handling.

Actions: We have modified the text in section *4. Approaches and Guidelines* (page 14-15) to briefly develop some basic sampling and sample-handling procedures to achieve good $^{210}$Pb profiles. We also provide some references that the reader could use to expand on the topic such as Brenner and Kenney (2013) and IAEA-TECDOC-1360 (2003).

Text now reads (page 14, lines 16-32; page 15, lines 1-11): *"Prior to analysis, researchers can have control over some factors such as coring, sampling, or sample-handling, that can create artefacts in $^{210}$Pb profiles and therefore contribute to dating error. Guidelines for core sampling for the analysis of $^{210}$Pb and other radionuclides have been described in detail, for example, in Brenner and Kenney (2013) and in the technical report IAEA-TECDOC-1360 (2003). Some knowledge on the expected sedimentation rate is useful to decide how to section a sediment core for $^{210}$Pb measurements, as well as the length that a core must have to reach the depth of the excess $^{210}$Pb horizon. Low sedimentation rates ($\sim$1-2 mm yr$^{-1}$) and/or coarse sediments may imply that the $^{210}$Pb datable part of sediment cores is limited to the very top centimetres. In such situation, fine sectioning intervals (0.5 - 1 cm) would be required. Longer cores (of about 100 cm) should be collected if high sedimentation rates are expected (several mm yr$^{-1}$) so that the entire excess $^{210}$Pb inventory is captured and the CRS model can be applied. These can be sliced at thicker intervals without compromising the temporal resolution of the $^{210}$Pb record. If the order*

*of magnitude of sedimentation rates are not known a priori, it is best to choose fine sampling intervals (e.g, at 0.5 cm along the upper 20 cm, at 1 cm from 20 to 50 cm, and at 2 cm below 50 cm) to ensure sufficient resolution.*

*After collection, a visual description (e.g., colour, sediment texture, presence of roots, organisms or layers) of the sediments and measurement of parameters such as water content, OM and grain size are relatively low-cost actions that provide information to interpret $^{210}Pb$ distribution and the pattern of accumulation. Indeed, the type of sediment (e.g., fine vs. coarse, rich in carbonates, homogeneous or with organic debris embedded) is a factor that should be considered (IAEA-TECDOC-1360, 2003). Coarse particles or coarse-grained carbonates where excess $^{210}Pb$ is less preferentially adsorbed (Wan et al., 1993) may hinder the detection of any excess $^{210}Pb$ in vegetated coastal sediments. In such situations, the analysis of $^{210}Pb$ in the smaller sediment fraction (i.e. < 63 □m or < 125 □m) is recommended to concentrate $^{210}Pb$ and reduce the dilution effect caused by coarse fractions. This methodology has been applied in mangrove ecosystems from arid regions where excess $^{210}Pb$ flux is low (Almahasheer et al., 2017) and in Florida Bay carbonate-rich seagrass sediments (Holmes et al., 2001). Similarly, large organic material such as roots and leaves should be removed from the sediment samples prior to $^{210}Pb$ analyses as these may contribute to the dilution of the excess $^{210}Pb$ specific activity.*

*The analytical methods for $^{210}Pb$ measurements can also be chosen depending upon the type of sample. While indirect determination of $^{210}Pb$ by alpha spectrometry of its granddaughter $^{210}Po$ will provide a significant better limit if detection (< 1 Bq kg$^{-1}$), direct determination of $^{210}Pb$ by gamma spectrometry can simultaneously provide data for supported $^{210}Pb$ ($^{226}Ra$) and relevant radionuclides to validate the $^{210}Pb$ geochronologies. For a detailed description of the analytical methods and their advantages and disadvantages see for instance Corbett and Walsh, 2015 and Goldstein and Stirling, 2003)."*

Response: This has been clarified in the revised version of the manuscript.

[revised manuscript text omitted]

---

## Referee Report (RR1)

Although I am convinced of the interest of this article, the article still suffers from a moderate expertise of the subject. I have added comments/corrections on the pdf file. The major criticisms concern:

- the introduction of the excess notion: to be explained and it is also necessary to correct the figure 1.

- the introduction to the different models and assumption are not always precise, despite the text length. I would suggest the authors to be less precise and to avoid few repetitions.

- 137Cs :it is possible to determine 137Cs in southern sediments (by in that case peak is 1965)

- again I point out the need to use appropriate name (concentration→ activity; inventory I; etc see the pdf). It would make easy to read to change excess 210Pb concentration by 210Pbxs

- use realistic conditions for simulation. Inventories are often low in shallow sediments. What about the interest to simulate a sand ?

- there are options you have not consider. For example when the peculiar layer (sand for example) is identified, it is possible to subtract it to produce a corrected depth-profile on which to determine CF-CF age.

- Figures 1 and 6 to combine and see comments on the pdf

- Table 1: I am really surprised of the narrow range of SAR and MAR. Provide range and not mean.

I think this article still need improvements and a careful reading to simplify and shorten the purpose. The main message is to advice future works on the difficulties to obtain a realisable chronology of such sediments using 210Pb.

[revised manuscript text omitted]

---

## Author Response (AR2)

Berkeley, CA, 19th of October 2018

Dear Dr. Sarin,

Below, please find our response to the comments raised by Dr. Schmidt to the revised manuscript entitled "Reviews and syntheses: $^{210}$Pb-derived sediment and carbon accumulation rates in vegetated coastal ecosystems - setting the record straight", along with a description of the changes included in the revised version of the manuscript. The revised version of the manuscript is attached after the response letter.

Ariane Arias-Ortiz on behalf of the authors

Response to comments by Reviewer #4 Dr. Schmidt

I have checked the revised version produced by Ariane Arias-Ortiz et al.

Thanks Dr. Schmidt for revising and providing constructive comments to our manuscript. Please, find the responses to the concerns raised below.

Although I am convinced of the interest of this article, the article still suffers from a moderate expertise of the subject. I have added comments/corrections on the pdf file. The major criticisms concern:

- the introduction of the excess notion: to be explained and it is also necessary to correct the figure 1.
Response: We concur. Figure 1 has been corrected as suggested. The excess $^{210}$Pb concept has been further explained in the introduction and excess $^{210}$Pb has been expressed as $^{210}$Pb$_{xs}$ in the revised version of the manuscript. Text now reads (page 4, lines 4-10):

"$^{210}$Pb is a naturally occurring radionuclide with a half-life of 22.3 years. Particles that sink and accumulate in the bottom of aquatic systems scavenge $^{210}$Pb that is present in the water column due to the decay of $^{222}$Rn in 1) the atmosphere and ulterior dry and wet deposition and 2) in the water. This is known as "Excess $^{210}$Pb" ($^{210}$Pb$_{xs}$) and is added to the "Supported $^{210}$Pb" ($^{210}$Pb$_{sup}$), which is continuously produced by in-situ decay of $^{226}$Ra in bottom sediments. The accumulation of sediments over time ideally generates a decreasing distribution of $^{210}$Pb specific activity as a function of depth (or cumulative mass in g cm$^{-2}$) governed by the decay of the $^{210}$Pb$_{xs}$, as illustrated in Figure 1. The $^{210}$Pb dating models are based on the interpretation of the $^{210}$Pb$_{xs}$ rate of decline in a sediment profile.

[Figure]

Figure 1

-Revise units for the effective mixing coefficient $k_m$

Units have been revised and we have modified units for $C$ (before Bq kg$^{-1}$, now Bq g$^{-1}$) so that units for $k_m$ are correct as g$^2$ cm$^{-4}$ yr$^{-1}$ .

$$\frac{\partial C}{\partial t} = \frac{\partial}{\partial m}\left(k_m \frac{\partial C}{\partial m}\right) - MAR\frac{\partial C}{\partial m} - \lambda C \qquad \text{(Eq. 2)}$$

Units for cumulative mass (*m*) are g cm$^{-2}$ and for MAR are g cm$^{-2}$ yr$^{-1}$

- the introduction to the different models and assumption are not always precise, despite the text length. I would suggest the authors to be less precise and to avoid few repetitions.

The description of the $^{210}$Pb dating models has been revised and simplified following the reviewer's notes embedded in the pdf file (page 4-5)

- 137Cs: it is possible to determine $^{137}$Cs in southern sediments (by in that case peak is 1965)

Response: Indeed, we have specified that for sediments in the Southern Hemisphere the peak is found at 1965. We have also changed the wording in page 16, line 1, and changed "absence" for "detection".

Text in page 15 lines 23- 26.

*"$^{137}$Cs and $^{239+240}$Pu were released to the environment through the testing of high-yield thermonuclear weapons in 1950s to 1960s and can be used as chronometers in sediments, either by assuming that the peak in activity corresponds to the fallout peak in 1963 or 1965 in the Northern and Southern hemispheres, respectively, and/or that the depth of its first detection corresponds to the onset of fallout in the mid-1950s* (Ribeiro Guevara and Arribére, 2002; Stupar et al., 2014).*"*

Text in page 16 line 1-2 now reads:

*"In addition, the detection of $^{137}$Cs is more difficult in sediment cores from habitats located in the Southern hemisphere and near the Equator"*

- Again I point out the need to use appropriate name (concentration→ activity; inventory I; etc see the pdf). It would make easy to read to change excess 210Pb concentration by 210Pbxs

As suggested, all the below entries have been reviewed and substituted in the revised

version of the manuscript:
- "Concentration" has been substituted by "specific activity" since we are referring to activity per unit of mass (i.e. Bq/kg).
- The [210]Pb inventory at a particular depth has been expressed as $I_m$ instead of $A_m$, and this has been applied throughout the revised version of the manuscript.
- All excess [210]Pb entries have been substituted by [210]Pb$_{xs}$.

- use realistic conditions for simulation. Inventories are often low in shallow sediments. What about the interest to simulate a sand?
We have changed the coarse sand simulation (scenario K) to make it more realistic by assuming that the sediment profile consists of 10% silt, 20 % medium sand and 70% coarse sand, since the reviewer pointed out that some labs do not measure [210]Pb$_{xs}$ in sediments not containing any silt or clay. MAR results estimated by the CF:CS model do not change, even with the addition of a silt fraction. The deviations in MAR estimated by the CRS model have now been reduced from 15% to 10%. This has been modified in the results section *3.2.4 Sediment grain size distribution* of the revised manuscript. Figures 4 and 5 and the supplementary material Tables 6a and 6b have been also modified accordingly.

- there are options you have not consider. For example when the peculiar layer (sand for example) is identified, it is possible to subtract it to produce a corrected depth-profile on which to determine CF-CF age.
This approach would work when [210]Pb$_{xs}$ profiles are affected by a pulsed sediment event that has caused the sudden deposition of a sand layer. This option has now been included in the results section *3.2.4 Sediment grain size distribution (*page 12, lines 12-14) and in Box 1.

Text in section *3.2.4 Sediment grain size distribution (*page 12, lines 12-14) now reads:
*"Indeed, the deposition of coarse sediments may indicate exceptional increases in sedimentation in the case of storm surge deposits or pulsed sediment deliveries. In these cases, the CF:CS model could be applied if the event layer can be identified and can be subtracted to produce a corrected depth-profile from which to determine the CF:CS derived ages and mean mass accumulation rate."*

Text in Box 1 now reads:
*"If the initial [210]Pb$_{xs}$ concentration ($C_0$) is known, the CIC model could be useful to constrain dating when it is difficult to precisely define the thickness of such deposits. Otherwise the CF:CS model could be applied if the event layer is identified (e.g., using XRF, [226]Ra or granulometry) and can be subtracted to produce a corrected depth-profile from which to determine derived CF:CS ages and mean mass accumulation rates."*

- Figures 1 and 6 to combine and see comments on the pdf

Figure 1 is the [210]Pb cycle and Figure 6 is the summary of the profile types, the processes involved and the actions that can be taken to reduce errors in the estimation of sediment and carbon accumulation rates. We believe it is easier for the reader to keep these figures separated. But maybe the reviewer meant to merge Figure 2 and Figure 6?. We understand the interest in merging these two figures and therefore we have done so in the revised version of the manuscript. We have moved Figure 6 up front to substitute Figure 2.

We have addressed the comments highlighted in the .pdf by Dr Schmidt.
    - Figure 1 has been corrected and now it reads "[210]Pb scavenging" rather than "scavenging [210]Pb$_{xs}$".

- In the description of the profile types in section *3.1 Types of excess $^{210}Pb$ profiles,* we have included that type III could also be related to recent increases in MAR as the reviewer suggests in Figure 2.

- Table 1: I am really surprised of the narrow range of SAR and MAR. Provide range and not mean.

Ranges of SAR and MAR are now provided in Table 1

I think this article still need improvements and a careful reading to simplify and shorten the purpose. The main message is to advice future works on the difficulties to obtain a realisable chronology of such sediments using 210Pb.
Following the suggestions made by the reviewer in the pdf file, we have removed unnecessary wording from the text. At the end of the *Conclusions* section we have added a sentence highlighting the need to develop a strategy to obtain reliable chronologies and MAR in vegetated coastal sediments using $^{210}Pb$ (page 21, lines 21-24):

*"While attention should be paid to the limitations of $^{210}Pb$-derived results in vegetated coastal ecosystems, the guidelines provided here should help interpreting altered $^{210}Pb$ profiles obtained from vegetated coastal sediments and to develop a strategy to strengthen the evaluation of MAR and CAR."*

Additionally, we have addressed all other minor comments that Dr. Schmidt highlighted in the pdf file.
- We have mentioned the problem of using the CRS model if sediments are sieved (page 15, lines 6-9):
  *"the analysis of $^{210}Pb$ in a fine sediment fraction (i.e. < 63 $\mu m$ or < 125 $\mu m$) is recommended to concentrate $^{210}Pb$ and reduce the dilution effect caused by coarse fractions. However, the application of the CRS model would then be limited to those cases where the $^{210}Pb_{xs}$ is contained entirely in the sieved sediment fraction. Sieving combined with $^{210}Pb$ dating has been applied to mangrove sediments from arid regions where $^{210}Pb_{xs}$ flux is low (Almahasheer et al., 2017) and to carbonate-rich seagrass sediments in Florida Bay (Holmes et al., 2001)."*

- We have been more cautious regarding the use of excess $^{228}Th$ in coastal sediments, page 18 lines 7-13:

[revised manuscript text omitted]